# Size-resolved simulations of the aerosol inorganic composition with the new hybrid dissolution solver HyDiS-1.0 – description, evaluation and first global modelling results

François Benduhn[1,2], Graham W. Mann[1,3], Kirsty J. Pringle[1], David O. Topping[4], Gordon McFiggans[4], Kenneth S. Carslaw[1]

[1]School of Earth and Environment, University of Leeds, Leeds, UK.

[2]Max Planck Institute for Chemistry, Mainz, Germany.

[3]National Centre for Atmospheric Science, School of Earth and Environment, University of Leeds, Leeds, UK.

[4]Centre for Atmospheric Science, School of Earth, Atmospheric and Environmental Science, The University of Manchester, Manchester, UK.

*Correspondance to*: François Benduhn (francois.benduhn@gmail.com)

**Abstract.** The dissolution of semi-volatile inorganic gases such as ammonia and nitric acid into the aerosol aqueous phase has an important influence on the composition, hygroscopic properties and size distribution of atmospheric aerosol particles. The representation of dissolution in global models is challenging due to inherent issues of numerical stability and computational expense. For this reason, simplified approaches are often taken, with many models treating dissolution as an equilibrium process. In this paper we describe the new dissolution solver HyDiS-1.0 that was developed for the global size-resolved simulation of aerosol inorganic composition. The solver applies a hybrid approach, which allows some particle size classes to establish instantaneous gas-particle equilibrium while others are treated time dependently (or dynamically). Numerical accuracy at a competitive computational expense is achieved by using several tailored numerical formalisms and decision criteria, such as for the time- and size-dependent choice between the equilibrium and dynamic approaches. The new hybrid solver is shown to have numerical stability across a wide range of numerical stiffness conditions encountered within the atmosphere. For ammonia and nitric acid, HyDiS-1.0 is found to be in excellent with a fully dynamic benchmark solver. In the presence of sea salt aerosol, a somewhat larger bias is found under highly polluted conditions if hydrochloric acid is represented as a third semi-volatile species. We present first results of the solver's implementation into a global aerosol microphysics and chemistry transport model. We find that (1) the new solver predicts surface concentrations of nitrate and ammonium in reasonable agreement with observations over Europe, the US and East Asia; (2) models that assume gas-particle equilibrium will not capture the partitioning of nitric acid and ammonia into Aitken mode sized particles, and thus may be missing an important pathway whereby secondary particles may grow to radiation and cloud-interacting size; and (3) the new hybrid solver's computational expense is modest, at around 10% of total computation time in these simulations.

# 1 Introduction

The inorganic composition of the aqueous phase of atmospheric aerosol particles is continuously subject to exchange with the gas phase. Whereas $H_2SO_4$ condenses irreversibly under tropospheric conditions, semi-volatile species such as $H_2O$, $HNO_3$, $HCl$ and $NH_3$ may re-evaporate from the aerosol aqueous phase depending on the temperature and chemical composition of the atmosphere. $NH_3$ combines with $H_2O$ to give $NH_4OH$, which along with $HNO_3$ and $HCl$ tends to dissociate in the aerosol aqueous phase, with water taking the role of a solvent. This combination of condensation and partial dissociation is usually referred to as gas-particle conversion or dissolution.

The dissolution of semi-volatile gases into the aerosol phase has an ambiguous effect on aerosol particle size. The dissolution of $NH_3$ within acidic $H_2SO_4$ particles decreases their hygroscopicity, resulting in a decrease in water content and particle size, while chemical interaction between a dissolving acid and a dissolving base, such as $HNO_3$ and $NH_3$, may result in a substantial increase in particle size due to the considerable amount of dissolved matter and water that is bound by it. Variations in particle size and hygroscopicity affect aerosol-radiation and aerosol-cloud interactions, with influences on climate processes such as atmospheric circulation and the water cycle. Because the chemical composition of particles varies substantially with size, the effects of these semi-volatile gases are non-uniform across the particle size distribution. Dissolution affects atmospheric chemistry via its influence on atmospheric composition and also impacts aerosol heterogeneous chemistry via the aerosol surface and the pH of the aerosol aqueous phase. Finally, the dissolution-mediated modification of the aerosol radiative properties will also affect the photolysis reactions within the atmosphere.The potential of dissolved inorganic species, especially $NH_3$ and $HNO_3$, to act upon these climatologically relevant factors is high, as they are a major constituent of the atmospheric aerosol, especially in polluted areas (e.g. Adams et al., 1999, Feng and Penner, 2007, Metzger and Lelieveld, 2007, Pringle et al., 2010, Morgan et al., 2010).

The global simulation of the aerosol inorganic aqueous chemistry is computationally expensive due to the complexity of the process and the numerical stiffness property of the related differential equations. The so-far adopted approaches may be divided into equilibrium approaches (e.g. EQUISOLV, Jacobson et al., 1996, Jacobson, 1999b; ISORROPIA, Nenes at al., 1998, Fountoukis and Nenes, 2007; EQSAM, Metzger et al., 2002a, Metzger and Lelieveld, 2007), selectively dynamic so-called hybrid approaches (HDYN, Capaldo et al., 2000, see also, Trump et al., 2015); and fully dynamic approaches (MOSAIC, Zaveri et al., 2008). The motivation for equilibrium approaches is that the stiffness of the system leads to numerical instability that may involve prohibitive computational expense when integrated in time. Hybrid approaches seek to reduce the computational expense by assuming that only a fraction of the aerosol size distribution is in equilibrium with the gas phase. This fraction in equilibrium is usually the smaller end of the size distribution, which would require the shortest integration time step if treated kinetically. The remaining fraction of the size distribution is treated dynamically because the larger particles are not in equilibrium with the gas phase, as shown by theoretical studies (Wexler and Seinfeld, 1990; Meng and Seinfeld, 1996) and model investigations that demonstrate a much better agreement with observations (e.g. Hu et al., 2008).

The primary advantage of a fully dynamic approach is its accuracy, but the mathematical stiffness of the governing equations means that an appropriate numerical integration scheme must be carefully chosen. Although fully dynamic solvers may prove to be computationally efficient, numerical and system dynamical properties constraints tend to put an upper limit to the time step that may be adopted (Zaveri et al., 2008). Hybrid schemes seek to overcome this limitation, but their partial

equilibrium assumption may prevent them from fully resolving the dynamics of concurrently dissolving and chemically interacting species. In particular, due to competition effects, the composition of smaller particles may (in some conditions) also require appreciable timescales to reach equilibrium (Wexler and Seinfeld, 1992), thus contradicting what is commonly assumed by hybrid models. Enforcing equilibrium may thus lead to a misrepresentation of pressure gradients at particle surface and partitioning fluxes of semi-volatile species through the gas phase (Zaveri et al., 2008). In this study, we seek to

overcome this seemingly inherent limitation of the hybrid approach through a careful choice of the fraction of the size spectrum that is assumed to be in equilibrium. We also note that any concern about the precise accuracy of the hybrid approach needs to be balanced against its potential advantages. For example, as will be explained herein, the hybrid approach may also offer the opportunity for a more accurate representation of certain system dynamical properties, such as the species' chemical interaction. Furthermore, computational efficiency is a fundamental aspect of a global modelling

scheme, and it is therefore important that its time stepping is flexible.

This study describes and evaluates the new hybrid solver HyDiS version 1.0 for the dissolution of semi-volatile inorganics into the aerosol aqueous phase. The solver uses a hybrid mechanism that strives to combine computational efficiency with an accurate representation of the dynamical property of the process. To achieve this, the solver makes use of some existing numerical methods and concepts, such as the semi-implicit and semi-analytical integration method for dynamic dissolution

that was developed by Jacobson (1997), the mixed time integration method (Zhang and Wexler, 2006; Zaveri et al., 2008) or the analytical approach for the estimation of the equilibrium composition (Nenes et al., 1998), some of them modified fundamentally. Other methods are novel as they have been developed specifically for the purpose of HyDiS-1.0. All of these numerical methods are applied within different branches of an overall formalism that is based on the system dynamical and numerical properties of dissolution and involves a set of specific decision criteria for the distinction between the different

dynamical regimes. HyDiS-1.0 is not intended to be a complete gas-particle exchange solver, as it does not take into account the formation of a solid or mixed particle phase. Furthermore, the version of HyDis presented here does not consider semi-volatile organics, although a future development to achieve this is planned. Section 2 describes the dynamical properties of the simultaneous dissolution of several inorganic compounds and the numerical constraints these put on the design of an efficient hybrid solver. Section 3 explains the mechanism of the solver in detail. Section 4 evaluates the solver against fully

dynamical model runs in a box model configuration. Finally Section 5 presents first results from an implementation of the scheme into a 3-D chemistry transport model to demonstrate its computational efficiency and numerical reliability.

## 2 Dynamical properties of dissolution

### 2.1 Non-linear properties

To understand what turns dissolution into a tedious numerical problem, it is necessary to analyse its dynamical properties in detail. This section analyses these properties using the example of $HNO_3$ and $NH_3$ dissolving into an aqueous solution of

$H_2SO_4$. The concurrent dissolution of a base and an acid into an acidic solution is characterised by a positive feedback phenomenon involving the two dissolving species. Initially, the dissolution of $HNO_3$ is impeded via the strong acidity of $H_2SO_4$. Conversely, the continuous neutralisation of an acidic solution by a dissolving base such as $NH_3$ will eventually prevent the dissolution of further basic matter. However, in the presence of both a dissolving acid and base, the continuous neutralisation by the base may be effectively counterbalanced by the dissolving acid, thus giving way to further dissolution

of basic matter. The effective interaction between the two dissolving chemicals causes numerical stiffness, as there is one variable, in this case the pH of the solution, that is contrarily influenced by the two dissolving species.

The transition from an initially binary solution of $H_2SO_4$ and $H_2O$ to a solution in equilibrium with gas phase $HNO_3$ and $NH_3$ may be divided into 3 stages (see Fig. 1):

1) Initial neutralisation of the particle. The solubility of $NH_3$ is high, while the particle is too acidic for large amounts of

$HNO_3$ to dissolve. In an atmosphere with significant amounts of $HNO_3$ and $NH_3$ there is a momentary contrast of their equilibration times because particulate $HNO_3$ tends to be in equilibrium with the atmosphere whilst $NH_3$ partitions quickly into the aqueous phase in the presence of a large pressure gradient.

2) Efficient interaction between $HNO_3$ and $NH_3$. Both species are dissolving because the pH is high enough for $HNO_3$ and still low enough for $NH_3$ to dissolve. During this phase particle pH controls the dissolution of both species.

3) Asymptotic convergence towards equilibrium. The interaction of $HNO_3$ and $NH_3$ is reduced as each species is separately close to equilibrium with the aqueous phase.

Dynamically speaking the system is kinetically limited during Stage 1 via the contrast of the particle surface and atmospheric pressures of $NH_3$. Consequently, during this stage $NH_3$ is the driving species, while $HNO_3$ may be described as the following species. In contrast, during Stage 2 the system is chemically limited. It is the stage of effective interaction between $NH_3$ as a

base and $HNO_3$ as an acid. The pressure contrast between the aqueous and the gas phase of both $NH_3$ and $HNO_3$ is substantial enough for the interaction to be fast. Stage 3 is also chemically limited, however both species are close to equilibrium. In contrast to the preceding, it is the stage of ineffective interaction between the acidic and the basic species, as their effective dissolution is hampered by the lack of pressure contrast. As will be seen in the next section, the specific dynamical properties of each stage also entail specific numerical issues.

### 2.2 Numerical stiffness properties

A chemical system may be said to be numerically stiff if its interacting variables show disparate equilibration times, such that the time step of numerical integration has to be adapted to those variables that drive the system and vary quickly (e.g.

Zaveri et al., 2008). Here, we choose to generalize the concept of numerical stiffness to the following *ad hoc* definition: A system of one or more variables is numerically stiff when its dynamical properties require an integration time step that is small in comparison to the amount of time that is required for its transition to equilibrium.

Following this definition each of the above stages of equilibration of the particle aqueous phase with the gas phase may be associated with a specific form of numerical stiffness, as follows:

1) During Stage 1 the following species' equilibration time is much shorter than the one of the driving species. This property points to the usual definition of numerical stiffness, except that the species with the shorter equilibration time is not driving the system. A time step that is too large causes consecutive over- and undershoots for the following species, which may result in oscillating model results. Furthermore, under some conditions the oscillations may grow from time step to time step, and may eventually produce non-physical values.

2) During Stage 2 the system is evolving rapidly in the presence of moderate vapour pressure gradients and efficient chemical interaction. A scheme of numerical integration that catches the interactive nature of the system may still misrepresent its dynamics when the time step is inappropriately large. Figure 1 shows a clear deviation of the large time step values from the small time step values during this stage, such that inaccurate model results may be obtained if the transition time through this stage is sufficiently large relative to the integration time step of the model.

3) The numerical stiffness associated with Stage 3 may be described as follows: Each species taken individually shows an equilibration time that is short relative to the entire transition period. As explained for Stage 1 and shown by Fig. 1, this may result in an oscillatory behaviour, with the inherent risk of non-physical values. As the chemical interaction between the dissolving species is considerable, the propensity for oscillation is substantial and more pronounced than within Stage 1, as shown by Figure 1.

4) Finally, numerical stiffness may arise as follows. During Stage 3, the system is interacting chemically, albeit the aqueous phase is typically close to its equilibrium composition. However, effective chemical interaction may also set in early on in connection with low vapour pressure gradients. The resulting situation is thus a hybrid between Stage 2 and 3. We found this form of stiffness to occur in the context of elevated $HNO_3$ and $NH_3$ concentrations that result in the formation of ammonium nitrate particles. During the formation process of ammonium nitrate the fraction of dissolving species remaining within the gas phase typically becomes very low. However, in the presence of size-resolved aerosol particles that are in disequilibrium among each other, their equilibration timescale will become very long, as the equilibration flux needs to transit through the bottleneck of low gas phase concentrations, thus resulting in further numerical stiffness. In Figure 1 the concentrations of ammonia and nitric acid are too low for this variety of numerical stiffness to occur. The artefacts obtained remind those associated with the numerical stiffness that may occur during Phase 2. Whereas the third variety of numerical stiffness may arise on its own, the fourth variety always transitions slowly to the third one.

# 3 Solver description

In this section we give a comprehensive description of v1.0 of the HyDiS dissolution solver, explaining in detail how it simulates the reversible dissolution of semi-volatile inorganic species into the aerosol aqueous phase. HyDiS-1.0, like all hybrid dissolution solvers, includes decision criteria to determine whether the exchange of semi-volatiles between the gas phase and particle phase should be solved dynamically or treated as an equilibrium process. Accordingly, the solver consists of modules to calculate both dynamic and equilibrium formulations of dissolution, as well as an overarching routine which specifies the controlling decision framework.

The HyDiS-1.0 dynamic and equilibrium sub-solvers are described in sections 3.1 and 3.2, with the decision framework detailed within section 3.3. An alternative dynamic scheme is presented within section 3.4, which allows the bias in equilibrium solutions in relationship with computational efficiency considerations to be reduced. Finally, section 3.5 provides an overview on the entire mechanism, linking the formalism of the solver to the numerical stiffness properties of dissolution.

*Naming conventions:*

The reversible condensation and dissolution of semi-volatile species into the aerosol aqueous phase is simulated with the HyDiS-1.0 dissolution solver. For reasons of convenience, these species will henceforth be qualified as *dissolving* species.

In the following equations, subscript letters refer to aerosol size class i, dissolving species j, non-volatile species k (or a species whose volatile nature is not considered at a particular point), at a particular time step or iteration number n, and at a time t. Exceptionally they may be placed as superscripts in order to differentiate from various additional variable attributes. Exponents occur as numbers only.

## 3.1 Dynamic dissolution

The flux of dissolving molecules *n* onto a particle surface element *dS* is (e.g., Pruppacher and Klett, 1997):

$$d\left(\frac{\partial n_j}{\partial t}\right) = \nabla \cdot \left(\frac{D}{kT}\nabla p\right)dS \, ,$$

**Eq. 1**

where $D$ $[m^2 s^{-1}]$ is the Brownian diffusion coefficient in the gas phase corrected for condensation in the dynamic regime and for sticking efficiency, $k$ $[J\ K^{-1}]$ is the Boltzmann constant, $T$ $[K]$ is the absolute air temperature, and $p$ $[Pa]$ is the partial pressure of the diffusing species.

Assuming pseudo-equilibrium and constant temperature within the volume of air within which diffusion takes place, one obtains after integration over particle surface:

$$\frac{\partial c}{\partial t} = 4\pi r D N \left( C - \frac{p_S}{kT} \right),$$

where $c$ $[m^{-3}]$ is the aerosol aqueous phase number concentration of the dissolving species (molecule number per unit volume air), $r$ $[m]$ is the radius of the aerosol particles, $N$ $[m^{-3}]$ is the aerosol particle number concentration, $C$ $[m^{-3}]$ is the gas phase number concentration of the dissolving species, and $p_S$ is the vapour pressure of the dissolving species at the particle surface.

Surface partial pressure may be related to the number concentration of dissolved molecules via the so-called dimensionless Henry coefficient $H'$, which for a mono-acid is defined as follows (Jacobson, 1999a):

$$H' \equiv \frac{ckT}{p_S} = \frac{N N_A m_w}{\gamma_{HA}^2 [H^+]} kTH ,$$

where $m_w$ $[kg]$ is the aerosol water mass per particle, $N_A$ $[mol^{-1}]$ is the Avogadro constant, $\gamma_{HA}$ $[-]$ is the mean molal activity coefficient of the dissolving monoacid $HA$, $[H^+]$ is the molal proton concentration in the aerosol aqueous phase and $H$ $[mol^2$ $kg^{-2}$ $Pa]$ is the Henry constant of the dissolving mono-acid given by:

$$H = \frac{\gamma_{HA}^2 [H^+][A^-]}{p_S}.$$

Similar expressions may be derived for a dissolving base.

In the preceding expression, the partial dissociation property of the dissolving species is neglected, which is an underlying assumption for HyDiS-1.0. In so-doing, the number of degrees of freedom of the considered chemical system reduces by one for each dissolving species. Other properties, such as the species' chemcial interaction, can then be taken into account more throroughly, as analytical solutions may be derived more readily. We have seen in the preceding section that the chemical interaction between acids and bases plays an essential role to the numerical stiffness property of the system.

In contrast to the Henry constant, the dimensionless Henry coefficient does not express a physical law. Its value expresses the ratio between the partial pressure of the dissolved molecules if they were evaporated and their actual surface pressure. These values are unequal due to chemical interaction among the species that make up the aerosol aqueous phase, and the resulting partial dissociation of the dissolving acid or base. As such the dimensionless Henry coefficient is not a constant, but varies as a function of the pH and the mean activity coefficient. This feature will turn out to be important when Eq. 2 is numerically integrated in time.

Within the framework of a discretised representation of aerosol particle sizes, dissolution of several species may take place onto several aerosol size classes simultaneously. If only one species is considered and the chemical interaction of several species is neglected, then an implicit semi-analytical solution may be derived when the following equation that results from the combination of Eq. 2 and Eq. 3 is considered:

$$\frac{\partial c_i(t)}{\partial t} = 4\pi r_{i,t} D_{i,t} N_{i,t} \left( C_{t+\delta t} - \frac{c_i(t)}{H'_{i,t}} \right).$$

Eq. 5

Combining the semi-analytical solution of the preceding equation (Jacobson, 1997):

$$c_{i,t+\delta t} = H'_{i,t} C_{t+\delta t} + \left( c_{i,t} - H'_{i,t} C_{t+\delta t} \right) \exp\left( -\frac{4\pi r_{i,t} D_{i,t} N_{i,t}}{H'_{i,t}} \delta t \right)$$

Eq. 6

10 with the mass balance equation:

$$C_{t+\delta t} + \sum_i c_{i,t+\delta t} = C_{tot}$$

Eq. 7

yields for $C_{t+\delta t}$ (Jacobson, 1997):

$$C_{t+\delta t} = \frac{C_{tot} - \sum_i c_{i,t} \exp\left( -\frac{4\pi r_{i,t} D_{i,t} N_{i,t}}{H'_{i,t}} \delta t \right)}{1 + \sum_i H'_{i,t} \left( 1 - \exp\left( -\frac{4\pi r_{i,t} D_{i,t} N_{i,t}}{H'_{i,t}} \delta t \right) \right)}.$$

15                                                                                                           Eq. 8

Although the preceding set of equations is unconditionally stable, as shown by inspection for $\delta t \to \infty$, $c_i = H'_i C$ and $C = C_{tot}$ /$(1+\Sigma H'_i)$, such that no unphysical values may occur, it should be noted that the convergence to a static equilibrium between the aerosol aqueous phase and the gas phase is not unconditionally ensured: the solution given by Eq. 6 and Eq. 8 is disconnected from the pH of the particle aqueous phase, as the dimensionless Henry coefficient is held constant. In addition,

20 the simultaneous dissolution of several species affects the mean activity coefficient, which is also held constant. The integration time step may thus not be chosen arbitrarily otherwise oscillatory behaviour may occur.

The preceding semi-analytical solution is therefore best used under conditions of relatively quick variations of the gas phase concentrations and a relatively stable pH of the aerosol aqueous phase, as for instance in the presence of a large amount of sulphuric acid. In the event of a monoacid dissolving into a particle with a highly variable particle pH and a relatively stable gas phase concentration, the following semi-analytical solution may prove to yield more stable results:

$$c_{i,t+\delta t} = \frac{\lambda_1 + \lambda_2 B(\lambda_1,\lambda_2)\exp((\lambda_2 - \lambda_1)\delta t)}{a(1 + B(\lambda_1,\lambda_2)\exp((\lambda_2 - \lambda_1)\delta t))},$$

**Eq. 9**

with:

$$B(\lambda_1,\lambda_2) = \frac{ac_{i,t} - \lambda_1}{\lambda_2 - ac_{i,t}},$$

$$a = \frac{k_{i,t}\gamma_{HA}^2}{\alpha^2 HkT}, \quad \alpha = N_{i,t}N_A M_w, \quad k_{i,t} = 4\pi r_{i,t}D_{i,t}N_{i,t}$$

$$\lambda_{1,2} = -0.5 \cdot b +/- \sqrt{0.25 \cdot b^2 + ac}$$

$$b = a\sigma_{i,t}, \quad c = k_{i,t}C, \quad \sigma_{i,t} = \left(\sum_k n_k A_{k,i,t} - \sum_k m_k B_{k,i,t}\right)$$

**Eq. 10**

Eq. 9 is a semi-analytical solution to the generic differential equation $dx/dt = -ax^2 - bx + c$ resulting from the combination of Eq. 2, Eq. 4 and the ion balance equation given by:

$$H_{i,t}^+ = c_{i,t} + \sum_k n_k A_{k,i,t} - \sum_k m_k B_{k,i,t},$$

**Eq. 11**

where $c_i$ denotes the dissolving monoacid for which Eq. 9 is solved, and $A_k$ $[m^{-3}]$ and $B_k$ $[m^{-3}]$ stand for any other anions $A^{n-}$ and cations $B^{m+}$ that are present in the aerosol aqueous phase, respectively.

In accordance with Eq. 4, the dissolving monoacid is assumed to dissociate entirely. The aqueous phase of the atmospheric aerosol contains in general a variable fraction of sulphuric acid whose degree of dissociation should be taken into account when calculating particle pH. Within the preceding equation $OH^-$ is presumed to be negligible relative to $H^+$. Similarly to the negligence of the partial dissociation property of the dissolving species, this assumption is an underlying simplification within the framework of the hybrid solver described in this paper. A model $H^+=0$ is thus to be associated with an actually neutral pH=7. In this context Eq. 6 may yield a negative concentration of $H^+$, both via an evaporating acid and a condensing

base, as the variation of the pH is not taken into account within the normalised Henry coefficient $H'$. In this context, which adds to the numerical stiffness property's requirements, the choice of an appropriate time step $\delta t$ is all the more essential.

Both the choice between Eq. 6 and Eq. 9, and the choice of an appropriate time step require an appropriate criterion that stands for a representative variation of the gas and aqueous phase compositions and/or the typical amount of time to reach that variation. In the framework of these equations, which neglect chemical interactions among several species dissolving concurrently, the characteristic variation or time scale of the aerosol aqueous phase is inherently specific to each dissolving species. Consequently, the time step that will ultimately be chosen must not exceed the one that is characteristic of the species that for some specific reason is chosen as the most relevant one. The specific upper time step limit to be used in conjunction with the above equations should therefore fulfil the following condition:

$$\delta t_{j,i,t} = \kappa_{\delta t}\, t_c^{j,i,t}, \quad \kappa_{\delta t} \leq 1,$$

**Eq. 12**

$\kappa_{\delta t}$ being the numerical time step criterion for dissolution in the dynamic mode. In this study we choose $\kappa_{\delta t}=1.0$. The characteristic equilibration time $t_c$ of the aerosol particles contained in size class $i$ with respect to a dissolving and supposedly non-interacting species $j$ may be related to an approximate equilibrium composition, solving:

$$\left(\left[H^+\right]_{eq}^{i,j,t}\right)^2 + \left(\alpha_{i,t} - \sigma_r^{i,j,t}\right)\left[H^+\right]_{eq}^{i,j,t} - \alpha_{i,t}\sigma_r^{i,j,t} - \frac{\alpha_{i,t} A_{tot}^{i,j,t}}{\beta_{i,t}} = 0$$

$$\sigma_r^{i,j,t} = \frac{\sigma_{i,t}}{\beta_{i,t}}, \quad \alpha_{i,t} = \beta_{i,t}\frac{HkT}{\left(\gamma_{HA}^{i,t}\right)^2}, \quad \beta_{i,t} = N_i N_A m_w^{i,t}$$

,

**Eq. 13**

where $\sigma$ is the molal proton concentration in the aerosol aqueous phase as given by the ions $A_k$ and $B_k$. Eq. 13 gives the equilibrium proton concentration in the aerosol aqueous phase following dissolution of the monoacid $HA_j$. Note that $\sigma_r$ is conserved, as the particle water mass, the mean activity coefficient of the dissolving species and the degree of dissociation of sulphuric acid that is required for the estimation of $\sigma_r$ are supposed to remain constant as an approximation. The preceding equation is obtained when Eq. 4 and Eq. 11 are inserted into the aerosol size class relevant mass conservation equation of the dissolving species:

$$A_{tot}^{i,j,t} = C_{j,t} + c_{i,j,t}.$$

**Eq. 14**

The approximate equilibrium concentration $c_{eq}^{i,j,t}$ may then be obtained using Eq. 11.

Considering Eq. 5, the amount of time to reach that equilibrium naturally exceeds:

$$t_c^{i,j,t} = \frac{1}{4\pi\, r_{i,t} N_{i,t} D_{j,t}} \cdot \frac{\left| c_{eq}^{i,j,t} - c^{i,j,t} \right|}{\max\left( C_{j,t}, \dfrac{c_{i,j,t}}{H'_{i,j,t}} \right)} \cdot$$

**Eq. 15**

As indicated by Eq. 6, the equilibration time is determined by an asymptotic variation of the amount of dissolved molecules: when the solution is getting closer to equilibrium, the pressure gradient, which acts as driving force, diminishes by the same amount. It is our purpose to assess for each individual species a characteristic time interval that is representative of the kinetic constraints to dissolution. This time interval clearly cannot be infinite. We have seen above that Stages 2 and 3 of the particle aqueous phase equilibration correspond to a period of effective or ineffective chemical interaction that is driving the evolution of the pressure gradient. The individual species' kinetically limited equilibration time is illustrated by the time interval of Stage 1. Its order of magnitude is generally not obtained as a function of the pressure gradient, which may reflect the chemical interaction during Stage 2 or 3, but rather by the *potential* of the gas phase or the aqueous phase to generate a condensation or evaporation mass flux, as expressed in the above equation.

Eq. 15 defines a characteristic time interval that may serve as maximum integration time step to the dynamic dissolution solver. It reflects the physical nature of its purpose and has the additional advantage of being computationally inexpensive. Among several size classes, the smallest value needs to be chosen in order to avoid numerical instability due to competition among the size classes. Equilibration is eventually driven by chemical interaction during Stages 2 and 3, but this phenomenon cannot set in within a time interval that is smaller than the one required for individual species equilibration. For this reason it is possible to choose the maximum value among the dissolving species within one size class, and the overall integration time step reads:

$$\delta t_t = \min_i \left( \max_j \left( t_c^{i,j,t} \right) \right), \quad \delta t_t \le \Delta t,$$

**Eq. 16**

where $\delta t_t$ is the internal numerical integration time step chosen by the dynamic solver, and $\Delta t$ is the relevant external time step of the model the dissolution solver is embedded into.

The related approximate equilibrium concentration $C_{eq}^{j,t}$ and surface pressures $p_{S,eq}^{i,j,t}$ may serve to distinguish between gas and aqueous phase driven dissolution. Dissolution is assumed to be aqueous phase driven when the relative variation of the gas phase concentration is less than 1% of the relative variation of the surface pressure:

$$\frac{\left|C_{eq}^{j,t} - C^{j,t}\right|}{C^{j,t}} < \kappa_{g,l}^{i,j,t} \frac{\left|p_{S,eq}^{i,j,t} - p_S^{i,j,t}\right|}{p_S^{i,j,t}}, \quad \kappa_{g,l}^{i,j,t} = 0.01,$$

**Eq. 17**

$\kappa_{g,l}$ being the distinction criterion between gas and aqueous phase driven dissolution in the dynamic mode.

When found to be aqueous phase driven the semi-analytical scheme given by Eq. 9 and Eq. 10 for a dissolving mono-acid is preferred over Eq. 6 and Eq. 8. The more numerically stable earlier solution is only preferred when within one time increment the species-specific variability of particle pH is substantially higher than the corresponding variability of the gas phase. The aqueous phase driven solution should be avoided whenever possible, as it is computationally more expensive and does not provide a semi-analytical framework that accounts for the interdependence of the aerosol size classes.

Dynamic dissolution as given by Eq. 5 requires the dimensionless Henry coefficient (Eq. 3), which depends on particle pH and the mean activity coefficient of the dissolving species. The time dependence of the activity coefficient is relatively low but the pH may span several orders of magnitude within one time increment. The high variability of particle pH reflects the numerical stiffness properties that are typical of concurrent dissolution of chemically interacting species (see above). According to Eq. 16, the time step is chosen such that it should be shorter than the typical time interval of chemical interaction. However, in a global model, transport may perturb species concentrations in such a way as to upset the equilibration tendencies of chemically interacting species. Under this circumstance the aqueous phase may be rendered completely out of balance. The use of the approximate analytical equilibrium pH as given by the roots of Eq. 13 proved to be an efficient fix to this transport-added numerical instability issue. Instability occurs whenever the aqueous phase is predicted to lose protons, and in combination with a relatively large time increment would tend to lose more than it contains. This tendency may be easily checked by comparing the chemically driven change in protons by a dissolving base or an evaporating acid (Eq. 13) with the amount of protons available. The dynamic dissolution solver takes an implicit approach towards particle pH, via the use of the approximate equilibrium pH, rather than an explicit approach, when the proton demand exceeds half of the number of protons present:

$$\left(c_{H,eq}^{i,j,t} - c_H^{i,t}\right) > -\kappa_{pH} \cdot c_H^{i,t}, \quad \kappa_{pH} = 0.5,$$

**Eq. 18**

$\kappa_{pH}$ being the distinction criterion between implicit and explicit particle pH for dynamic dissolution.

Within this section we have given semi-analytical solutions to dynamic gas phase and aqueous phase-limited dissolution and defined a criterion that allows these regimes to be distinguished. Furthermore we have derived a characteristic equilibration time that may be used to determine an appropriate integration time step. It is based on the observation that single species equilibration time is strictly shorter than equilibration resulting from chemical interaction among several species. The overall

integration step is derived as the smallest size-specific value, chosen within the ensemble of the largest species-specific value within each size class. Finally we have defined a criterion to distinguish between the use of the momentary and estimated equilibrium pH with the gas phase driven solutions to dynamic dissolution. With these criteria and characterisitc time interval, it is ensured that dynamic solver chooses a time step that is as large as possible while numerical stability remains ensured. Still, under particular circumstances the numerical stiffness is such that the time step requirements would constrict computational efficiency. For this reason the dynamic solver can only develop its full potential in association with an efficient equilibrium solver.

## 3.2 Equilibrium dissolution

This section describes a new numerical formalism for the equilibration of the aerosol particle composition with the gas phase. The underlying principle of the solver is to use semi-analytical solutions that take into account the interactive nature of the problem as much as possible. The solver has certain *ad hoc* properties. The number of dissolving species that are linked through chemical interaction cannot exceed three. The number of particle size classes should not exceed the typical framework of a modal representation of the aerosol size distribution, that is 3-4 size classes. These properties come as a limit to its flexibility, however, they help optimize the accuracy and the computational expense of the scheme considerably.

In analogy to the dynamic solver, distinction is made between a regime of gas phase-limited equilibration and a regime limited by chemical interaction. In terms of the equilibrium solver, gas phase-limited equilibration corresponds to an initial stage of approximate equilibration with large variations to the dissolving species in both phases. The formalism allows for a succession of quick iterations delivering an approximate solution. Chemically limited interaction is handled during a second stage. It is both formally and numerically more complex, and therefore computationally more expensive. The equilibrium solver is thus divided into two independent sub-solvers that are linked by appropriate decision criteria.

### 3.2.1 Gas phase driven equilibration

The gas phase driven equilibration sub-solver uses a variational method. For each dissolving species, particle size classes are treated conjointly. Chemical interaction, water content, sulphuric acid dissociation and activity coefficients are taken into account via simple iterations. The resolving equation for single species dissolution into one aerosol size class is quadratic in $[H^+]$ (see Eq. 13). Due to this quadratic dependence, there is no analytical solution for multiple size classes, so the quadratic dependence has to be approximated with a partial linearisation, as follows.

The ion and mass balance equations, and Henry's law read variationally, for a dissolving acid:

$$\delta C_{j,n} = -\sum_i \delta c_{i,j,n}$$

$$\delta c_{i,j,n} = \delta c_H^{i,n}$$

$$H_j = \frac{\left(c_{i,j,n-1} + \delta c_{i,j,n}\right)\left(c_H^{i,n-1} + \delta c_H^{i,n}\right)}{r_{i,j}\left(C_{j,n-1} + \delta C_{j,n}\right)}, \quad r_{i,j} = \left(\frac{\gamma_{HA,i}^2}{kT_i N_A^2 N_i^2 M_{w,i}^2}\right)^{-1}$$

**Eq. 19**

The previous expressions for the Henry's law and the ion balance may be combined using the following linearisation assumption:

$$\left(c_{i,j,n-1} + c_H^{i,n-1}\right)\cdot \delta c_{i,j,n} + \delta c_{i,j,n}^2 \approx \delta c_{i,j,n}\cdot\left(c_{i,j,n-1} + c_H^{i,n-1} + \delta c_{inv}^{i,j,n}\right),$$

**Eq. 20**

yielding:

$$\delta c_{i,j,n} = \frac{H_j r_{i,j}}{c_{i,j,n-1} + c_H^{i,n-1} + \delta c_{inv}^{i,j,n}}\,\delta C_{j,n} + \frac{H_j r_{i,j} C_{j,n-1}}{c_{i,j,n-1} + c_H^{i,n-1} + \delta c_{inv}^{i,j,n}} - \frac{c_{i,j,n-1} c_H^{i,n-1}}{c_{i,j,n-1} + c_H^{i,n-1} + \delta c_{inv}^{i,j,n}},$$

**Eq. 21**

where $\delta c_{inv}$ is the invariant variation of the dissolving species in the aqueous phase following its equilibration with a constant gas phase.

Consistently, $\delta c_{inv}$ may be assessed solving the square equation resulting from Eq. 19 for $\delta C_{j,t}=0$. Eq. 21 may then be inserted into the mass balance expression of Eq. 19 leading to a solution of the type $\delta C_{j,t}\,\Sigma(1+a_i)=\Sigma b_i$.

For a dissolving base an equivalent expression to Eq. 21 is reached by analogy to Eq. 19. However, here the non-linear relationship between the gas and the aqueous phase at equilibrium does not arise via the second degree relationship $\delta C_j=f(\delta c_j^2)$ but rather from $\delta C_j=f(1/\delta c_j)$. Under this circumstance linearisation is obtained when the variation of the gas phase counterpart in the denominator of the following expression is neglected:

$$\delta c_{i,j,n} = \frac{-c_{i,j,n-1} + H_j r_{i,j}\left(C_{j,n-1} + \delta C_{j,n}\right)c_H^{i,n-1}}{1 + H_j r_{i,j}\left(C_{j,n-1} + \delta C_{j,n}\right)},$$

**Eq. 22**

with:

$$H_j = \frac{c_{i,j,n-1} + \delta c_{i,j,n}}{r_{i,j}\left(c_H^{i,n-1} + \delta c_H^{i,n}\right)\left(C_{j,n-1} + \delta C_{j,n}\right)}, \quad r_{i,j} = \left(\frac{K_{H2O}\,\gamma_B}{kT\gamma_H}\right)$$

**Eq. 23**

Via the combination of Eq. 22 with the mass balance equation (see Eq. 19) the variation of the gas phase due to a dissolving base may then be obtained in similar fashion.

The variational approximations developed in this section resemble analytic solutions to the equilibration of a non-chemically interactive species dissolving into a size-resolved aerosol. Due to their approximate character these methods require iteration for each dissolving species notwithstanding that they assume certain variables to be constant and neglect chemical interaction among the dissolving species.

### 3.2.2 Chemically driven equilibration

Gas phase-limited dissolution is driven by the partial pressure gradient between the particle surface and the gas phase, and is relatively independent of the non-linearities due to chemical interaction among several dissolving species. The application of a computationally efficient variational solver that is based on iterations proves to be advantageous under this circumstance. The same does not apply to chemically limited dissolution for which numerical instability may easily occur via the pH, and the number of iterations required may turn out to be very elevated due to numerical stiffness. Analytical solutions (Nenes et

al., 1998) offer the advantage of being unconditionally stable and computationally inexpensive. In the context of the concurrent dissolution of several species into a size discretized aerosol they nevertheless have several drawbacks. (1) For two or three dissolving species the equilibration of the aerosol aqueous phase requires the analytical solution of an equation of the third and the fourth degree, respectively (see below). The high degree of precision that is necessitated by equations of such an elevated degree may not be readily obtained for numerically stiff systems. Similarly, the number of dissolving

species whose chemical interaction may be fully taken into account may not exceed three, as no analytical solution is readily available to an equation beyond the fourth degree. (2) In the presence of a non-linear system, a comprehensive analytical solution may not be obtained (see above), entailing the need for iterative treatment, if the equilibrium composition is to be determined with a high degree of confidence. (3) It is not possible to solve analytically for chemically interacting species within several aerosol size classes (see previous section), which adds to the need for iterative treatment.

In the following, the derivation of the resolving equation of equilibrium pH is described for the example of several acidic species. Similar equations may be derived for any combination of dissolving bases and acids. Activity coefficients, particle liquid water content and the degree of dissociation of sulphuric acid are not predicted by the analytical scheme, being instead included as parameters (for one dissolving species, the dissociation of sulphuric acid is taken into account analytically, however). The variables are therefore the gas phase and aqueous phase concentrations of the dissolving species within one

size class and the particle pH within that class. We are thus dealing with a system of *2n+1* equations, *n* being the number of dissolving species. The governing equations are equivalents of Eq. 19, these read generically:

$$(1) \quad A_j = \frac{x_i y_{i,j}}{Y_j}$$

$$(2) \quad Y_j^{tot} = Y_j + y_{i,j} + \sum_{l \neq i} y_{l,j},$$

$$(3) \quad x_i = \sum_{j}^{n} y_{i,j} + \sum_{k} a_{i,k}$$

**Eq. 24**

where *a* stands for the anions that derive from presumably non-volatile species, like sulphate and bisulphate, and the cations and anions that result from sea salt dissociation and are assumed to remain in the liquid phase (according to model setup, see below). Eq. 24(1) stand for the Henry's law (c.f. Eq. 4), equations (2) for the mass balances (c.f. Eq. 7) and (3) for the ion balance (c.f. Eq. 11). The sea salt ions may be grouped together as they will not contribute to the variability of the pH during equilibration.

The resolving equation for particle equilibrium pH is obtained when Eq. 24(1) are solved for $Y_j$ and then inserted into Eq. 24(2). These in turn are then solved for $y_{i,j}$ which are then inserted into the ion balance equation. One obtains then a polynomial for $x_i$ (=H$^+$) whose degree is equivalent to the number of dissolving species plus one (*d=n+1*). When the above system is solved for any of $Y_j$ or $y_{i,j}$ without solving for $x_i$ before, the resolving equation is of degree *d=n+2*. This stresses the primordial importance of particle pH for chemical equilibration as it acts as a linkage among the concurrently dissolving species. It is possible to include sulphuric acid dissociation in the above system (with constant activity coefficients only). Under that circumstance the degree of the resolving equation for $x_i$ is *d=n+2*. In order to limit computational expense and to limit the degree of the resolving equation to four (*d<5*) in the presence of three dissolving species, sulphuric acid dissociation was not included in the analytical equilibrium solver described here, except when the solver equilibrates for only one dissolving species.

**3.2.3 Equilibrium solver implementation**

The implementation of the preceding formalisms of chemically and gas phase driven equilibration into a unified equilibrium solver requires an effective criterion of distinction between these two regimes. The variational formalism allows for quick equilibration within the size-discretized aerosol. When equilibrium is almost reached in terms of the individual species' pressure gradient, the system becomes driven by chemical interaction, and the efficiency of the formalism decreases rapidly. For this reason an appropriate distinction criterion between chemical and gas phase driven equilibration is:

$$\kappa_{c,g}^{j} = \frac{\left|C_{j,n} - C_{j,n-1}\right|}{C_{j,n-1}}.$$

**Eq. 25**

The minimum value for gas phase driven equilibration chosen in this study is $\kappa_{c,g}=0.1$. When equilibration is initiated in the gas phase driven mode, $\kappa_{c,g}$ decreases with each iteration. Once the threshold is reached, equilibration is switched to the

chemically driven mode, upon which $\kappa_{c,g}$ increases again as chemical interaction will trigger higher exchange fluxes. In order to avoid oscillations between the chemical and the gas phase mode, a switch back from chemically to gas phase driven equilibration is formally excluded.

The equilibrium solver follows an iterative scheme. Both the gas phase driven and the chemically driven equilibration mechanism do not account for the variability of the activity coefficients, the particle water content and, in most

circumstances, the degree of dissociation of sulphuric acid. Within the chemical sub-solver, equilibration for these variables is carried out on an *internal* level of iterations. The maximum number of internal iterations was set to 5, as a number that reconciles the need for numerical stability and the limitation of computational expense. The chemically driven scheme solves equilibrium for all dissolving species within one particle size class. For this reason an *external* level of iterations is required that accounts for equilibrations among the size classes. Its maximum number was set to 20, which was found to be sufficient

under the numerically stiff conditions that are typical to chemically driven dissolution. In general there is an inclination for the smaller particle size classes to have a lower condensation sink than the larger ones. For this reason, the larger size classes eventually tend to act as process drivers although their equilibration requires more time. The chemically driven equilibrium scheme iterates consequently in the reverse size order. The gas phase driven scheme solves for all aerosol size classes simultaneously. Limited chemical interaction as reflected in the variability of certain variables like the activity coefficients

and the water content may be jointly tackled within a common iteration level. The iteration level of the gas phase driven scheme is therefore formally identical to the external level of chemically driven equilibration, and the associated total number of iterations is also limited to 20.

Chemically driven equilibration at the internal iteration level is dominated by the variation of pH. For this reason a representative criterion of convergence at this level is:

$$\kappa_{conv,\mathrm{int}}^{i} = \frac{\left|c_{H}^{i,n} - c_{H}^{i,n-1}\right|}{c_{H}^{i,n-1}}.$$

**Eq. 26**

A sufficient degree of equilibration is assumed to be reached at the internal level when $\kappa_{conv,int}<0.1$. At the external level the degree of convergence is estimated with the following criterion:

$$\kappa_{conv,ext}^{i,j} = \frac{\left| c_{i,j,n} - c_{i,j,n-1} \right|}{c_{i,j,n-1}}.$$

**Eq. 27**

In this study convergence is assumed to be reached when $max\left(\kappa_{conv,ext}^{i,j}\right) < 10^{-3}$. Under the circumstance of several competitive aerosol size classes and pronounced chemical interaction, the quantity of dissolvable matter in the gas phase may become very limited (see above). The resulting numerical stiffness sharply increases the amount of external iterations necessary for equilibration under the chemically driven scheme. A criterion to diagnose this numerically stiff equilibration situation is:

$$\kappa_{conv,stiff} = \frac{max\left(\kappa_{conv,ext}^{i,j}\right)^{n-1} - max\left(\kappa_{conv,ext}^{i,j}\right)^{n}}{max\left(\kappa_{conv,ext}^{i,j}\right)^{n-1}}.$$

**Eq. 28**

When $\kappa_{conv,stiff}$ is found to be inferior to 0.1, then convergence among size classes is assumed to be inhibited by slow transition of the dissolving species through the gas phase and the external convergence criterion Eq. 27 is increased from $10^{-3}$ to $10^{-2}$. An increase of the convergence criterion reduces the precision of the equilibrium solver, and in consequence appears to affect the accuracy of the hybrid solver as a whole. Dynamically speaking, it turns out that this need not be the case. Knowing that this type of numerical stiffness comes with a sharp elongation of the transition period to equilibrium, some of the size classes that the solver attempts to equilibrate will not reach equilibrium within the overall time step of the model. This circumstance will be taken into account, as their composition will be corrected separately according to the concept of pseudo-transition, which is described in the following section.

### 3.3 Hybrid solver implementation

The formal combination of the dynamic and equilibrium solvers requires the definition of an appropriate decision criterion for distinction between these two regimes. The reduction of computation time is the compelling reason for the preference of a hybrid formalism over a fully dynamic one, which would obviously be the more accurate one. According to Eq. 16, the time step of the dynamic solver tends to be much more limited by individual size classes, among which the smallest value is chosen, than by the individual species in terms of their characteristic equilibration time. It is therefore advantageous to assume a maximum number of size classes to be in equilibrium. Among the size classes considered not to be equilibrium, some species may still be set to equilibrium. Although this latter choice would not allow for an increase of the time step of dynamic dissolution, as the maximum characteristic time interval is chosen for each size class, it would still have a positive influence on numerical stability if it targets the most numerically stiff species. From its underlying principle, this approach may be considered to essentially be a minimalistic version of a mixed time integration method (Zhang and Wexler, 2006;

Zaveri et al., 2008), with the temporal evolution of some species simulated explicitly while others are assumed to equilibrate instantaneously, the distinction based on considering their specific characteristic time interval.

The characteristic time interval for dynamic dissolution is tailored to the numerical stability requirements of the dynamic dissolution solver. It differs from the actual equilibration time, as it does not take into account chemical interaction, and appears to be quite specific, as it does not consider the actual partial pressure gradient. It will now be argued why the decision criterion between the equilibrium and the dynamic regime may follow a similar approach. First, the pressure gradient is only a momentary snapshot of the saturation state the particle is in. Chemical interaction actually determines the equilibrium time in many if not most cases. Typically, during most of the process of equilibration a strong gradient will be conserved in time. The gradient will only become smaller once the solution is close to equilibrium. Second, pronounced chemical interaction requires small time steps due to its related numerical stiffness. It should therefore be avoided as much as possible, and the corresponding size classes should be put to equilibrium. Their composition would have to be corrected by other means in order to ensure that the solver is as accurate as possible.

The ideal criterion of choice between the dynamic and the equilibrium solver should therefore:

1) Determine the size classes that are clearly in equilibrium due to their actual equilibration time being much shorter than the overall time increment of the model.

2) Determine the dissolving species that are clearly in equilibrium among the remaining size classes.

3) Identify circumstances of pronounced chemical interaction whose dynamical treatment would entail prohibitive computational expense.

The distinction criterion may therefore be stated as follows:

$$\kappa_{teq}^{i,j} = a \cdot t_c^{i,j},$$

**Eq. 29**

where $a$ is an *ad hoc* proportionality constant. Then, a sufficient condition for the particle in size class $i$ to be in equilibrium with respect to the species $j$ would be:

$$\kappa_{teq}^{i,j} < \Delta t,$$

**Eq. 30**

where $\Delta t$ is the overall time step of the model the dissolution solver is imbedded into. The proportionality factor $a$ within Eq. 29 has a double physical and numerical meaning, as it represents the extent of chemical interaction beyond individual species equilibration that should be taken into account dynamically, and the maximum number of internal time steps that one is willing to accept, considering a balance between the computational efficiency and accuracy requirements of the solver. In

this study *a=2.0*, such that the number of internal time steps would be limited to two at this point. The complete formalism of the solver will further complicate this picture.

A complementary choice criterion between the dynamic and equilibrium solver is introduced as follows. When an aerosol size class is put into the equilibrium mode, its influence on the mass balance of the dissolving species is disconnected from the ones kept in the dynamic mode. Due to their formal separation a choice must be made on the order in which dynamic and equilibrium dissolution are calculated. In this study the dynamic solver is carried out first on grounds of the tendency that the corresponding size increments have the larger condensation sink. Furthermore, from a dynamical point of view, it is plausible that faster reacting particles adapt to slower ones rather than the other way around. In consequence, the influence of the equilibrium size classes on the mass balance should be kept as low as possible, as given by:

$$\kappa_{meq}^{i,j} = \frac{\left| c_{eq}^{i,j} - c_{i,j} \right|}{c_{tot}^{j}},$$

**Eq. 31**

where $\kappa_{meq}$ is the distinction criterion between the dynamic and equilibrium mode by reason of mass balance considerations. In this study mass balance conditions are supposed to be fulfilled when $\kappa_{meq} < 0.1$.

A size class is put into equilibrium mode when it fulfils both the mass balance and the equilibration time criteria with respect to all the dissolving species it contains. The mass balance criterion may thus lead to an increase of the number of time steps required by the dynamic solver, as some size classes that may be found to be dynamically close to equilibrium may not be found so in terms of their mass. As the mass balance criterion does not catch chemical interaction either, as it also follows the gas phase driven approach, the size classes that are numerically stiff are still effectively filtered out, and the overall computational efficiency is preserved.

Decision on which species are placed into the equilibrium regime within a size class that is otherwise treated dynamically follows an analogous approach. However, due to numerical stability considerations, the equilibration time criterion is applied exclusively under this circumstance, and only those species may be put in equilibrium that do not act as chemical driver within the size class under consideration. The chemical driver to dissolution is defined to be the species that shows the longest equilibration time. For computational efficiency, equilibrium species are treated non-iteratively using the analytical solutions that have been derived for chemically driven equilibration (see above).

Size classes in the dynamic mode are rechecked after each internal time step against the remaining fraction of the overall time step. In consequence, the equilibration time criterion is adapted sequentially to the remaining integration time interval via $\kappa_{teq} < \Delta t - \delta t$, where $\delta t$ stands for the cumulative amount of time that has been integrated over so-far. Through this procedure, the maximum number of time steps required by the dynamic solver may still increase by one after each time increment. In practice, however, the probability for this to happen several times is very low as the characteristic equilibration

time $t_c$ is formulated in a way that it is relatively invariant (see above). The number of time steps required by the dynamic solver thus typically does not exceed three in the absence of mass balance constraints.

If an aerosol size class is put into the equilibrium mode, it is kept on hold for treatment by the equilibrium solver until the dynamic solver has finished. It might seem appropriate to redirect these size classes to time-resolved dissolution, on the basis of regular rechecks of their dynamical statute after each time increment of the dynamic solver. However, such a procedure would be inconsistent, as those classes previously chosen to be in the dynamic mode would have evolved in time in the meantime. On the other hand, it is possible to return equilibrium species within a size class to the dynamic mode, as in this circumstance dynamic and equilibrium dissolution have been carried out simultaneously.

### 3.4 Pseudo-transition correction

On grounds of the above criterion (3) for distinction between the dynamic and the equilibrium mode, size classes that are numerically stiff are set to equilibrium, notwithstanding their actual dynamical state. In order to correct for the consequent bias the following formalism is adopted. For every dissolving species the equilibration time is estimated after each external iteration increment of the chemical sub-solver. The equilibration time considered here is not equivalent to the characteristic time interval for dynamic dissolution $t_c$, but rather stands for the actual species specific equilibration time in a framework of effective chemical interaction that is marked by low pressure gradients. Eq. 2 may provide an estimation of the actual equilibration time $t_{eq}$:

$$\kappa_{pt}^{i,j,n} = t_{eq}^{i,j,n} = b \cdot \left(4\pi\, r_i D_j N_i\right)^{-1} \left| \frac{c_{eq}^{i,j,n} - c_{i,j,t-1}}{C_{j,t-1} - \dfrac{p_S^{i,j,n}}{kT}} \right| ,$$

**Eq. 32**

where $\kappa_{pt}$ is the distinction criterion for numerically stiff size classes in the *pseudo-transition mode* (see below), and $b$ is a proportionality constant that takes into account the variability of the pressure gradient during equilibration. In this study, we choose *b=1.0* as a first approximation. This value may be roughly justified as follows: (1) under circumstances of chemically driven equilibration, the pressure gradient tends to be relatively constant, and (2) a certain amount of the temporal variability of the pressure gradient is already being taken into account due the fact that $\kappa_{pt}$ is updated after each external iteration, thus allowing for competition between size classes.

If the equilibration time is found to exceed the overall time step $\Delta t$ for more than one of the dissolving species, then the species showing the largest excess is chosen as the relevant driver. For the driving species the following linear correction is made:

$$c_{i,j,t} = c_{i,j,t-1} + \frac{\Delta t}{t_{eq}^{i,j,n}} \left( c_{eq}^{i,j,n} - c_{i,j,t-1} \right)$$

**Eq. 33**

The non-driving species are then equilibrated to the newly estimated value of the driving species with the full chemically driven equilibrium sub-solver including internal iterations. This process is re-initialised at each external iteration of the chemical sub-solver, such that it becomes formally part of the equilibration process, and is repeated until full convergence. Size classes whose time-resolved transition to equilibrium is mimicked with the above *a posteriori* correction method are henceforth said to be in the pseudo-transition mode.

### 3.5 Overview

In the previous sections we have described the numerical mechanisms that make up the new inorganic dissolution solver. Due to its hybrid nature, the solver is divided into a dynamic and an equilibrium sub-solver. The equilibrium solver allows for an additional pseudo-transition correction for size classes that are not treated with a fully dynamic approach by reason of computational efficiency.

The equilibrium solver is partially based on an analytical approach, which was shown to be computationally efficient by previous modelling experience (e.g., Nenes et al., 1998). The analytical approach is chosen whenever dissolution is found to be chemically driven via effective interaction of the species contained in the aerosol aqueous phase. In this study, the analytical approach is followed as rigorously as possible, as the equilibrium particle pH is computed for the concurrent dissolution of several species. The degree of the resolving equation is equal to the number of dissolving species plus one (the latter standing for $H^+$, $OH^-$ is neglected in the ion balance equation), thus limiting the number of dissolving species that may be taken into account to three.

The dynamic solver is principally based on the semi-analytical approach followed by Jacobson (1999a). It has the advantage of solving simultaneously for an unlimited number of particle size classes, thus providing for their mutual competition for condensable matter in the gas phase. However, this formalism cannot account for the chemical interaction between the species. Dissolution may be very close to equilibrium for certain particular species, while it may be not for certain other species, which actually serve as driving species (c.f. numerical stiffness category 1). Furthermore, dissolution may also be numerically stiff for the driving species via the variability of particle pH (stiffness category 2). Therefore a species-selective equilibrium assumption is made and a predictive (=implicit) formalism for aqueous phase pH is used, respectively.

The basic functioning of the hybrid solver is depicted by the flow chart shown in Figure 2. To begin with, a characteristic time interval is estimated for each particle size class in the model. A size class is found to be in equilibrium when its characteristic time interval corresponds to less than half the integration time step of the aerosol microphysical model the solver is embedded into, and when its equilibration requires less than 10% of the total available matter for each of the

dissolving species. The characteristic time interval reflects the amount of time that would be required for the equilibration of a size class with respect to a particular dissolving species, thus neglecting additional equilibration time requirements due to chemical interaction. This definition ensures that size classes that show numerical stiffness according to categories 3 and 4 are mostly treated by the equilibrium solver. For those size classes that are treated dynamically, time integration is performed

at a time step that is as large as possible while numerical stability is still ensured. The time step is chosen according to the requirements of the size class that is closest to equilibrium. In case more than one time step is required, each of the dynamic mode classes is retested whether they can be put into the equilibrium mode. After typically 1-3 dynamic time steps the composition of the equilibrium classes is calculated. In choosing to calculate equilibrium composition after the dynamic calculation finished, two goals are pursued. First, equilibrium size classes tend to consume less matter than dynamic classes,

as they are ideally close to equilibrium, and hence smaller. This circumstance is of some relevance, because their mass balance is decoupled from the dynamic size classes, thus carrying the risk of artefacts due to misrepresentation of mutual competition for condensable matter. Second, it is ensured that those size classes that come close to equilibrium during integration may still be put into the equilibrium mode, such that both numerical stability and computational efficiency can be ensured.

Figure 3 depicts the formalism of the dynamic sub-solver for one internal time step. First, each size class is tested for whether certain species may be assumed to be in equilibrium. This test is carried out in accordance with the above time criterion for distinction between equilibrium and dynamic classes. Species that are found to be in the dynamic regime are subdivided further according to whether their equilibration is driven either by the gas phase or chemical interaction within the aqueous phase, and, in the former case, according to the variability of particle pH. As such, gas phase limited species are

integrated in time with Jacobson's semi-analytical method (Jacobson, 1999a), while aqueous phase limited species are integrated with an analytical method that provides for their larger numerical stiffness. The analytical method solves for one species in one size class, while the Jacobson method solves for one species in all size classes. The particle pH associated with the Jacobson method corresponds either to its momentary value (=diagnostic approach), or, if found to be beyond a certain variability threshold, to its individual species equilibrium value (=prognostic approach). In order to insure accurate

partitioning among the size classes, time integration is performed in parallel irrespective of the scheme that has been chosen. Finally, for the dissolving species that have been diagnosed to be in equilibrium in some or all of the classes, the composition of the dynamic classes is updated according to the analytical approach that is adopted in the equilibrium solver (see below).

The formalism of the equilibrium sub-solver is summarized within Figure 4. Using a specific criterion, the equilibrium

solver differentiates formally between so-called chemical and gas phase equilibration. In the first case equilibration is driven by chemical interaction among species dissolving simultaneously. The chemical sub-solver assesses the equilibrium composition of one size class with respect to all dissolving species using the above-described analytical approach, and iterates at an internal level for water content and activity coefficient variation, which cannot be accounted for analytically,

and at an external level for interaction among the size classes. For size classes whose equilibration is kinetically driven by the variation of the dissolving species' concentration in the gas phase, a specific solver was designed that is based on a variational method. The kinetic sub-solver presents the advantage of being computationally efficient, and solves individually for each dissolving species and simultaneously for all equilibrium size classes. If at least one size class is found to be

kinetically limited for at least one dissolving species then the kinetic sub-solver is used beforehand. The kinetic solver performs iterations with updated gas phase and surface pressures, water content and activity coefficients until further equilibration is found to be entirely chemically limited. Consequently, full equilibration is achieved with the chemical sub-solver. Size classes that are dynamically, however not chemically, close to equilibrium (numerical stiffness categories 3 and 4) are mostly tackled by the analytical solver. Especially classes that show numerical stiffness according to category 4 turn

out to have an actual equilibration time that is far longer than the individual species' equilibration time. At each external iteration of the chemical solver, the composition of these size classes is re-evaluated according to an estimation of their actual equilibration time. Size classes thus corrected are said to be in pseudo-transition and remain formally part of the equilibration process. The chemical sub-solver allows for a certain number of external iterations only. Ideally the chemical composition of the equilibrium and pseudo-transition classes converges prior to attaining the maximum number of iterations,

upon which the composition of the equilibrium classes is updated accordingly and the solver is exited.

## 4 Box Modelling Evaluation

### 4.1 Setup and method

#### 4.1.1 Modelling framework

The hybrid solver was implemented in the box model version of the modal aerosol microphysics scheme GLOMAP (Mann

et al., 2010). To facilitate a clear assessment of the operation of GLOMAP-HyDiS-1.0, the box model experiments have all microphysical processes except those specific to HyDiS-1.0 switched off. In all experiments, we assess the evolving aerosol population based on it comprising 4 hydrophilic modes (nucleation, Aitken, accumulation and coarse). The particle phase in these modes is purely liquid, consisting of aqueous $HSO_4^-$, $SO_4^{2-}$, $NO_3^-$, $Cl^-$, $NH_4^+$, and $Na^+$. $H^+$ is calculated via the ion balance, taking into account the partial dissociation of sulphuric acid, whilst nitric acid and hydrochloric acid are assumed to

be entirely dissociated. $OH^-$ is neglected all throughout the scheme. The chemical composition of sea salt is adopted from Millero et al. (2008) as Standard Mean Ocean Water (SMOW), with all cations assumed to be $Na^+$. Accordingly, the adapted composition of sea salt is $a$NaCl$\cdot b$Na$_2$SO$_4$, with $a$=0.9508 and $b$=0.0492.

Gas phase $HNO_3$, $HCl$ and $NH_3$ may dissolve into and evaporate from the aqueous phase, with activity coefficients, surface pressures and water content assessed via the Partial Derivative Fitted Taylor Expansion (PD-FiTE) aerosol thermodynamics

scheme (Topping et al., 2009). PD-FiTE was built on the concept used in the multicomponent Taylor expansion method (MTEM) model of Zaveri et al. (2005) in which activity coefficients of inorganic solutes are expressed as a function of water

activity of the solution. Unlike MTEM, PD-FiTE was designed to remove the need for defining sulphate poor and sulphate rich domains. In addition, the order of polynomials that represent interactions between binary pairs of solutes was allowed to vary to increase computational efficiency whilst retaining an appropriate level of accuracy. Fit to simulations from the Aerosol Diameter Dependent Model (ADDEM, see Topping et al., 2005a,b), the use of PD-FiTE within a dynamical framework was demonstrated for aqueous inorganic electolytes in Topping et al. (2009) and extended for inorganic-organic mixtures in Topping et al. (2012) using the Microphysical Aerosol Numerical model Incorporating Chemistry (MANIC, see Lowe et al., 2009). $H_2SO_4$ is not currently considered as a dissolving species within HyDiS-1.0. In principle, it may be treated by the solver as a chemically interacting semi-volatile species, given that the total number of dissolving species does not exceed 3, that its surface pressure would be provided by the thermodynamic scheme, and that slight adaptations to the chemical equilibrium solver are made. Alternatively, $H_2SO_4$ could be treated by the solver as a non-interacting semi-volatile species, or as a non-volatile species via a formal association of dissolution and condensation (Jacobson, 2002). Currently, $H_2SO_4$ is considered to be non-volatile, and its condensation is simulated within a separate routine (Spracklen et al., 2005).

### 4.1.2 Evaluation in the equilibrium operation

One element of the box model evaluation of GLOMAP-HyDiS-1.0 involves testing how well the equilibrium subsolver within HyDiS-1.0 compares to the benchmark equilibrium solver AIM III (Clegg et al., 1998). In these experiments the gas/particle exchange of $HNO_3$, $NH_3$ and HCl is simulated with HyDiS-1.0 operating at full equilibrium. The particle composition is set up to be size-independent, consisting initially of a mixture of sulphuric acid, sea salt and water. Relative humidity is set to 80% and temperature and pressure to standard conditions. Three sets of these experiments are carried out with the total concentrations of $NH_3$, $HNO_3$ and HCl set at equal values, being 0.1, 1 and 10 ppb. For HydiS-1.0 these total concentrations are initially confined to the gas phase. Within each set a series of experiments is performed, as the sea salt dry volume fraction is gradually increased from 0 to 100% (see Table 1). Surface total $HNO_3$ and $NH_3$ mixing ratios (over the gas and particle phase) are at most 10 ppb in polluted regions and typically around 1 ppb or less over remote oceans (Adams et al., 1999). Gas phase HCl ranges typically from 0.001 to 0.1 ppb over the Southern Hemisphere oceans (Erickson, 1999), and is less than 10 ppb under polluted continental conditions (e.g., Eldering, 1991; Nemitz, 2004). The concentration ranges for HCl, $NH_3$ and $HNO_3$ were not primarily chosen as being representative of any particular region or environment but rather with numerical stability testing considerations in mind. For the benchmark runs, the equilibrium particle compositions obtained with HyDiS-1.0 are used as input values to the online version of AIM III. As the formation of a solid phase is not simulated with GLOMAP-HyDiS-1.0, we de-activated this feature in the AIM III runs. The accuracy of the new solver's equilibrium scheme can then be evaluated by comparing the surface pressures and the particle water content obtained with the two schemes. We refer the reader to Topping et al. (2009) for an assessment of the degree of accuracy that may be obtained with the simplified thermodynamic scheme PD-FiTE.

### 4.1.3 Evaluation in the hybrid operation

The new solver's numerical reliability and its ability to reproduce the dynamics of the equilibration of the aerosol with the gas phase are investigated with two series of model experiments, one with particles within 4 modes initialised as binary mixtures of $H_2SO_4$ and $H_2O$, and the other with the finest 2 modes initialised to contain $H_2SO_4$ and $H_2O$ and the 2 coarser modes (accumulation and coarse) initially containing just sea salt and $H_2O$ (see Table 2). Within Series 1 only $HNO_3$ and $NH_3$ are allowed to dissolve, whereas in Series 2 HCl may dissolve additionally, thus providing for a more complex system with degassing HCl from the larger modes that can then dissolve also into the smaller modes. The fixed particle number concentrations within the log-normal modes are 1000 cm$^{-3}$ (nucleation), 250 cm$^{-3}$ (Aitken), 100 cm$^{-3}$ (accumulation), and 0.1 cm$^{-3}$ (coarse); the initial number median dry particle radii are 1, 25, 100 and 1000 nanometres, respectively. Five-day simulations are carried out at standard pressure and temperature conditions with an imposed diurnal temperature cycle of +-5K starting at maximum temperature at 6 pm. Relative humidity is set to 80%. The particle number concentrations and the non-volatile species are held constant for the dissolving species to converge towards a forced dynamic equilibrium. In order to assess the model evolution across a range of numerical stiffness conditions, each of the two series involves three experiments with concentrations of the dissolving species set at different values. The three experiments each follow the specifications above, but with the dissolving species and $HNO_3$, $NH_3$ and (within Series 2) HCl set to 1, 10 and 100 ppb.

We examine the results from each of these experiments, comparing between runs with (1) the full hybrid operation of HyDiS-1.0 (HYBR) choosing the dynamic or equilibrium sub-solvers according to the decision criteria, and with the pseudo-transition correction enabled, (2) the equilibrium sub-solver with the pseudo-transition correction enabled (PSEUDO), and (3) the equilibrium sub-solver without pseudo-transition (EQUIL). A benchmark for the PSEUDO and HYBR configurations is provided via the de-activation of the equilibrium sub-solver, of all equilibrium features of the dynamic sub-solver and all of its dynamic time stepping features, HyDiS-1.0 then running in fully transient mode (TRANS). The mathematical principle of the dynamic sub-solver was originally developed by Jacobson (1997). It follows a semi-implicit approach (see Eq. 5 to Eq. 8) that was specifically developed for computational efficiency, unconditional stability and relative robustness to numerical stiffness. Later on, Jacobson (2005) presented a more simple variant that does not adopt a partial analytical approach. This version is known to exhibit less of a numerical tendency to infringe the mass balance equation, to be computationally less expensive, and to be slightly less reliable formally speaking (Jacobson, 2005). We chose the earlier version for its mathematical accuracy and its related higher capacity to deal with numerical stiffness. Both variants are mathematically accurate, as for a time step $\delta t \rightarrow 0$ they tend to the exact solution. The mathematical precision of the scheme has been investigated for its later variant by Zaveri et al. (2008) against a stiff solver of ordinary differential equations, and found to deliver comparable results, provided that the time step of integration that the scheme is used with is sufficiently small. Unlike MOSAIC, HyDiS-1.0 adopts a hybrid approach, as equilibrium is assumed for certain size classes and selected species within non-equilibrium classes, and with a further simplification to apply the PSEUDO approximation in selected cases where the explicit simulation of the dynamic behaviour would require a high amount of computation time. Although

the dynamic benchmark solver is embedded within the solver, HyDiS-1.0 thus adopts a formal approach that is considerably distinct.

The embedded dynamic solver accommodates for the complex system of several chemically interacting species dissolving into a size-discretized aerosol, as it solves for all size classes concurrently, allows updating all relevant parameters after each internal time step. It therefore provides for a fast and precise benchmark if the internal time step is chosen appropriately (see Fig. 1). With a small enough time step the transition and equilibrium regimes will be accurately resolved, as the result will be graphically indistinguishable from the one obtained at a time step that is even smaller. Our analysis found that time steps of 1 and 0.01 seconds deliver accurate results for a size-discretised aerosol at 1 and 10 ppb, respectively. At 100 ppb numerical stiffness is pronounced to a degree that the Jacobson (1997) scheme cannot handle dissolution into the smaller modes unless a disproportionately short time step is chosen. For this reason, we benchmark this run against the TRANS configuration for the accumulation and the coarse modes only, at an internal time step of 0.01 seconds. The degree of precision of the HYBR and the PSEUDO runs is shown by their comparision with the benchmark TRANS run, as these should yield similar results if numerically accurate. The formal accuracy of the HYBR, PSEUDO, EQUIL and TRANS runs is mutually verified as they converge towards similar equilibrium values whenever they should do so by virtue of their system dynamical properties. The formal accuracy of embedded fully dynamic scheme is also verified by Figure 1, as the ambient and surface pressures of the dissolving species equalize at equilibrium.

## 4.2 Results

### 4.2.1 Comparison against AIM III

Fig. 5 evaluates the HyDiS-1.0 simulated equilibrium surface pressures and particle water content against the benchmark AIM III scheme as a function of the initial sea salt content at the initial concentrations of HCl, $NH_3$ and $HNO_3$ of 0.1, 1 and 10 ppb, respectively. At 0.1 and 1 ppb (Figs. 5a and Figs. 5b, respectively), and sea salt contents of 70 and 80%, the neutralisation of $H_2SO_4$ by the dissolving $NH_3$ and the degassing HCl reduces the water content of the aerosol, and would lead to the formation of $Na_2SO_4(s)$, which has been suppressed in these runs. Beyond 80% sea salt, water contents increase again as $H_2SO_4$ is much less neutralised by $NH_3$. At 10 ppb (Figs. 5c) the formation of $Na_2SO_4$ is precluded via the combined dissolution of $NH_3$ and $HNO_3$. Overall, the equilibrium surface pressures of $NH_3$ and particle water contents compare quite favorably at 0.1 and 1 ppb over the full range of sea salt content (Figs. 5a and 5b, respectively). For $HNO_3$ and HCl there is a bias, which is most pronounced at high molalities around 70-80% sea salt. At 10 ppb (Fig. 5c), the bias is most pronounced at low contents of sea salt and the associated high molalities, for both the surface pressures of HCl and $HNO_3$ and the liquid water content, while the pressure of $NH_3$ still seems to compare favourably over several orders of magnitude. The bias may not represent a limitation to the accuracy of the equilibrium solver, but rather the very pronounced sensitivity of the equilibrium surface pressure of highly concentrated and neutralised acids. At low proton concentrations, a pronounced bias in the surface pressure does not lead to a similarly pronounced bias in the liquid phase concentration, as the equilibration

occurs predominantly via a variation of the pH (see also below). Furthermore, it needs to be considered that HyDiS-1.0 uses a parameterized thermodynamic scheme. In accordance with Topping et al. (2009), for both $HNO_3$ and HCl the discrepancy increases with relative humidity, whereas for $NH_3$ the bias is larger at lower humidities. in all cases increasing with the molal concentration of the solutes. Given these relationships, the limited nature of the bias and the consistent agreement between

AIM III and HyDiS-1.0, it seems plausible that the difference between the equilibrium pressures and the water content is entirely related to the dissimilarity of the thermodynamic schemes.

### 4.2.2 Size-resolved dynamics – sulphate only particles

Figs. 6a and 6b compare the size-resolved $NO_3$ and $NH_4$ predicted by the new solver under the 1 ppb $HNO_3$ and $NH_3$ initialised TRANS, HYBR, PSEUDO and EQUIL configurations for binary sulphuric acid particles (experiment series 1).

Contents within the nucleation mode are not shown as they are negligible. The particle size and pH is also shown, however for the accumulation and the coarse mode only (Figs. 6c). For the coarse mode, the degree of saturation with respect to gas phase $NH_3$ and $HNO_3$ is also given (Figs. 6a.1 and 6b.1). Much more than to the time step, this value is related to the choice whether the temporal evolution of non-equilibrium modes is assessed with the dynamic sub-solver or with the pseudo-transition approximation. As will turn out, the 1 ppb run without sea-salt is the only one which the solver treats the coarse

mode dissolution fully dynamically. For the other 5 hybrid experiments, dissolution into the coarse mode is calculated applying (either partially or fully) the pseudo-transition approximation, illustrating the operation of the hybrid solver in conditions of numerical stiffness.

At 1 ppb without sea salt, ammonia is the driving species, dissolving quickly into the aqueous phase and partially neutralising the sulphuric acid. As a consequence, the particle pH increases initially before reducing again later once the

dissolution of the weaker nitric acid starts to occur. The accurate functioning of the equilibrium solver is demonstrated as the EQUIL particle pH is approximately –0.55 for both the accumulation and the coarse mode (Figs. 6c) , which reflects the fact that their simulated equilibrium composition is equal. By virtue of the temperature dependence of the Henry constant, nitric acid dissolves more readily at lower temperature (Kim et al., 1993; Nenes et al. 1998), so its content is maximal in the aqueous phase at 6 am. By contrast, the solubility of ammonia is not primarily determined by the temperature variability of

the Henry constant, but rather by the variability of the particle pH. For these 1ppb runs, ammonium has little diurnal variation as the sulphuric acid is not close to being fully neutralised and the tendency for $NH_3$ to evaporate at high temperature is buffered  by the degree of dissociation of sulphuric acid. Efficient chemical interaction does not set in under these conditions, with particulate ammonium around 2 orders of magnitude higher than that of nitrate.

Results obtained with the dynamic configurations are considerably different than when equilibrium is assumed. Treating the

partitioning dynamically accounts for the longer timescale for the nitric acid and ammonia to dissolve into the coarser particles. The dynamic runs predict night-time uptake to the smaller particle modes, and a subsequent slow transfer to the coarse mode as both ammonium and nitrate evaporate during the day. By contrast, this effect is missed by the equilibrium

approach. After five days the bias obtained with the equilibrium assumption is still of the order of 50% for ammonium (Fig. 6b.1-3), and of the order of 20% and 50% for nitrate (Fig. 6a.1-3) at high and low temperature, respectively. Likewise, the accumulation mode pH is still more than 30% higher than at equilibrium, while the coarse mode pH is still to increase further by about 10% (Figs. 6c.3-4). Interestingly, particle size is fairly stable, less than 5% from equilibrium for the accumulation mode, less than 1% for the coarse mode (Figs. 6c.1-2). Particle size decreases with the amount of dissolved $NH_3$ along with the neutralisation of the solution and the related decrease of its hygroscopicity. Note that all biases are relative to the benchmark TRANS run. The pronounced non-equilibrium conditions are revealed by the degree of saturation of the coarse mode (Figs. 6a.4 and 6b.4). While $HNO_3$ is close to equilibrium during the entire simulated period for all three dynamic runs, the driving $NH_3$ barely reaches 10% saturation after five days. The simulation of the coarse mode is fully dynamic with the HYBR configuration so the HYBR run is barely distinguishable from the TRANS run in Fig. 6. Runs performed at 0.1 ppb (not shown) are similar, with the solver also operating well under conditions of limited chemical interaction and numerical stiffness. Similarly, the PSEUDO configuration shows a high degree of accuracy, as the ammonium mixing ratios are very close to those of the HYBR and TRANS configurations, thus demonstrating the appropriateness of its underlying assumptions under moderately polluted conditions. Apart from their mutual consistency, the accuracy of the dynamic runs is corroborated as they tend to the solution that is provided by the equilibrium solver.

Fig. 7 shows the evolution of the simulated size-resolved particle composition, particle size and pH for the experiment with 10ppb gas phase $HNO_3$ and $NH_3$ dissolving into sulphuric acid particles in the 4 modes. As is testified by the equilibrium proton concentration of pH≈1.5 and the increase of particle size with the amount of dissolved matter, full chemical interaction sets in over the course of the 10 ppb simulation, with effective neutralisation of sulphuric acid, mixing ratios of nitrate and ammonium of the same order of magnitude, and surface vapour pressures close to saturation. While ammonia still acts as the initial driver of dissolution, the gas phase approaches pseudo-equilibrium within one day (not shown), thus confering a relatively high degree of numerical stiffness to the system. In these conditions, the fully dynamic configuration adopts short time steps while the hybrid algorithm prevalently invokes the equilibrium sub-solver due to CPU time considerations. This approximation introduces a bias (at most ~20%) into the hybrid solver, with the HYBR run (blue) now distinguishable from the TRANS run (purple) in Fig. 7. However, it is relatively uncommon for nitric acid and ammonia mixing ratios to reach 10 ppb in the troposphere, even in the most polluted areas, and our results confirm the hybrid solver is reliable in numerically challenging conditions. For the PSEUDO run, the temporal evolution of the ammonium content of the coarse mode is constantly mimicked via an approximation of its equilibration time, and a larger bias (~30%) is apparent. The equilibration time is estimated for each species individually, and does not take into account their chemical interaction. In consequence the equilibration time is over-estimated at 10 ppb, the flux of dissolving ammonia is somewhat too low and concentrations within the coarse mode are increasing too slowly. Similarly to the behaviour seen at 1 ppb, the errors incurred with the EQUIL configuration are considerable at around 50% low-bias for fine-mode ammonium and nitrate at night and ~20-30% high bias for coarse mode nitrate and ammonium depending on time of the day. Again, the results obtained with

the dynamic configurations are verified mutually via their similarity, and corroborated through their convergence to the equilibrium solution. The accuracy of the latter is verified by the equal pH in the accumulation and the coarse mode shown in Figs. 7c.3-4.

At 100 ppb dissolution is fully steered by chemical interaction as nitrate and ammonium contents are almost equal in both the gas and the liquid phase (Figs. 8a and 8b). Equilibrium particle pH increases to around 2.2 (Figs. 8c.3-4), the radius of the coarse mode increases considerably from an initial r≈1.6 μm to about 5 μm at equilibrium (Figs. 8c.2), and less than 5% of nitric acid and ammonia remain in the gas phase, resulting in a high variability in both particle pH and surface vapour pressure (the latter not shown). The resulting extreme numerical stiffness induces a slight artificial drift in the equilibrium regime (e.g., Fig. 8a.1), as the equilibrium sub-solver struggles to establish chemical equilibrium among the modes. Nevertheless, the solution determined by the equilibrium solver proves to be consistent, as the pH is resolutely similar in all the modes. Furthermore, the numerical stiffness results in a predominant invocation of the pseudo-transition approximation with the HYBR configuration. Whereas the HYBR run correctly chooses $NH_3$ as driver of chemical interaction, as is testified by the degree of saturation of the coarse mode (Figs. 8.4), and PSEUDO quickly switches to $HNO_3$ as a result of its simplified dynamical assumptions, both the HYBR and the PSEUDO runs yield very similar results, thus underlining the secondary relevance of the degree of saturation for the estimation of particle composition under conditions of pronounced chemical interaction. Also, the results obtained with the HYBR and the PSEUDO approximations are similar to those of the benchmark TRANS run, thus verifying the dynamical phenomena they predict. Both the accumulation and the coarse equilibrate very slowly, as $NH_3$ and $HNO_3$ may only evaporate very slowly from the accumulation mode and transit through the gas phase to the coarse mode due to the very sensitive suface pressures. After 5 days, the amount of dissolved matter in the coarse mode is still overestimated by the EQUIL configuration by a factor of 8, whilst uptake to the Aitken and the accumulation mode is biased low by a factor of 4. In conditions of ammonium nitrate formation equilibrium assumptions are thus susceptible to produce a significant bias across the entire particle spectrum whenever most of the dissolving species is in the aerosol phase and there is a substantial contrast in the equilibration time of the aerosol size classes. Even for these unrealistically high concentrations of $HNO_3$ and $NH_3$ of 100 ppb and the resulting numerical stiffness, both the HYBR and PSEUDO configurations prove to be reliable, remaining numerically stable and having little bias. After a period of 5 days it is still barely distinguishable for HYBR and of the order of 5% for PSEUDO.

### 4.2.3 Size-resolved dynamics – sea salt included

Fig. 9 compares Series 2 results for nitric acid, ammonia and hydrochloric acid at 1 ppb dissolving into an external mixture of sulphuric acid (nucleation and Aitken modes) and sea-salt particles (accumulation and coarse modes). The EQUIL configuration reveals a somewhat counterintuitive property, as the initial compositions of the accumulation and the coarse mode are equal, and their equilibrium compositions are not. It appears that particles memorize the origin of chloride, whether sea salt or dissolved hydrochloric acid: while the relative quantities of dissolved matter are equal at equilibrium, as is

testified by an equal proton concentration of approximately pH=3.4 (not shown), the relative amounts of chloride, as given by sea salt and hydrochloric acid, differ as these quantities depend on the respective condensation sink of the modes. The Aitken and the nucleation mode show a specific composition, as is testified by their equilibrium pH of approx. 0.75, which reflects their non-volatile sulphuric acid content. In the TRANS run, the coarse mode takes around 2 days to reach

equilibrium composition as nitric acid dissolves slowly into the aqueous phase and hydrochloric acid degases. The amount of hydrochloric acid in the accumulation mode increases as it adapts to the surplus released by the coarse mode (the inverse applies to nitric acid). This 2-day timescale for chemical equilibration of the sea salt particles is faster than the very slow equilibration of coarse sulphuric acid particles at 1 ppb (Figs. 6.3). The relatively high content of both nitrate and ammonium in sea salt particles at equilibrium indicates a much more effective chemical interaction under numerically stiff conditions.

This circumstance is also demonstrated by the similarity of the HYBR and the PSEUDO results in Fig. 9. Except for a very short initial period, the pseudo-transition approximation is constantly chosen with the HYBR configuration, as is testified by resolutely equal degrees of saturation of the coarse mode (Figs. 9.4). Nitrate serves as a driver, to which chloride and ammonium are equilibrated. Under moderately polluted conditions, the pseudo-transition approximation produces fair results, with a small bias in coarse mode nitrate and chloride of around 20 and 10%, respectively, due to some degree of

misrepresentation of the competition between these two, while the equilibration of ammonium proves to yield fairly accurate results in a context of short equilibration times along with low ammonia solubility at low particle pH. With the TRANS configuration, the degree of saturation of the coarse mode exhibits a pronounced daily cycle (Figs. 9.4). In contrast, except for nitrate, the pseudo-transition approximation assumes saturated conditions for the non-driving species. The low related biases reveal anew the secondary importance of the pressure gradient for a reliable simulation of particle composition under

conditions of pronounced chemical interaction. The bias obtained with the equilibrium assumption is much larger for chloride and nitrate (up to a factor of two), while ammonium is again in reasonable agreement. HCl and $HNO_3$ act as competitors, while $NH_3$ shows a low solubility that is conditioned by the particle pH, which in turn is relatively unaffected by the exchange process between the two acids.

Fig. 10 shows the results for sulphate and sea salt aerosols (Series 2) with the dissolving gases at 10 ppb. Under these

conditions, the previously observed chemical interaction under numerically stiff conditions is pronounced further, with very slow equilibration, and nitrate and ammonium contents much higher than in the 1ppb runs. In the process of chemical interaction the sea salt particles become acidified (not shown). Within the dynamic configurations, the diurnal temperature variation prevents the modes from ever reaching momentary equilibrium, as is testified by the degree of saturation of the coarse mode (Figs. 10.4), because the forcing exerted by temperature is faster than the equilibration of the coarse mode and

the latter is in competition with the smaller modes. For this reason, and in analogy to the 10 ppb dissolution into sulphate aerosol (Fig. 7), the TRANS run exhibits a much more pronounced diurnal variability of the accumulation and the Aitken modes (Figs. 10.1-2), which compensates for the relative inertia of the composition of the coarse mode (Figs. 10.3). As in the preceeding 1 ppb sea salt (Fig. 9), the HYBR configuration yields similar results to the PSEUDO configuration because the

hybrid solver constantly chooses the pseudo-transition approximation to avoid small time steps in the context of numerical stiffness. For the same reason, the degree of saturation of the coarse mode does not exhibit the pronounced daily cycle it does with the TRANS configuration for the non-driving species (Figs. 10.4). At 10 ppb dissolution into sulphate, the equilibration time of the coarse mode was overestimated in the PSEUDO run with respect to the driving ammonia, resulting in an underestimation of both ammonium and nitrate (magenta versus blue line in Figs. 7.3). The opposite applies here, with fine mode ammonium and nitrate slightly high biased in the PSEUDO (and HYBRID) runs compared to the fully dynamic simulation. In the presence of sea salt aerosol, nitric acid is chosen by the solver as the unique driver to dissolution, and the aqueous phase concentrations of ammonia and chloride are equilibrated to it. During the first day the solver appears to quite well catch the dynamics of the equilibration of the coarse mode. Past this point, however, the amount of all three dissolving species is overestimated in the coarse mode, thus leading to an underestimation of the contents in the Aitken and the accumulation modes. It appears then that nitric acid does not act as the sole driver, but that hydrochloric acid figures as a secondary driver. The instantaneous equilibration of hydrochloric acid to the nitrate content of the coarse mode in the pseudo-transition regime leads to its overestimation, which via chemical interaction leads likewise to an overestimation of the content of ammonium, and via competition to opposite effects in the smaller modes. The resulting bias of the hybrid solver is at most around 25% for chloride in the Aitken and accumulation mode, and up to 40% for chloride in the coarse mode. The respective biases for ammonium and nitrate are less, and in the gas phase all three dissolving species agree well throughout the simulation (not shown).

Figure 11 depicts the 100 ppb run within Series 2 with sea salt. Similarly to the 100 ppb sulphate particles, the EQUIL configuration shows a slight drift due to the extreme numerical stiffness at these very high mixing ratios. The pronounced chemical interaction property is exhibited by the fact that the particle ammonium content is almost twice that of chloride and nitrate, such that not more than approximately 1% of total $NH_3$ remains in the gas phase (Figs. 11.1), thus inducing a mechanically limited slow equilibration of the accumulation and the coarse mode. The sea salt particles are acidified as pH≈2.3 (not shown), which is remarkably close to the equilibrium pH at 100 ppb without sea salt, such that the equilibration is also chemically limited via unstable yet relatively constant surface pressures (see Figs. 11.4). While the variability of the aqueous phase ammonium concentrations looks similar to the one of 1 ppb sulphate aerosol (Fig. 6b), the phenomena are mechanistically different, for the equilibration at 1 ppb is entirely steered by the slow transition of $NH_3$ in the context of low gas phase concentrations, and thus dynamically limited, as is also reflected by the comparably low pH. The driving nature of $NH_3$ at a comparably high gas phase concentration at 100 ppb introduces a large bias in the HYBR and PSEUDO approximations, as they underestimate the equilibration time considerably. In analogy to 1 ppb and 10 ppb sea salt, and 100 ppb without, the HYBR and PSEUDO runs are almost identical, as the hybrid solver avoids small time steps with the choice of the pseudo-transition approximation. Although both make use of the pseudo-transition approximation, the HYBR and the PSEUDO configurations need not produce rigourously similar results, because HYBR may switch to the pseudo-transition approximation after a partial integration over the overall time step has occured. The bias introduced by HYBR and PSEUDO

is in excess of 50% for all dissolving species. Still, the disparity of the approximate dynamic configurations against the benchmark TRANS solution is less than in the full equilibrium case, so that the HyDiS-1.0 algorithm outperforms the equilibrium approach.

### 4.2.4 Computational expense

Computation times for the entire simulated time period of 5 days are compared in Table 3 for the box model test cases. The computational expense of HyDiS-1.0 is expressed as percentage of the time consumption of the standard GLOMAP-mode aerosol microphyiscal scheme, which comprises most notably routines for nucleation, condensation, coagulation, cloud and precipitation scavenging, sedimentation and dry deposition, mode merging and wet oxidation (see Mann et al., 2010). All standard GLOMAP microphysical processes were switched off for the dissolution tests above. Table 3 shows that for the test

cases the time consumption of the dissolution scheme amounts to a fraction of the standard GLOMAP-mode aerosol microphysical scheme only. Essentially, the computational time increases with the ambient concentration of the dissolving species along with numerical stiffness. The hybrid runs appear to require more computational time than the equilibrium runs while pseudo-transition appears to be relatively independent of ambient conditions. Two elements tend to break down the correlation between numerical stiffness and computational expense. First, the equilibrium solver diagnoses slow

convergence and limits the number of iterations accordingly. For this reason, the 10 ppb sea salt run requires more computation time that the 100 ppb run. Second, the hybrid solver comprises an equilibration mass criterion when distinguishing between equilibrium and dynamic modes (see above). When applied, this criterion may be related to relatively small time steps, which require a relatively high amount of computation time. The hybrid 1 ppb sea salt run illustrates this circumstance, as strong competition between the accumulation and the coarse mode for nitric acid requires both modes to be

treated dynamically.

## 5 First Global Modelling Results

### 5.1 Introduction

In this section, we describe the implementation of HyDiS-1.0 in the 3D global offline chemistry transport model TOMCAT (Chipperfield, 2006) as an extension of the GLOMAP-mode aerosol microphysics module (Mann et al., 2010).

The aims of this section are to (1) demonstrate that the new solver reliably delivers physically realistic results in the framework of a global 3-D model, (2) assess the extent to which the equilibrium and hybrid configurations of the solver may lead to different size-resolved partitioning of nitrate and ammonium, and (3) to demonstrate the new solver's competitive computational expense.

## 5.2 Global Model Implementation and Experimental Setup

We use the "coupled-chemistry" version of the TOMCAT-GLOMAP global aerosol microphysics model, as used in Schmidt et al. (2010), which uses the same sulphur chemistry as in Mann et al. (2010) in combination with an online tropospheric chemistry scheme, allowing for interactions between gaseous sulphur species and oxidants (see Breider et al., 2010). The TOMCAT tropospheric chemistry module provides gas phase nitric acid and ammonia concentrations, with the new solver then predicting their partitioning into the ammonium and nitrate components of each size mode. The coupled TOMCAT-GLOMAP chemistry module does not currently comprise hydrochloric acid. The wet oxidation of $SO_2$ is assessed within GLOMAP. $H_2SO_4$ is considered to be non-volatile, whether it originates from condensation or wet oxidation, it may not evaporate from the aqueous phase. HyDiS-1.0 simulates the influence of $H_2SO_4$ on the solubility of semi-volatile species via the particle pH and the activity coefficients, as the latter and the partial dissociation property of $H_2SO_4$ are assessed with the embedded thermodynamic scheme PDFiTE (see above).

Whereas in the box model simulations from section 4, only the 4 hygroscopic modes were activated, here we use the full 7-mode GLOMAP configuration that includes three insoluble modes containing hydrophobic carbonaceous and dust particles. The model no longer tracks a "sea-salt" component, instead separately tracking sodium and chloride masses in the accumulation and coarse mode, as well as nitrate and ammonium in each soluble mode, requiring an additional 10 aerosol tracers to be transported compared to the original configuration (see Fig. 12a and Fig. 12b for a comparison between the configuration of GLOMAP-mode with and without HyDiS-1.0).

The representations of the main aerosol processes are unchanged (as described in Mann et al., 2010), comprising nucleation, condensation, coagulation, cloud chemical processing, cloud and precipitation scavenging, sedimentation, dry deposition and wet removal. The model set-up routines were adapted to be consistent with the chemical species taken into account by the dissolution scheme (see Table 4). Liquid water content is calculated according to Topping et al. (2009), and particle density is assessed with a new routine that takes into account particle composition following the dissolution of inorganics. As in Mann et al. (2010), sea salt is emitted into the hydrophilic accumulation and coarse modes but the composition of sea salt is modified assuming mole fractions of 0.024, 0.512 and 0.464 for sulphate, sodium and chloride, respectively (see above). Ammonia emissions are from Bouwman et al. (1997), with $SO_2$, BC and POM emissions included from anthropogenic (Dentener et al. 2006) and biomass burning (van der Werf et al., 2003) sources. The dissolution solver is used to simulate the exchange of nitric acid and ammonium between the gas and the particle aqueous phase. Within this study, exchanges of hydrochloric acid and those that involve a solid phase may not be treated due to a lack of formal representation. Heterogeneous processes and the formation of secondary organics are not taken into account, as these are switched off. None of these processes is required with respect to the principal goal of the present global simulations, which is to verify the numerical functioning and performance of HyDiS-1.0.

In section 5.3 we present results from a 1-year simulation of the new model after 3 months spin-up. The simulations were carried out at T42 horizontal resolution (~2.8x2.8 degrees longitude/latitude) with 31 vertical levels on a hybrid sigma pressure coordinate.

The main transport time step for the model is 30 minutes, with the TOMCAT chemistry and GLOMAP aerosol microphysics each solved on a 15 minute time step. As described by Spracklen et al. (2005) and used in Mann et al. (2010), GLOMAP also includes a shorter "competition time step" of 3 minutes used when the condensation and nucleation are integrated in a process-split fashion. HyDiS-1.0 is implemented separately from these routines. It is the last process routine to be invoked within the GLOMAP aerosol model and is integrated with an overall time step of 15 minutes.

The uptake coefficient of nitric acid and ammonia are set to 0.2 and 0.1, respectively. The uptake coefficient of nitric acid is known to be strongly temperature dependent (Van Doren et al., 1990). The uptake coefficient of ammonia appears to depend significantly on both pH and temperature (Shi et al., 1999). In the ternary $H_2SO_4$, $NH_3$, $H_2O$ system, it also appears to be an explicit function of the degree of neutralisation of $H_2SO_4$ by $NH_3$ (Swartz et al., 1999). The update coefficients are thus an integral part of the interactive properties of aerosol chemistry, and the values we chose may only serve as a first approximation to a question that is treated in this study. In the context of this study, the uptake coefficient plays a role in the distinction between equilibrium and dynamic modes, as well as in the choice of the integration time step of the dynamic solver, as it determines the equilibration time. For this reason, a low uptake coefficient will tend to increase the computational expense of the solver along with numerical stiffness and the number of time steps required.

The findings of Section 5.3 have to be relativised against the absence of solid phase processes in the model. The formation of crystallized ammonium nitrate and/or crystal compounds of ammonium and sulphate is accompanied by the evaporation of ammonia and nitric acid that is in excess (see, e.g., Metzger and Lelieveld, 2007). Global model studies suggest that crystallized ammonium nitrate is mainly encountered under the cold and dry conditions of the Antarctic southern hemisphere winter, whereas the formation of ammonium sulphate particles under polluted and relatively dry conditions over the mid and low lattitude continents is mostly accompanied by the complete evaporation of particle nitrate Martin et al., 2004). In the boundary layer, about 70% of all particle nitrate was found to evaporate as it is in excess, while the remaining fraction would be about half and half in the aqueous and in the solid phase. In line with these results, the evaporation of ammonia was found to be limited to less than 10% on global average. The widespread incidence of solid and mixed phase formation shows that the liquid aerosol assumption is a rough simplification. This is particularly true for nitrate over the continents and the high latitudes. While not required for the verification of the reliability and performance of the solver, the accurate simulation of non-equilibrium effects requires an accurate representation of all gas-particle phase exchange processes. Whenever solid or mixed phase particles occur, the present results may therefore only serve as a preliminary indication for the importance of these effects.

## 5.3 Results

Fig. 13 shows the surface Northern Hemisphere distribution of annual-mean model particulate nitrate and ammonium mass concentrations compared against observations from the CASTNET/IMPROVE, EMEP and EANET measurement networks (compiled by Pringle et al. (2010) for the year 2002). Model results are obtained with the hybrid configuration of the dissolution solver. The solver delivers physically realistic amounts of particle ammonium and nitrate globally across both polluted and less polluted regions, thus demonstrating its numerical reliability within the parameter space of the atmosphere. Simulated nitrate has a substantial low bias in North America however. This inaccuracy need not be related to the model assumptions and simplifications, as there are other likely causes. The amount of nitric acid dissolving into the particle phase is highly dependent on particle pH, and thus the ability to accurately predict particulate nitrate in such sulphate-rich regions is dependent also on the amount of sulphuric acid versus ammonia (e.g., Xu and Penner, 2012). When comparing the model values to the observations, one also needs to consider the representativeness of the monitoring site in relation to the model resolution.

Figs. 14 and 15 compare size-resolved July 2003 nitrate and ammonia contents, as the left-hand and right-hand panels show results with the hybrid and equilibrium configuration, respectively. Values are shown as a molecular fraction of the sum of $Na^+$, $SO_4^{2-}$, $HSO_4^-$, $NH_4^+$, $NO_3^-$ and $Cl^-$ in the Aitken, accumulation and coarse mode aqueous phase excluding water and non-soluble species. Gas phase $NH_3$ and $HNO_3$ are also shown as volume mixing ratios, with equilibrium gas phase contents shown as the relative change from values with the hybrid configuration. The pseudo-transition configuration was also assessed, however results are not shown, as they are very similar to the values obtained in the hybrid configuration. In the hybrid run considerable amounts of nitrate occur in the Aitken mode both over the Arctic and Antarctic. The dissolution of nitric acid (Fig. 14) is highly temperature dependent and as such related to a pronounced seasonal cycle (e.g., Metzger et al., 2002b; Pringle et al., 2010). Although the Arctic is relatively warm in July and ammonia/ammonium concentrations are fairly low (of the order of 0.01 to 0.1 ppb), it may still suffice to neutralize the sulphate contained in the Aitken mode sufficiently, such that in conjunction with the relatively high relative humidity over the Arctic Sea nitrate comprises up to 90% of solutes present in particles at these sizes. The hybrid solver seems to catch the dynamics of dissolution with respect to a discretised aerosol as it predicts that the nitrate fraction is most important in the Aitken mode (Fig. 14a.1). In marine and remote regions sea salt is often present in the accumulation mode, and is therefore much more prone to the dissolution of nitric acid. However, the competing Aitken mode is faster to equilibrate such that the nitrate content of the accumulation mode remains constrained to typically less than 10% (Fig. 14a.2). The model suggests that the phenomenon would be most pronounced in the Antarctic, where the accumulation mode is dominated by sulphate. The model cannot reproduce the evaporation of excess nitric acid in a context of crystallisation of ammonium nitrate at this point, and it thus seems likely that the simulated nitrate within the Aitken mode is overestimated at the expense of the larger particles.

The importance of the dynamics for the fractionation of nitrate is demonstrated by the comparison with the equilibrium results. The equilibrium configuration results in a significantly different partitioning, as the nitrate fraction in the Aitken

mode is reduced due to efficient competition through the coarse mode (Fig. 14b.1 versus Fig. 14b.3). The accumulation mode (Fig. 14b.2) is squeezed between the Aitken and the coarse mode: While its nitrate content seemingly vanishes in the Arctic, it increases significantly in the Antarctic. However, it remains unclear to what extent the model is able to reproduce the effects that occur in this region.

The fractionation of ammonium (Fig. 15) appears to be much less dynamically driven, as sizeable amounts of ammonium are present in the Aitken, accumulation and coarse mode irrespective of the configuration of the hybrid solver. Rather, the partitioning of ammonium seems to be primarily driven by the ratio of particle sulphate to sea salt and secondarily by the total atmospheric ammonia content. In continental regions, ammonium typically accounts for more than 50% of the Aitken mode, with the notable exception of North Africa which is characterised by low ambient ammonia concentrations. This

finding should be relatively robust with respect to the prominent formation of ammonium sulphate over the continents, as the evaporative losses of excess ammonium appear to be limited overall. In marine regions, the accumulation and coarse modes are mostly dominated by sea salt, notwithstanding that ammonia concentrations are higher in the Northern Hemisphere.

The role of the sulphate to sea salt ratio is also apparent when comparing the global distribution of gas phase nitric acid and ammonia concentrations between the simulations with the hybrid and equilibrium configurations of the solver (Fig. 14a.4

versus Fig. 14b.4, and Fig.15a.4 versus Fig. 15b.4, respectively). Significant differences are apparent at high latitudes, for which it is suggested that significant nitric acid fractions would be present in the aqueous phase, and also more clearly in marine regions where particles are mostly dominated by sea salt. Over the Southern Ocean, although total nitric acid is very low, it dissolves readily into the abundant sea salt particles. The equilibrium configuration shows much lower gas phase nitric acid concentrations, by another 90% in this region. Similarly, the sensitivity of ammonia to the dynamical regime

resolved by the hybrid solver is highest in those areas in which it is scarce, while changes in its size-resolved partitioning are felt to a lesser degree. Via chemical interaction with nitrate, the ambient concentration of ammonia over the Southern Ocean is predicted to be lower by 10-25% than predicted by the equilibrium approach. At the high latitudes of the Northern Hemisphere, higher particle ammonium leads to a decrease of the ambient concentration of ammonia of typically more than 50% in the equilibrium regime.

Figures 16 and 17 show the January 2004 contents of nitrate and ammonium respectively, again left-hand and right-hand columns showing simulations with the hybrid and equilibrium configurations of the solver. These results should remain relatively unaffected if the crystallization of ammonium nitrate was taken into account, as model simulations suggest that it is not a widespread occurrence during the Northern Hemisphere winter (Martin et al., 2004). Total nitric acid and ammonia are high enough over Northern Hemisphere continents for nitrate aerosol to form within the Aitken and accumulation modes

even deep into the mid-latitudes during wintertime low temperature conditions, each comprising ~40-50% of total solute mole fraction over large parts of Siberia and Canada. The fraction of ammonium tends to decline with increasing latitude along with its decreasing total atmospheric concentration, whereas nitrate remains substantial due to more effective partitioning at lower temperatures. The Aitken mode competes efficiently for available nitrate and ammonium with the accumulation and the coarse mode, with accumulation fractions tending to be significantly lower, and substantially lower

fractions in the coarse mode, especially for nitrate. Consistently, a very pronounced seasonal cycle for nitrate is revealed by comparing the January and July global surface maps (Fig. 14 versus 16). In contrast, the seasonal cycle of ammonium is less pronounced (Fig. 15 versus 17). Its dissolution appears to be less temperature dependent, as it is the result of the superposition of the temperature and the vegetation cycle.

The comparison of January equilibrium and hybrid results predicts similar effects as those seen for July. When the equilibrium assumption is made, more nitrate partitions into coarse particles with the Aitken mode nitrate fraction reduced from typically 40-50% in continental regions to 30-40%, with similar figures occurring in the accumulation mode. Likewise, the equilibrium assumption also leads to discrepancies in the gas phase concentrations of nitric acid and ammonia. Nitric acid is most affected in areas that show either low total concentration in combination with sea salt, or high aqueous phase

concentrations in combination with a shift in its fractionation (Fig. 16a.4). Ambient nitric acid concentrations are consistently lower in the equilibrium regime, by typically 25-90%, except for limited areas in Siberia where ambient nitric acid is predicted to increase (Fig. 16b.4). The effect of the equilibrium assumption on ambient ammonia appears to be similarly related to its overall abundance, and triggered by chemical interaction with dissolved nitric acid in relationship to low temperatures and/or sea salt (Fig. 17a.4 versus Fig 17b.4). Over the Arctic, the equilibrium assumption reduces the low

predicted ammonia in the hybrid configuration by more than another 99% via the increased dissolution of nitric acid.

### 5.4 Computational expense

In this sub-section we assess the computational expense of the dissolution solver in the global model, comparing the hybrid, pseudo-transition and equilibrium configuration to a control run with dissolution disabled. Table 5 indicates the seasonally resolved computational expense for each of the solver configurations as a relative to control. The hybrid configuration is

most expensive in southern hemispheric winter and spring, which likely reflects increased occurrence of shorter time steps, matching with increases in CPU cost seen for the pseudo-transition configuration. In contrast, the equilibrium configuration is fastest at this time of year, being much slower in northern hemispheric winter, due to larger numbers of stiff grid boxes during the formation of nitrate aerosol. On yearly average, the hybrid configuration of the solver is only marginally more expensive than the equilibrium configuration but as seen in section 5.3 gives more accurate results. The pseudo-transition

configuration comes with more than double the amount of extra computation time. At the same time its seasonal dependence is much less pronounced. The extra amount of computational expense of the pseudo-transition configuration is most certainly related to the larger amount of multi-modal equilibration iterations required by this configuration, as the estimation of the composition of the pseudo-transition modes is fully embedded into the iterative equilibration process among aerosol size classes (see above).

The absolute computation time required by the three configurations is indicated in the final column of Table 5. In analogy to the approach taken by Zaveri et al. (2008), we present this time interval as normalised per grid cell, aerosol size class and time step. The present calculations were carried out on the phase 2a configuration of the UK national supercomputing resource "High End Computing Terrascale Resource" (HECToR), with 8 AMD Opteron Quad Core 2.3 GHz nodes (32

CPUs). Although we multiplied by the CPU number we realise that computation time is not proportional to the number of CPUs, nor is it inversely proportional to the number of size classes. Nevertheless, the calculation may allow a useful way to roughly compare to the cost of other published solvers. The present solver was written in a way that the number of internal time steps required by the dynamic sub-solver does not normally exceed two or three, considering that classes requiring a

higher number of internal time steps are typically in equilibrium with respect to the overall time step. In doing so, it is ensured that the internal time step of the solver tends to increase in parallel with the overall time step. Other solvers might not follow this approach, thus adding to the complexity of comparing computational expense.

Zaveri et al. (2008) obtained an average computational expense of about 125 µs on a single INTEL Xeon single-core 3 GHz CPU (without providing any further information about the system that was used), while the expense of the new solver in the

hybrid regime is less than 20 µs. However, the reader should note that MOSAIC also resolves solid phase processes, used 8 size classes rather than 4, that their time step was 5 minutes rather than 15, that the number of CPUs was one rather than 32, and that the figure given by Zaveri et al. (2008) includes the computational expense of the aerosol microphysics. For this reason, a more appropriate comparison between the MOSAIC and the HyDiS-1.0 computational expense might be obtained as the figure of 20 µs is doubled for the computational expense of the microphysics within GLOMAP to be taken into

account conservatively. Although, the above mentioned normalisation may filter some of the effects of the limitations, which may also be counterbalancing to some extent, it appears that the schemes are very dissimilar and the reader should only take these figures as an indication that the solvers' computational expense seems to roughly be of the same order of magnitude.

## 6 Conclusion

Within this paper we have presented the new dissolution solver HyDiS-1.0. The formalism of the solver allows a maximum

of three chemically interdependent species to dissolve conjointly, and combines an aerosol size selective equilibrium and dynamic approach. Depending on tailored decision criteria size classes that are diagnosed to be in non-equilibrium are treated fully dynamically, species selectively dynamically or corrected with an ad hoc approximate method that relies on the estimation of the equilibration time with respect to a pre-defined driving species. In particular, a certain number of specific numerical schemes were developed, such as an adaptive time stepping method that largely sets the time step as a function of

the overal time step of the model, and equilibrium solvers for chemically and gas-phase driven dissolution that are based on a species interactive analytical and a variational principle, respectively.

The numerical stability and accuracy of the new solver was investigated through box model experiments. In order to maximise the numerical stiffness property, the box model experiments were partially performed beyond the realistic range of atmospheric concentrations of dissolving species. Full equilibrium runs with bulk aerosol particles were found to be in good

agreement with AIM III, as the limited discrepancies may be related to the use of dissimilar thermodynamic schemes. In comparison to a benchmark fully dynamic run, results with the hybrid configuration, a size-resolved aerosol and two dissolving species show a very high level of accuracy. Similarly, with three dissolving species, the level of accuracy is high

under prevalent atmospheric conditions. For the most polluted ones, a non-negligible amount of bias is discernible, however. The bias is related to a misrepresentation of the competition effect among more than two driving species to dissolution and an underestimation of the equilibration time of the driving species in the context of concurrent chemical and mechanical limitation, for which situations we have not yet found a more accurate formalism that associates numerical stability with computational efficiency under stiff conditions. In its hybrid configuration the solver allows reproducing a certain number of remarkable dynamical phenomena, such as slow transition to equilibrium due to inter-modal competition at low gas phase concentrations or chemical interaction at high concentration, or the existence of a dynamical equilibrium under an external forcing condition, such as an imposed temperature cycle.

First results from an implementation of the solver in a global modelling environment of an aerosol and chemistry transport model have confirmed its computational efficiency and its formal and numerical reliability. The additional expense of computation time is of the order of 10% only in both the hybrid and equilibrium configuration. Despite some important model limitations, the results obtained are to the least in reasonable agreement with an inventory of measurement data under polluted conditions, and underline the relevance of the dynamic property of the dissolution of inorganic species for the accurate representation of aerosol composition. The validation of the solver against global measurement data sets and the evaluation of non-equilibrium effects to aerosol composition will be addressed in greater detail within follow-on publications. With respect to the existing model limitations, more development will be required for the aerosol inorganic composition to be simulated more accurately.

## 7 Code availability

The code for the dissolution solver, as used in the TOMCAT-GLOMAP simulations, can be made available to reviewers upon request via the GLOMAP code repository as maintained at the University of Leeds by Dr. Kirsty Pringle (K.Pringle@leeds.ac.uk) and Dr. Steven Pickering (isssjp@leeds.ac.uk).

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

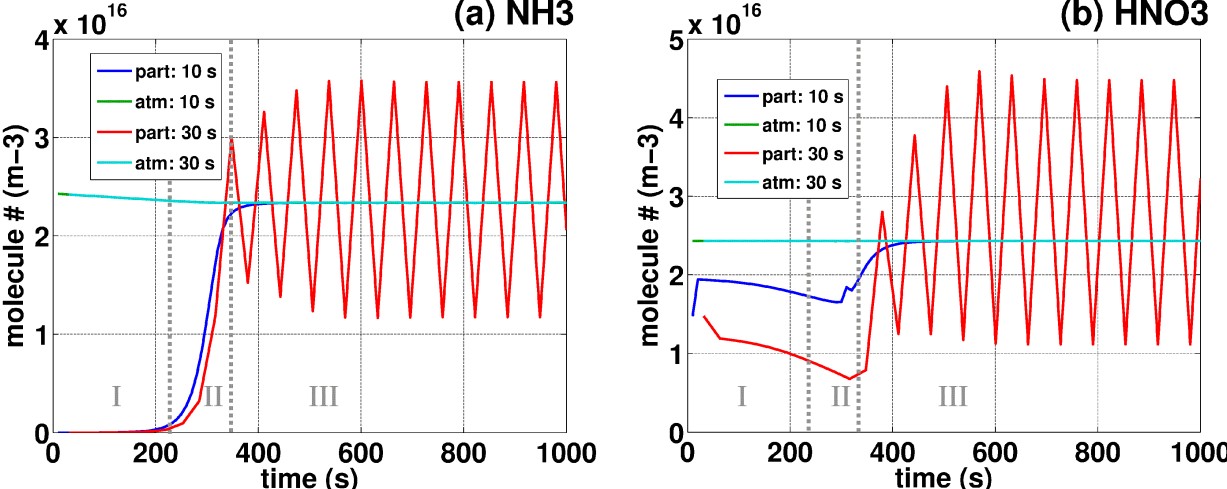

**Figure 1: Ambient (='atm') and particle surface (='part') partial pressures given as molecular number concentration equivalents of (a) $NH_3$ and (b) $HNO_3$ as a function of time, in a typical example of how chemical interaction may lead to oscillations and thus limit the numerical integration time step. For both species, initial ambient concentrations are equivalent to 1 ppb. Species are dissolving into a monodisperse aerosol at standard temperature and pressure conditions, relative humidity is 80%, dry particle size is r=50nm, their concentration is 100 cm$^{-3}$. Initially, the aerosol aqueous phase consists exclusively of a binary $H_2SO_4/H_2O$ liquid. The temporal evolution of the partial pressures is simulated with the Jacobson (1997) scheme, integrated at fixed time steps of 10 and 30 seconds, respectively. The 3 characteristic stages of the equilibration of the particle aqueous phase with the gas phase are indicated (I-III, see text). Phase 1 is equivalent to the initial 200 s of fast dissolution of $NH_3$ at almost constant surface pressure. Phase 2 corresponds to the next 200 s during which surface pressure of $NH_3$ increases along with pH, thus leading to the dissolution of $HNO_3$. Phase 3 corresponds to the oscillating period during which the system is close to equilibrium as chemical interaction has become ineffective. Note that for both $NH_3$ and $HNO_3$ the atmospheric concentrations are sensibly equal at both time steps. For the gas phase, the data obtained at a time step of 10 s (green) may thus not be distinguished from the one obtained at the larger time step (cyan).**

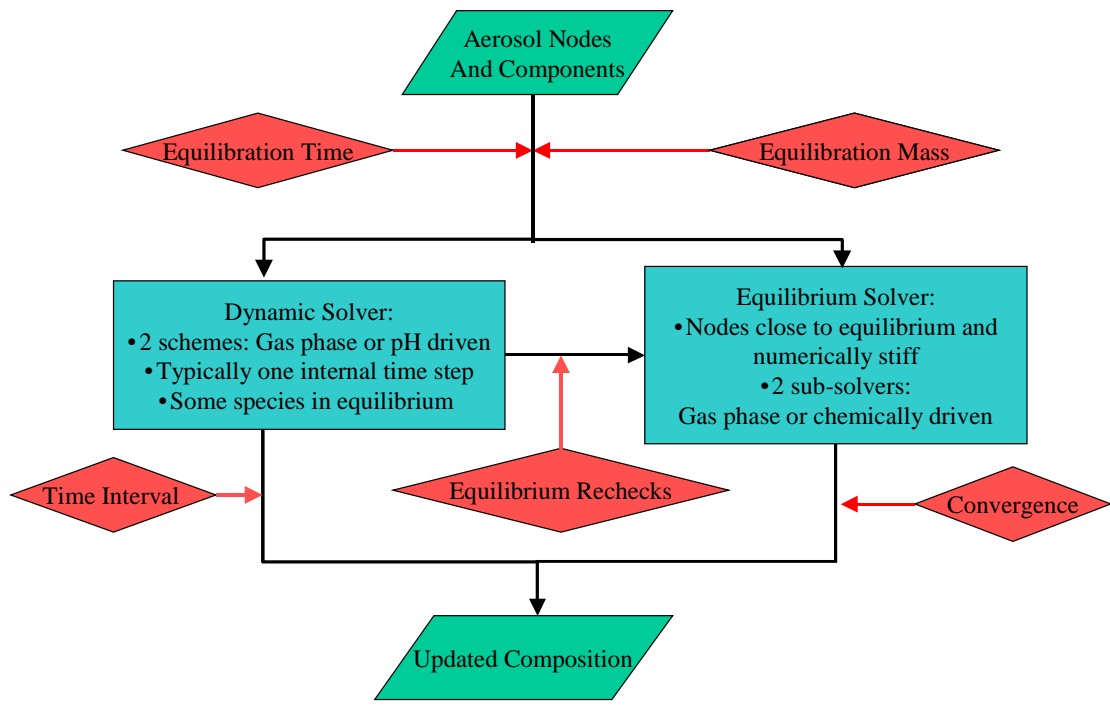

**Figure 2: Formalism of HyDiS-1.0 in its hybrid configuration.**

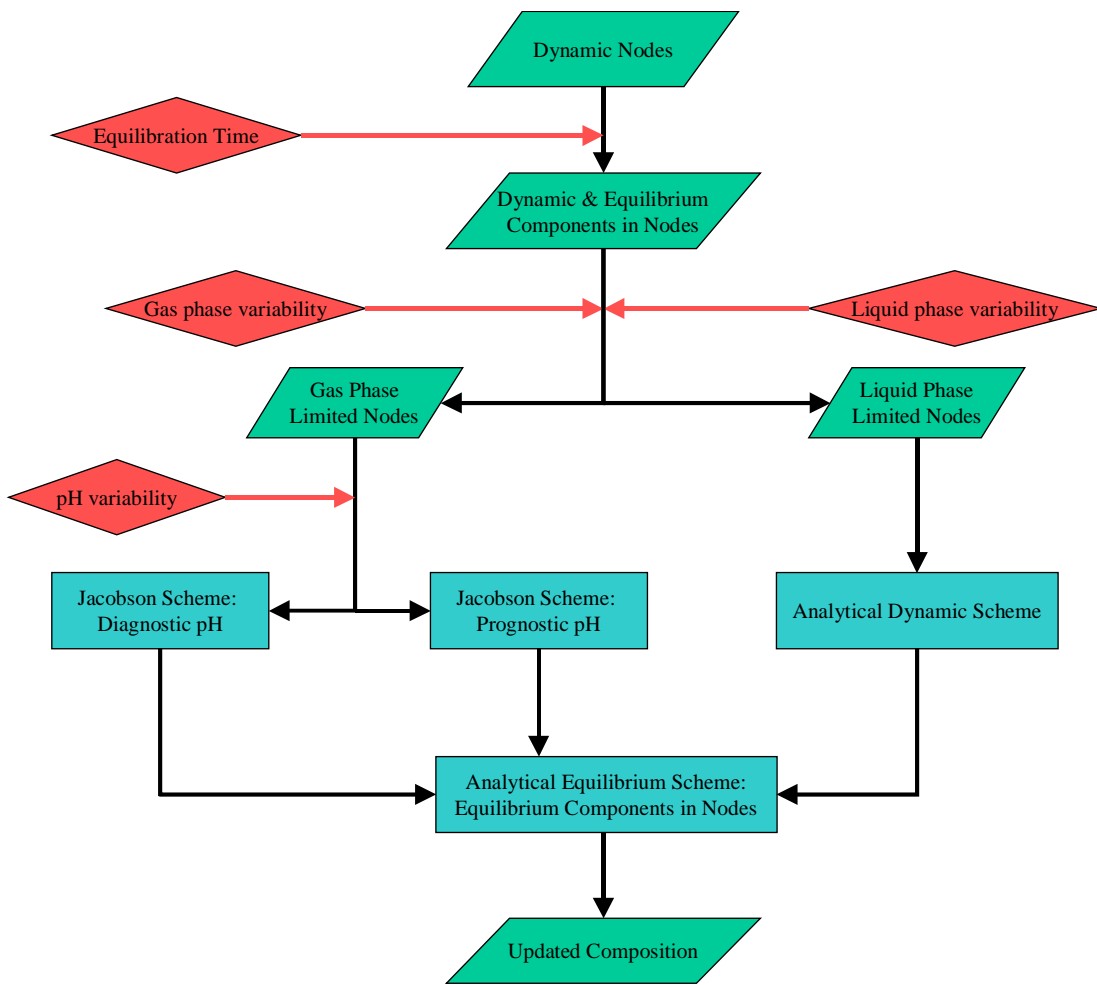

**Figure 3: Formalism of the dynamic solver.**

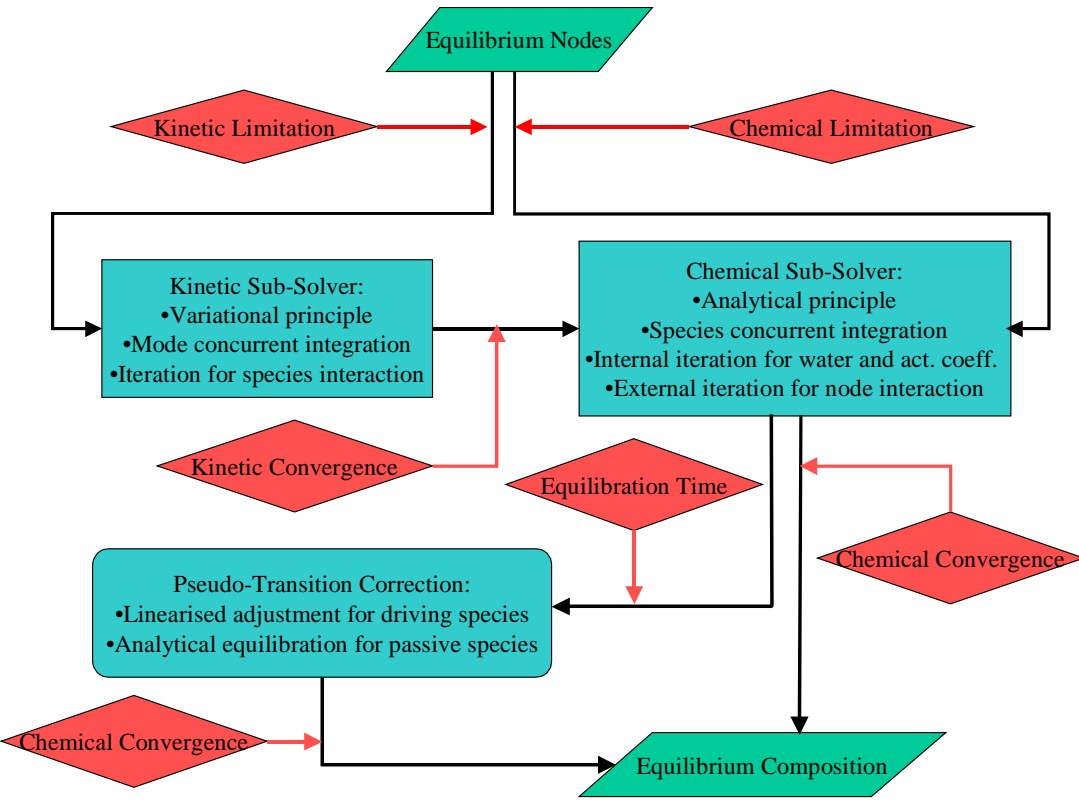

**Figure 4: Formalism of the equilibrium solver.**

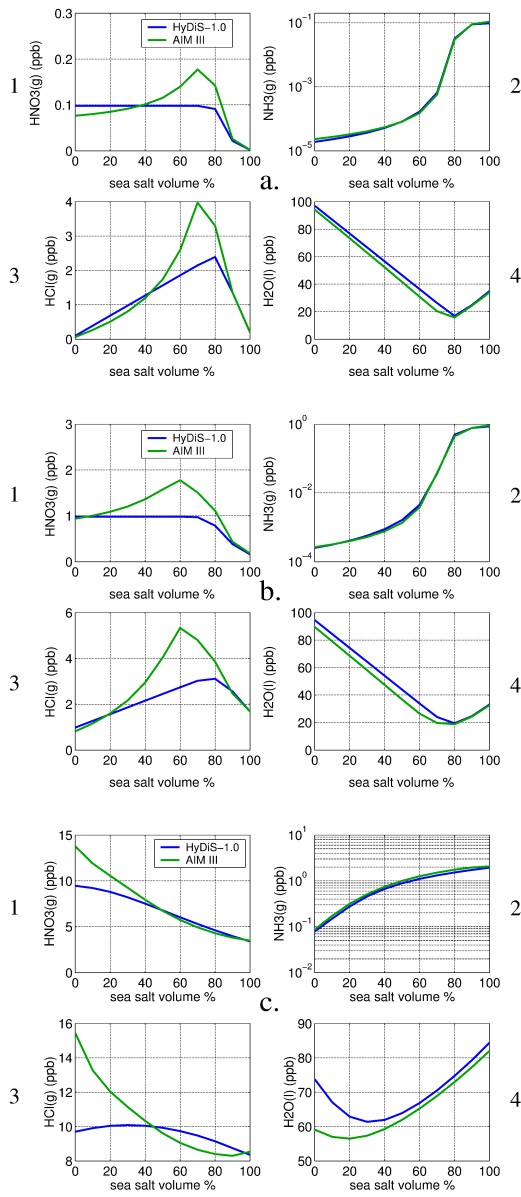

**Figure 5: Box modelling evaluation of HyDiS-1.0 against AIM III with initial mixing ratios of HNO₃(g) and NH₃(g) of 0.1 (a), 1 (b) and 10 (c) ppb. The sea salt volume fraction is varied according to the values given in Table 1. The equilibrium vapour pressures are calculated with HyDiS-1.0 and AIM III for the equilibrium particle composition assessed with the former. The vapour pressures are given as mixing ratio equivalents for HNO₃ (1), NH₃ (2) and HCl (3) along with the total particle water mass (4). Note that particle composition is not size-dependent and that due to equilibrium composition discrepancies the AIM III HCl vapour pressure may be larger than its total amount.**

# 1 ppb

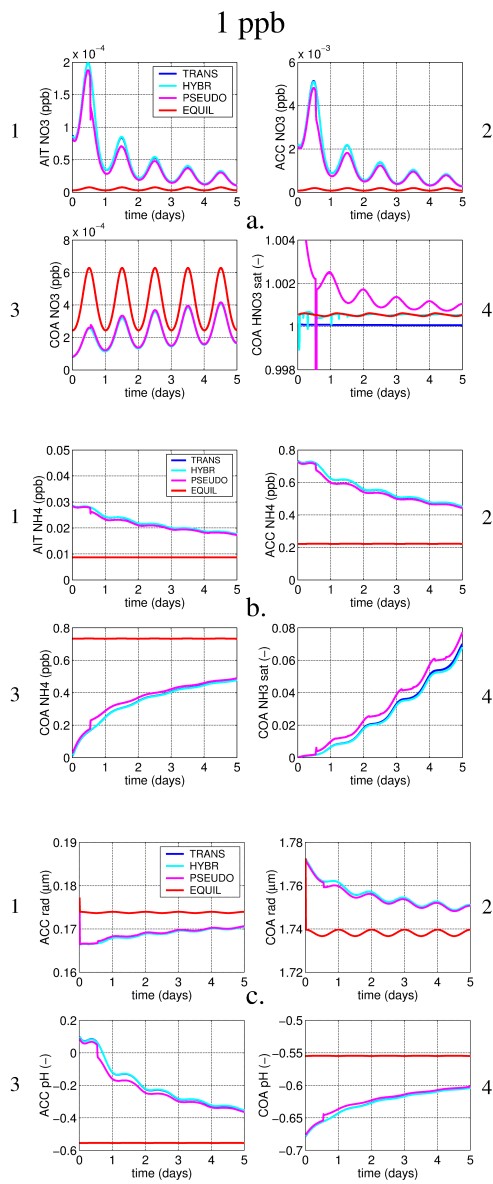

**Figure 6: Size resolved box modelling evaluation of HyDiS-1.0 with H₂SO₄ aerosol and initial mixing ratios of HNO₃(g) and NH₃(g) of 1 ppb (series 1, see text). The solver is run in the fully dynamic (=TRANS), hybrid dynamic and equilibrium (=HYBR), equilibrium with pseudo-dynamic correction (=PSEUDO) and full equilibrium configurations. Atmospheric volume mixing ratios of nitrate (a) and ammonium (b) in the aqueous phase as a function of time for the Aitken (1), accumulation (2) and coarse (3) mode, respectively. For the coarse mode, the degree of saturation (4) with respect to nitric acid and ammonia is also shown. Accumulation and coarse mode radii (c.1-2) and pH (c.3-4), respectively. A diurnal temperature cycle of T=298.15±5 K is imposed. Note that TRANS may not be distinguishable from HYBR, or even both of them from PSEUDO, due to similar values.**

10 ppb

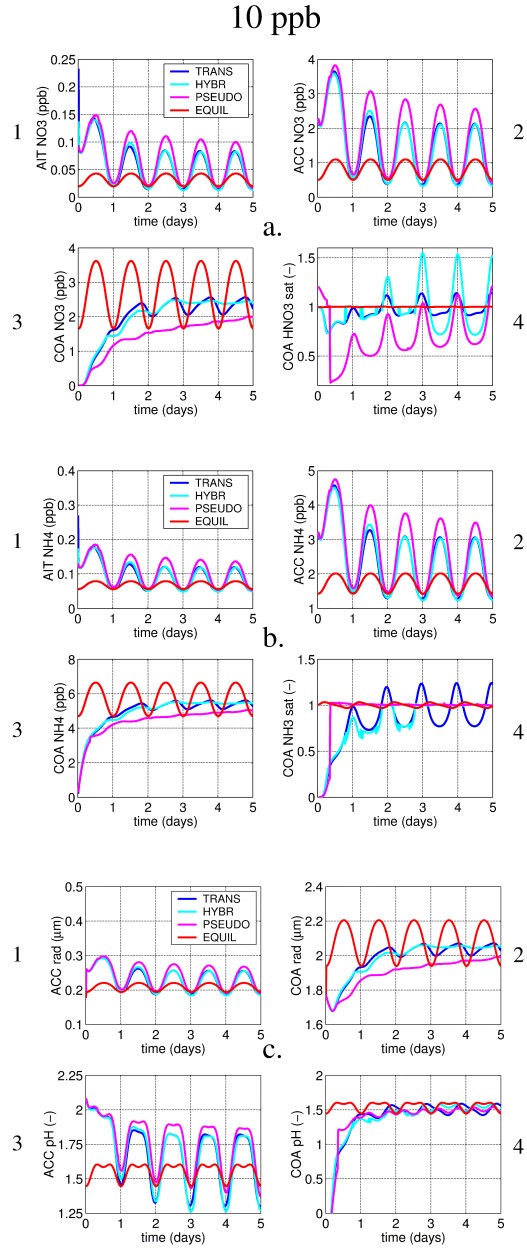

**Figure 7: Same as Fig. 4 with initial mixing ratios of HNO₃(g) and NH₃(g) of 10 ppb.**

## 100 ppb

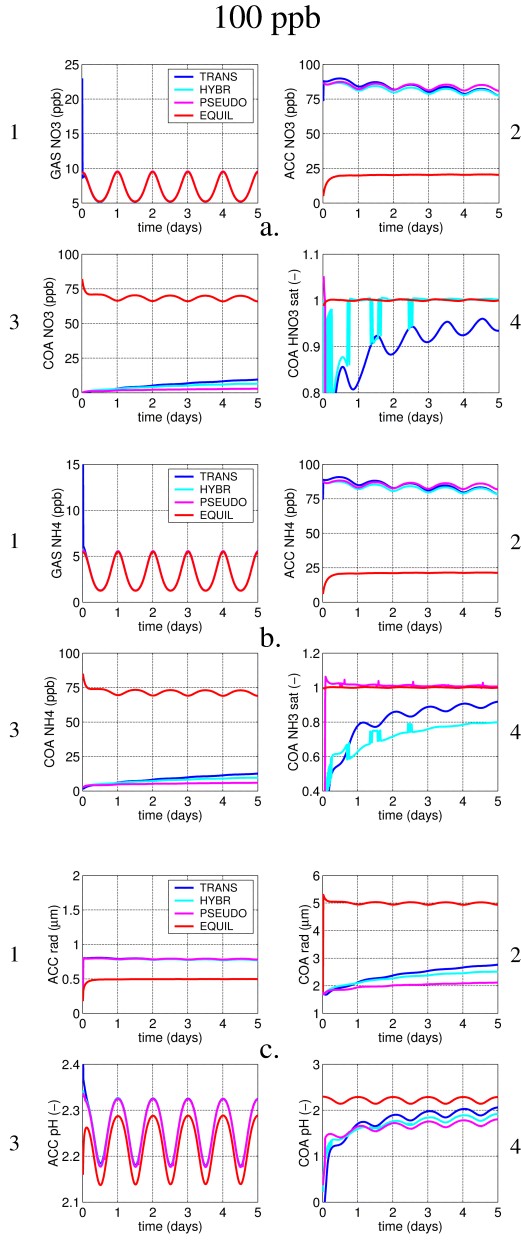

a.

b.

c.

**Figure 8: Same as Fig. 5 with initial mixing ratios of HNO₃(g) and NH₃(g) of 100 ppb, and the gas phase (a.1 and b.1) shown instead of the Aitken mode, respectively. In panel (c.3), TRANS, HYBR and PSEUDO produce largely similar values.**

1 ppb

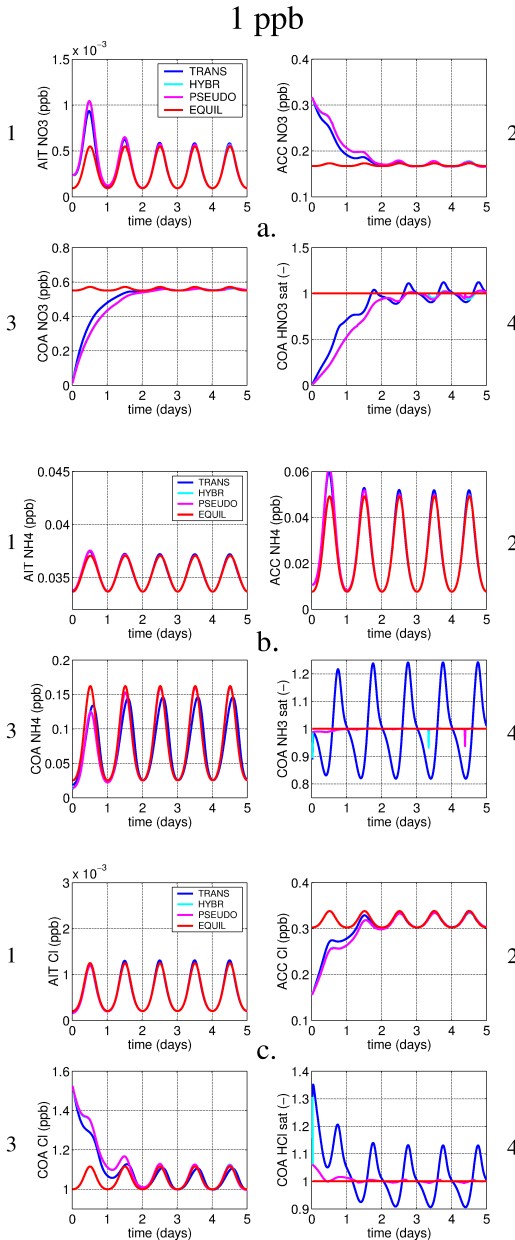

**Figure 9: Size resolved box modelling evaluation of HyDiS-1.0 with $H_2SO_4$ / sea salt aerosol and initial mixing ratios of $HNO_3(g)$, $NH_3(g)$ and $HCl(g)$ of 1 ppb(series 2, see text). The solver is run in the fully dynamic (=TRANS), hybrid dynamic and equilibrium (=HYBR), equilibrium with pseudo-dynamic correction (=PSEUDO) and full equilibrium configurations. Atmospheric volume mixing ratios of nitrate (a), ammonium (b) and chloride (c) in the aqueous phase as a function of time for the Aitken (1) mode (=$H_2SO_4$), and the accumulation (2) and the coarse (3) mode (=sea salt), respectively. For the coarse mode, the degree of saturation (4) to nitric acid, ammonia and hydrochloric acid is also shown. Note that due tosea salt, total chloride is larger than 1 ppb. A diurnal temperature cycle of T=298.15±5 K is imposed. Note that HYBR is not distinguishable from PSEUDO due to almost similar values.**

# 10 ppb

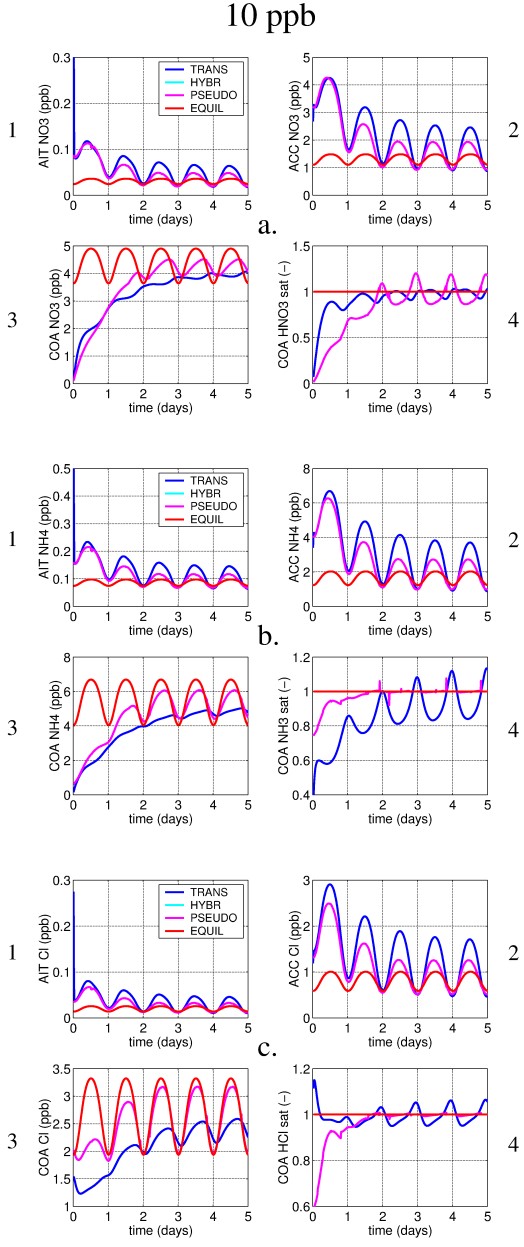

**Figure 10: Same as Fig. 9 with initial mixing ratios of HNO₃(g), NH₃(g) and HCl(g) of 10 ppb. Note that due to sea salt, total chloride is larger than 10 ppb. HYBR is indistinguishable from PSEUDO due to almost similar values.**

## 100 ppb

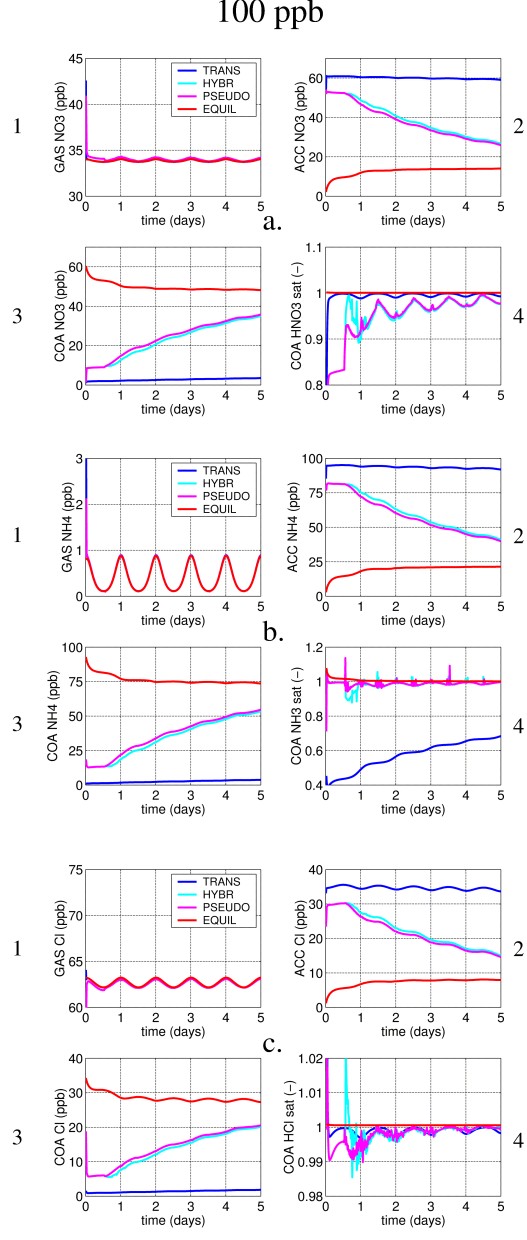

**Figure 11: Same as Fig. 9 with initial mixing ratios of HNO₃(g), NH₃(g) and HCl(g) of 100 ppb, and the gas phase (1) shown instead of the Aitken mode, respectively. Note that due to sea salt, total chloride is slightly larger than 100 ppb.**

# (a) GLOMAP-mode configuration for dissolution

Species in internally mixed modes:

**sulphate**, **sodium**, **black carbon**, **organic carbon**, **dust**
**ammonium**, **nitrate**, **chloride**

*Sea salt as mixture of sodium chloride & sodium sulphate*
*Sea salt and secondary sulphate*

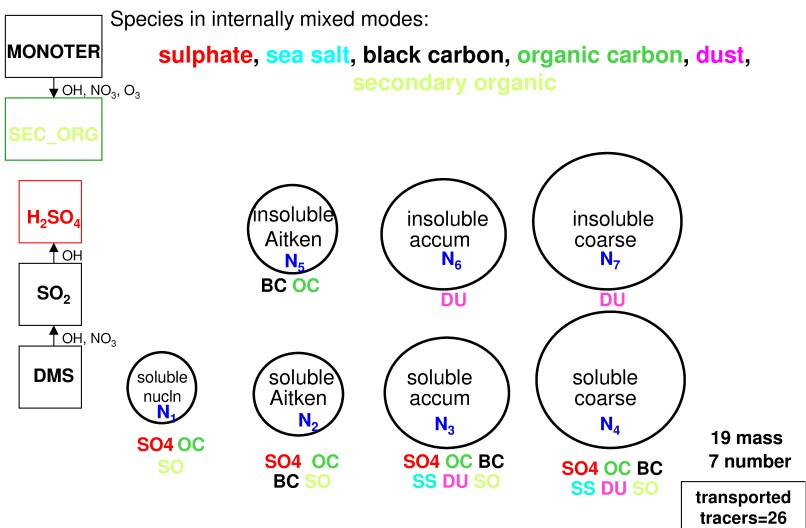

29 mass
7 number

transported
tracers=36

# (b) GLOMAP-mode standard configuration

Species in internally mixed modes:

**sulphate**, **sea salt**, **black carbon**, **organic carbon**, **dust**,
**secondary organic**

19 mass
7 number

transported
tracers=26

**Figure 12: Diagram illustrating the species contained in each size mode for the HyDiS-1.0 extended configuration of GLOMAP-mode (top) compared to that for the standard GLOMAP-mode (bottom), as described in Mann et al. (2010).**

a.

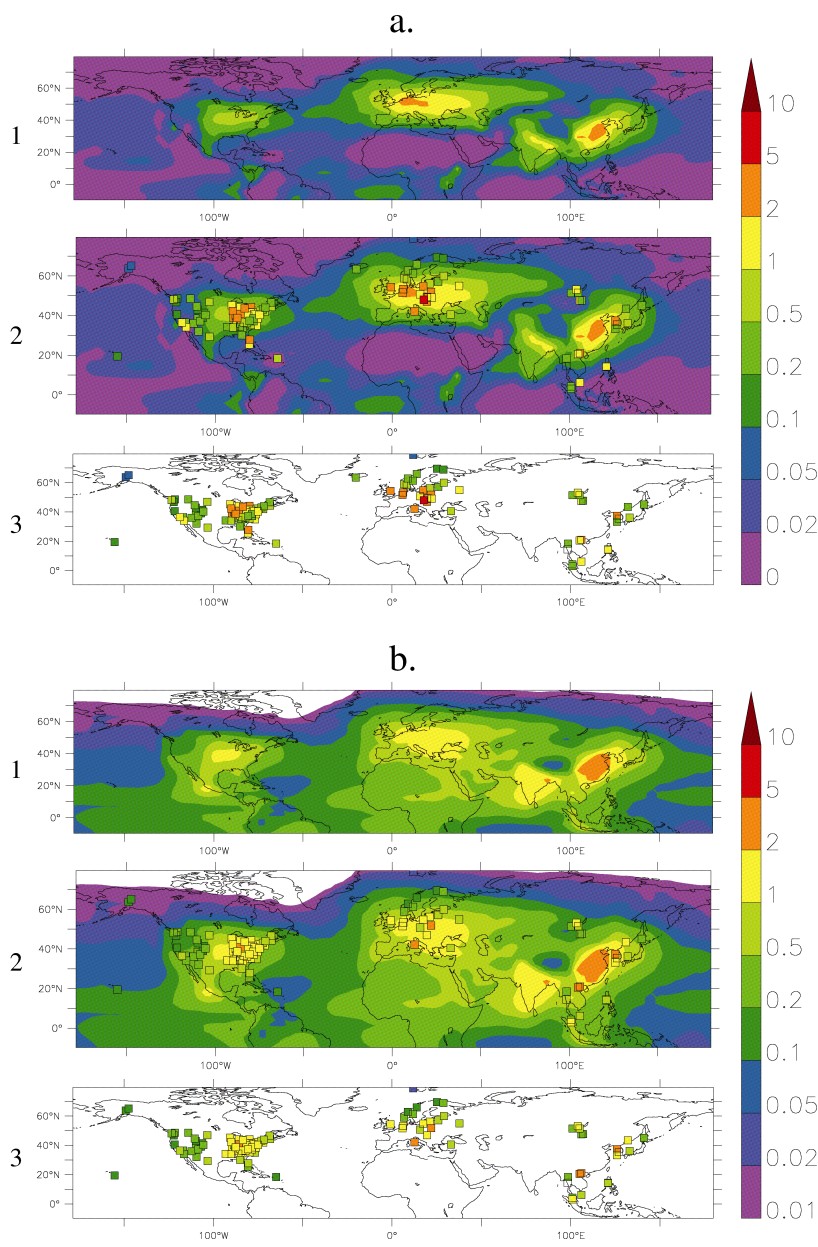

b.

**Figure 13: Comparison of 3-D CTM modelling results obtained with HyDiS-1.0 in the hybrid configuration (1) against observations (3), also compared directly on top of each other (2), for total aerosol nitrate (a) and ammonium (b) at ground level, all in μg/m³ and annually averaged for the year 2002.**

July

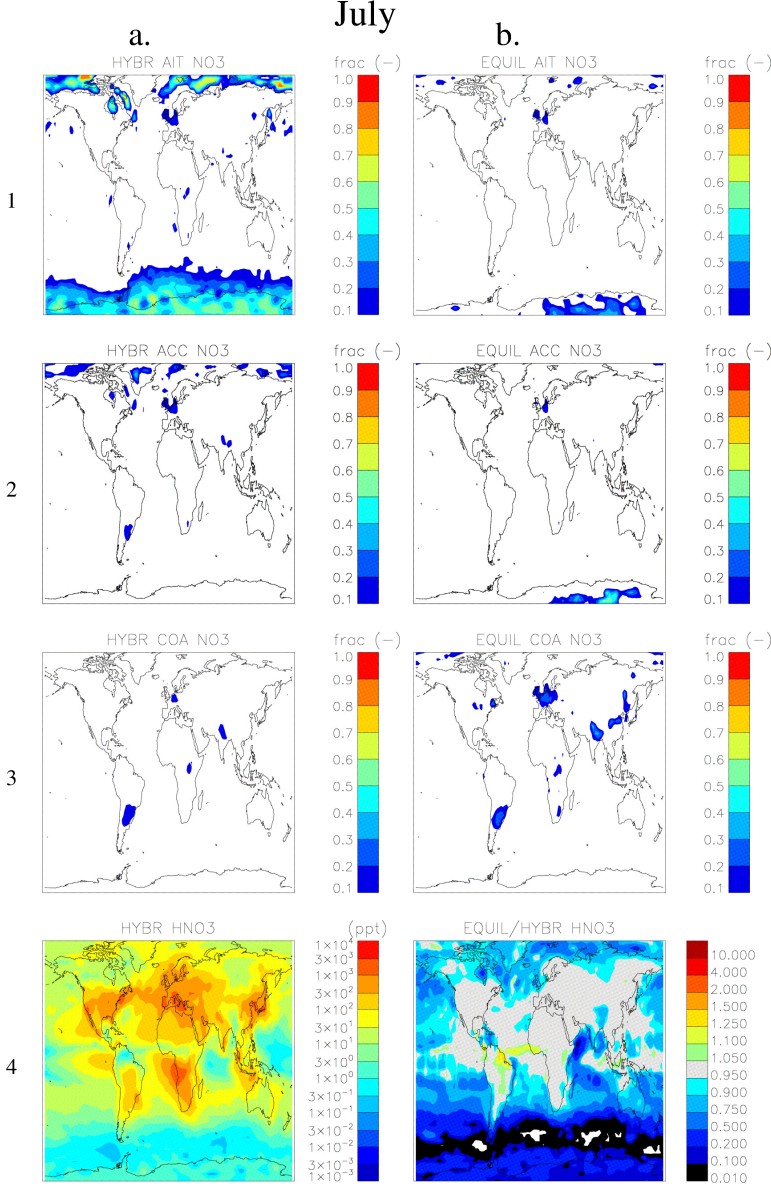

**Figure 14: Top to bottom, July 2003 ground level nitrate fraction (-) in the Aitken (1), accumulation (2) and coarse (3) mode, and nitric acid gas phase (4) mixing ratio (ppt), obtained with the hybrid (a, on the left) and the equilibrium (b, on the right) configuration of HyDiS-1.0. The equilibrium gas phase mixing ratio is shown relative to the hybrid results.**

July

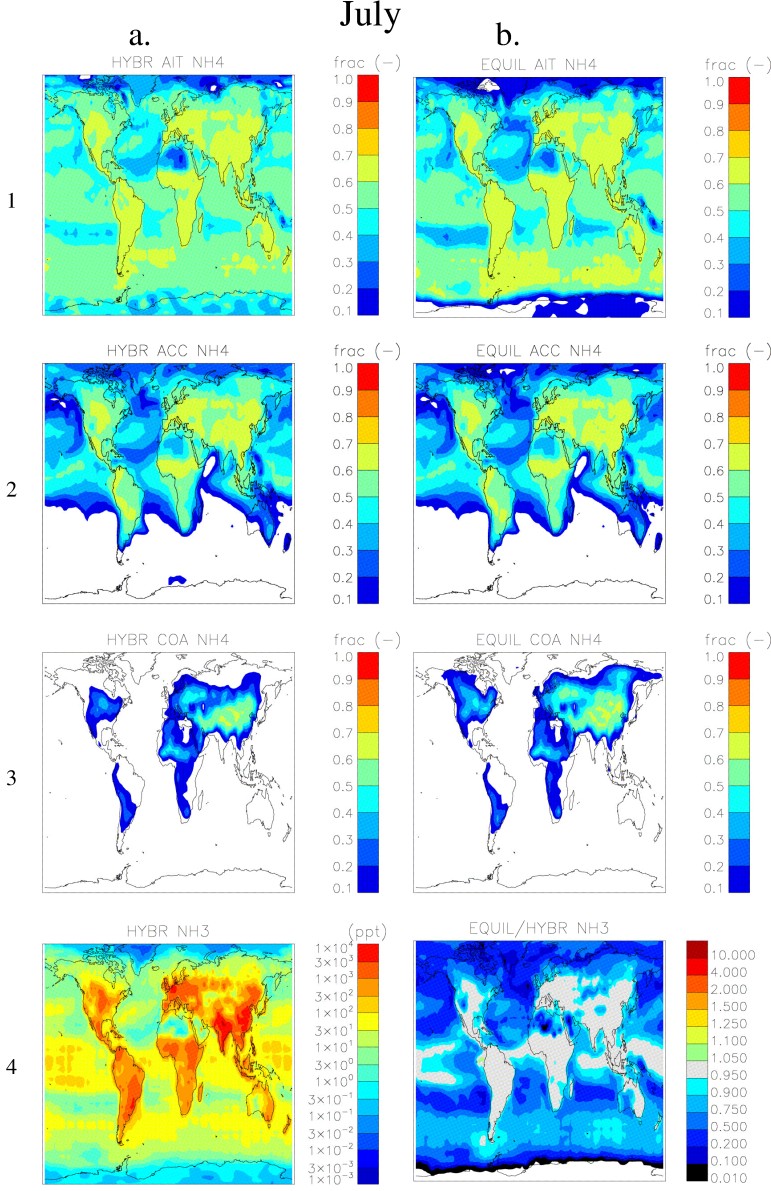

**Figure 15: Same as Figure 14 for ammonium (1-3) and ammonia (4).**

# January

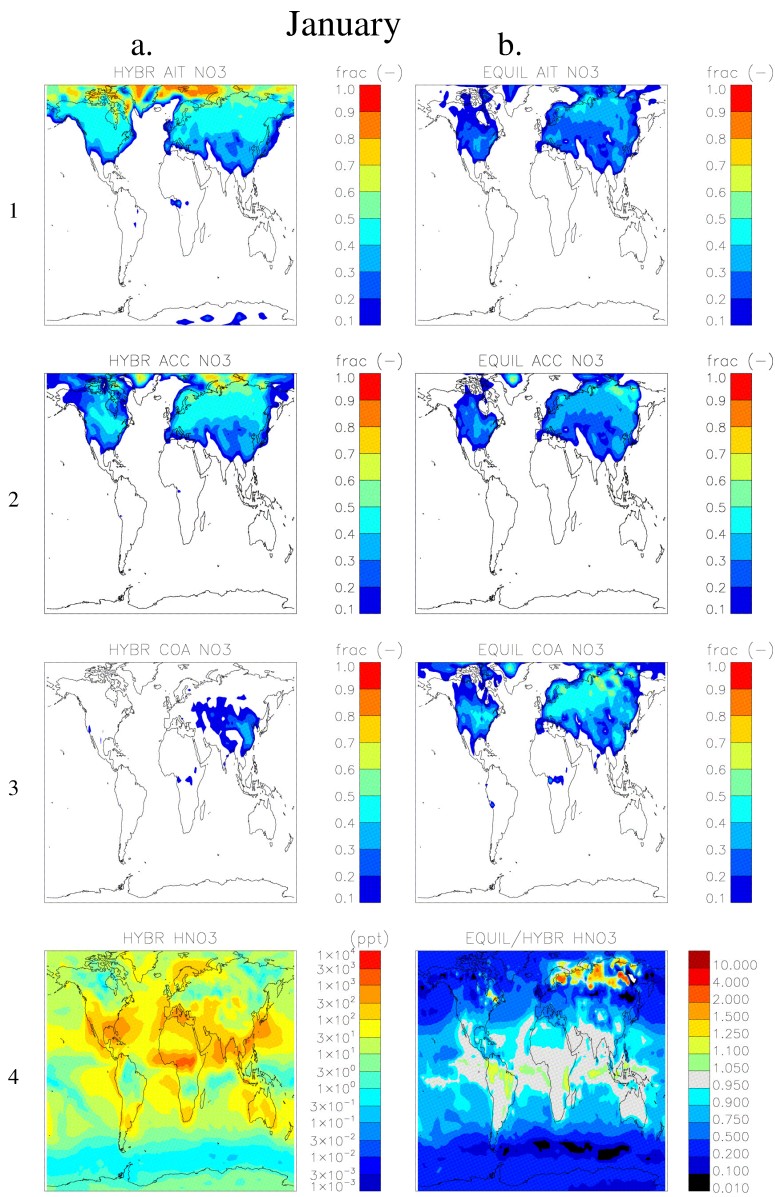

**Figure 16: Same as Figure 14 for January 2004.**

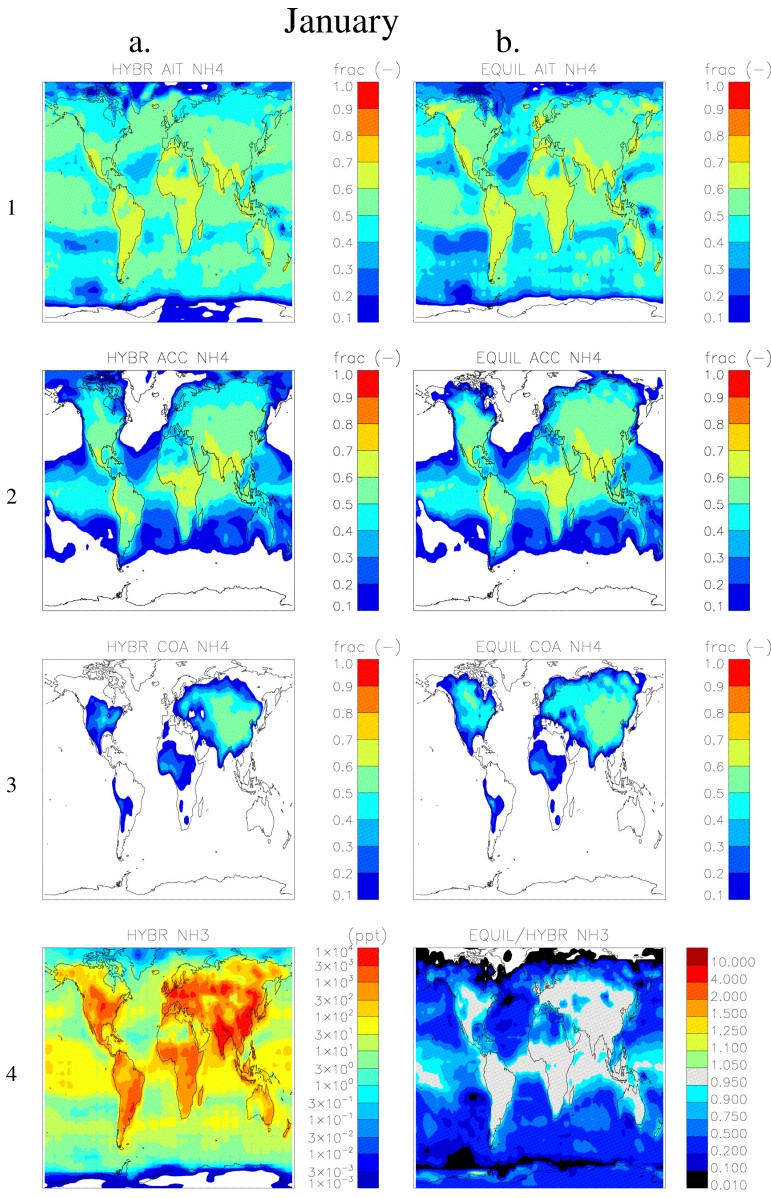

**Figure 17: Same as Figure 15 for January 2004.**

**Table 1: Initial particle composition (ppb) in the box modelling evaluation against AIM III.**

| Volume Fraction[a] | 0 | 10 | 20 | 30 | 40 | 50 | 60 | 70 | 80 | 90 | 100 |
|---|---|---|---|---|---|---|---|---|---|---|---|
| NSS-Sulphur[b] | 6.36 | 5.73 | 5.09 | 4.45 | 3.82 | 3.18 | 2.54 | 1.91 | 1.27 | 0.63 | 0.00 |
| Sodium | 0.00 | 0.33 | 0.67 | 1.00 | 1.33 | 1.66 | 2.00 | 2.33 | 2.66 | 2.99 | 3.33 |

[a]Sea salt dry volume fraction (%). The densities of pure $H_2SO_4$ and sea salt are 1769 and 2196 $kg/m^3$, respectively.

[b]Non sea salt sulphur. See text for the composition of sea salt.

**Table 2: Aerosol setup of the size-resolved box modelling evaluations Series 1 and 2.**

|  | NUC | AIT | ACC | COA |
|---|---|---|---|---|
| Particle # ($cm^{-3}$) | 1000 | 250 | 100 | 0.1 |
| Average Radius (nm) | 1 | 25 | 100 | 1000 |
| Sigma[a] | 1.59 | 1.59 | 1.59 | 2 |
| Initial Composition (ppb) | | | | |
| Series 1: | | | | |
| NSS-Sulphur[b] | 4.76E-06 | 1.86E-02 | 4.76E-01 | 1.57E+00 |
| Sodium | 0.00E+00 | 0.00E+00 | 0.00E+00 | 0.00E+00 |
| Series 2: | | | | |
| NSS-Sulphur | 4.76E-06 | 1.86E-02 | 0.00E+00 | 0.00E+00 |
| Sodium | 0.00E+00 | 0.00E+00 | 4.85E-01 | 1.60E+00 |

[a]Standard deviation of lognormal particle size distribution. The mean volume is proportional to $\exp(4.5\log^2\sigma)$.

[b]See text for sea salt composition.

**Table 3: Box-model computation time required by HyDiS-1.0, given as fraction of the computation time used by the GLOMAP aerosol microphysical scheme.**

|  | Sulphate Aerosol | | | Sea Salt Aerosol | | |
|---|---|---|---|---|---|---|
|  | Hybrid | Pseudo | Equil | Hybrid | Pseudo | Equil |
| 1 ppb | 0.31 | 0.29 | 0.16 | 0.74 | 0.40 | 0.30 |
| 10 ppb | 0.72 | 0.47 | 0.33 | 0.47 | 0.45 | 0.36 |
| 100 ppb | 0.81 | 0.47 | 0.20 | 0.59 | 0.50 | 0.35 |

**Table 4: Phase transition properties of aerosol components in HyDiS-1.0 extended GLOMAP:**

| Component | Interactive[a] | Condensable | Volatile[b] |
|---|---|---|---|
| Sulphate | Yes | Yes | No |
| Sodium[c] | Yes | No | No |
| Chloride[d] | Yes | No | No |
| Ammonium | Yes | Yes | Yes |
| Nitrate | Yes | Yes | Yes |
| Black carbon | No | No | No |
| Organic carbon | No | No | No |
| Dust | No | No | No |

[a]**Chemical interaction of aqueous phase species according to Topping et al. (2009)**

[b]**Both volatile and condensable species are assessed with the new dissolution scheme.**

10 [c]**Sea salt is assumed to be a mixture of NaCl and Na2SO4 (see text).**

[d]**Non-volatile as gas phase chemistry of HCl is not available.**

**Table 5: 3-D CTM computation time requirements of HyDiS-1.0.**

|  | % CPU TIME[a] | | | | | CPU TIME[b] (µs) |
|---|---|---|---|---|---|---|
|  | APR | JUL | OCT | JAN | AVG | AVG |
| Hybrid | 8.1 | 12.8 | 11.9 | 9.7 | 10.6 | 19 |
| Pseudo | 23.8 | 27.4 | 29.7 | 26.0 | 26.7 | 47 |
| Equil | 8.7 | 6.7 | 13.1 | 10.2 | 9.7 | 17 |

[a]**Given as percentage of the total CPU time of the 3-D CTM without HyDiS-1.0.**

[b]**Absolute CPU time per grid cell, aerosol size class and time step (see text).**