# Peer review of "Size-resolved simulations of the aerosol inorganic composition with the new hybrid dissolution solver HyDiS-1.0 – description, evaluation and first global modelling results"

_Geoscientific Model Development, 2015_

## Referee Comment (RC1) · Anonymous Referee #1 · 30 Mar 2016

This study present the development and application of a new solver that is used to describe the partitioning of inorganic compounds between gas and particle phase. The novelty of the solver is the fact that it can safes computing power by deciding on dynamical and equilibrium approaches. Its application to a global model clearly shows that the new solved can capture the dynamical, time-dependent features of nitrate partitioning that are not properly reproduced by equilibrium assumptions. The manuscript is well written although it is somewhat lengthy at some places. However, I do not have any clear suggestion on how to considerably shorten it since all parts seem essential and show the logical steps from development, over sensitivity studies in a box model

to application in a global model. I recommend the manuscript for publication after consideration of the comments below.

Main comments

- Throughout the manuscript, the particle phase is referred to as 'liquid phase'. Many recent studies have suggested that there might be multiple liquid phases (aqueous/organic) in an aerosol particle. Therefore, I suggest changing 'liquid' to 'aqueous' in the text.

- Eq. 4 and p. 8, l. 15: Usually the effective Henry's law constant $H^*$ for acids includes the acid dissociation constant $K_a$, i.e. $H^* = H (1+ K_a/[H+])$ EqI Does the fact that you assume $H_2SO_4$ as being completely dissociated lead from EqI to Eq-4? Would your approach be applicable using EqI for acids that are not completely dissociated?

- Currently, the number of interacting species is limited to three. Will it possible to rigorously extend the solver to more compounds, given the complex composition of ambient atmospheric aerosol?

- The discussion of the model results (both box and global) in Sections 4.2 and 5.3., respectively, are poorly connected to the figures. All figure panels should be labeled a, b, c, etc and the discussion should be tied more closely to the individual panel so that it is easier to reader to follow. - It is not clear what the 'entire microphysical box model' (e.g., p. 28, l. 32; and Table 1) includes. How far is a comparison of the microphysical model meaningful to the chemical model that includes the new solver? Does using the new solver in the microphysical model lead to redundancy?

- The comparison to the efficiency of previous models is only touched on briefly (end of Section 5.4). Given the extensive length of the current manuscript already, I am not asking for a detailed discussion/ However, a brief statement of the ability of the model by Zaveri et al. (2008) to reproduce the dynamical features as opposed to equilibrium assumptions should be added. In general, how well did the model by Zaveri et al., to

observations?

Minor comments

p. 2, l. 11: At most RH values, the amount of water exceeds the amount of solute mass, e.g. (Liao and Seinfeld, 2005). Do you mean here the increase of particle size due to solute mass and its associated water?

p. 6, l. 3: Do you mean ' the concentration of the dissolving species in the aerosol aqueous phase'?

p. 11, l. 17: This is unclear, in particular for the conditions described here. 1) Make clear that the aqueous s phase 'is predicted to loose protons'. 2) One would assume that under the acidic conditions described here, there is a huge excess of protons.

p. 14, l. 26: Are these parameters (activity coefficients, liquid water content, and dissociation) held constant throughout the simulation or during one time step?

p. 16, l. 21: What is the extent to which the pH varies within the iteration? Are these extreme values or are they realistic over the course of a time step, when aerosol might be exposed to ambient conditions?

p. 17, l. 9: The wording is ambiguous, I think. The convergence criterion is stricter; however, the value is DEcreased from 0.1 to 0.01. If I misunderstood this, clarify.

p. 18, Eq.29: Does 'a' have a physical meaning? What is the range of meaningful values for this parameter?

p. 19, l. 18: What are the physical/chemical parameters that determine equilibration time of a dissolving species? Is it e.g. solubility (Henry's law constant) or something else? Could threshold values be given above/below which equilibration is achieved in short time, relatively to the model time steps?

p. 20, l. 13: Does 'a' here have the same meaning as in Eq. 29? If not, choose a different symbol to avoid confusion.

p. 23, l. 23: Not clear what 'its' refers to here.

p. 23, l. 22 and 24: Spell our ADDEM, MANIC

p. 24, l. 1: What are the particle sizes for the various modes?

p. 24, l. 31: Is the temperature dependence of HNO3 dissolution very different than that of NH3? Can you support this trend by numbers (T-dependence of H?)?

p. 26, l. 3ff: Are there any simultaneous measurements of NH3 partitioning in gas and particle phase that show similar trends of subsaturation?

p. 27, l. 14/15: What is different in terms of (physico)chemical properties of chloride and nitrate vs ammonium that could explain their different behavior?

p. 27, l. 31: 'Particle concentrations' usually refers to mass or number concentration of particles. Do you mean 'the concentration of ammonia and chloride in the aqueous phase'?

p. 29, l. 20: What is included as sulphur chemistry? Both gas and aqueous phase processes? When sulphate is formed in the aqueous phase, does it contribute to the species that are equilibrated between the phases?

p. 32, l. 15: Reword: ". . . the ambient concentration of ammonia over the Southern Ocean is predicted to be lower by 10-25% than predicted by the equilibrium approach" or similar in order to clarify that you compare two model results.

p. 32, l. 28: Is this very low nitric acid concentration in the Arctic in agreement with observations?

Figure 1, caption: 1) Is the particle size the wet radius? At what RH?; 2) Could you mark the steps 1-3 in the figure (e.g. by shading or vertical lines)?

Figures 6, 7 and 9, 10 should have a somewhat more detailed caption. What does the 10 ppb etc refer to?

[Figure]

Figure 11: Some of the text in the figure is very blurry, e.g., 'HCl', 'soluble nucl.' In the top panel and 'OH, NO2' (?) in the bottom panel

Figure 12: In the caption only two panel (top and bottom) are described, but there are three panels (top, middle, bottom) shown. Clarify and improve caption.

Technical comments

p. 1, l. 15: 'developed' misspelled

p. 3, l. 14/15: acidic property –> acidity

p. 3, l. 26: 'serves as the factor that' can be omitted

p. 10, l. 25: should this read '. . . relative variation of the surface pressure'?

p. 18, l. 27: 'increments' misspelled

p. 15, l. 7: Should it be 'Equations (24)'?

p. 20, l. 8: Should that be 'Eq 32'?

p. 23, l. 12: 'hydrophilic' misspelled

p. 26, l. 29: spell out 'approximately'

p. 28, l. 15: 'concents' = 'concentrations' or 'contents'?

p. 29, l. 5: 'the number' redundant

p. 30, l. 3: 'hydrophilic' misspelled

p. 32, l. 18: 'superposition of..' – something seems missing here

Figure 8, caption: 'nucleation' misspelled

Additional reference

Liao, H., and Seinfeld, J. H.: Global impacts of gas-phase chemistry-aerosol interactions on direct radiative forcing by anthropogenic aerosols and ozone, Journal of Geophysical Research: Atmospheres, 110, D18,D18208, 10.1029/2005JD005907, 2005.

---

## Referee Comment (RC2) · Anonymous Referee #2 · 22 May 2016

The authors have developed a new hybrid solver for partitioning NH3, HNO3, and HCl to aqueous aerosol particles. The solver is supposed to be computationally efficient while maintaining accuracy. However, the box model tests and global modeling results presented do not clearly demonstrate the accuracy of the solver against a benchmark method and fail to properly evaluate its efficiency (for a chosen level of accuracy) or directly compare it with previously published solvers. The new solver is also incomplete because it presently appears to ignore gas-particle mass transfer to solid particles, which may be present under low relative humidity conditions. The manuscript is lengthy and a bit confusing in various places due to imprecise notations and terminologies. Unfortunately, it would require a major revision before it can be considered for publication in GMD.

Major Comments

1) The new hybrid solver described here treat dynamic (and equilibrium) gas-particle partitioning of HNO3, HCl, and NH3 for aqueous particles only. The implicit oversimplifying assumption is that all aerosol particles are always fully deliquesced at all relative humidities. This assumption may not hold at low relative humidities (below about 35-40%) where dissolved salts can effloresce to form a solid phase. The solver also ignores heterogeneous uptake of HNO3 on dust particles containing calcite, which is an important sink for nitrate. The proposed solver is therefore incomplete and premature for implementation in a global atmospheric transport model. These are major weaknesses that must be rectified before the present work can be considered for publication.

2) Page 2, Lines 24-25: The literature review of dynamic partitioning solvers is inadequate as there is lot more work done than what's discussed here. In addition to Capaldo et al. (2000) and Zaveri et al. (2008), the following papers describe different approaches to dynamically solving partitioning of HNO3, HCl, and NH3, which need to be properly discussed. Also, MOSAIC is incorrectly classified under the hybrid approach as it always performs fully dynamic mass transfer for all size bins.

References:

Jacobson, M. Z. (1997), Numerical techniques to solve condensational and dissolutional growth equations when growth is coupled to reversible reactions, Aerosol Sci. Technol., 27, 491– 498.

Jacobson, M. Z. (2002), Analysis of aerosol interactions with numerical techniques for solving coagulation, condensation, dissolution, and reversible chemistry among multiple size distributions, J. Geophys. Res., 107(D19), 4366, doi:10.1029/2001JD002044.

Jacobson, M. Z. (2005), A solution to the problem of nonequilibrium acid/base gas-particle transfer at long time step, Aerosol Sci. Technol., 39, 92– 103.

Wexler, A. S., and J. H. Seinfeld (1991), Second-generation inorganic aerosol model, Atmos. Environ., 25A, 2731– 2748.

Zhang, K. M., and A. S. Wexler (2006), An Asynchronous Time-Stepping (ATS) integrator for atmospheric applications: Aerosol dynamics, Atmos. Environ., 40, 4574– 4588.

3) Figure 1. There are quite a few things that I don't understand in this example/figure:

a) This example is shown before the different solvers are introduced in section 3, so it is not clear which solver was used to illustrate this example. Please clarify. Also what relative humidity was used?

b) According to the text on page 3 and the figure caption, the initial particles are pure $H_2SO_4$ with radius 50 $\mu$m and the number concentration 100 cm-3? If this is correct, then nearly all of the $NH_3$ will be absorbed into the particles in less than 10 s. But since the gas-phase $NH_3$ does not change appreciably with time, I am assuming the initial radius is 0.05 $\mu$m (not 50 $\mu$m).

c) Assuming the radius is 0.05 $\mu$m, why does it take $NH_3$ about 200 s before it begins to dissolve appreciably into $H_2SO_4$ particles? My calculation shows that $NH_4/SO_4$ molar ratio reaches $\sim$0.8 at 200 s.

d) Why does substantial amount of $HNO_3$ gas dissolve in $H_2SO_4$ particles during the first 200 s when $NH_3$ hasn't yet neutralized the acidity to some extent?

e) Why are there no oscillations for the time step of 10 s?

f) Please clearly define in the caption the short forms for terms used in the legend: "part" and "atm".

g) Are the units on the Y-axis [molecule m-3(air)] for both gas- and particle-phase

species concentrations?

h) Why do the gas- and particle-phase concentrations of become equal (blue line and green line) after 400 s for both NH3 and HNO3?

4) What is the difference between "size bin" and "size increment"? If they mean the same thing, then please stick with "size bin." Otherwise, please clearly define a "size increment."

5) Page 5, Line 16: What are the "dissolving" and "non-dissolving (=passive)" species considered in this work? It's not clear what a "non-dissolving" species even means. This terminology becomes especially confusing on page 8, lines 10-15, where a monoacid is treated as "dissolving" but the anions and cations are treated as "non-dissolving" in Eq. 11. This equation is supposed to give H+ ion concentration from the difference between anions and cations, so I don't understand why the additional term "c_i" is even included here.

6) The derivation of Eq. 9 is also very confusing. First, please clearly define the terms a, b, and c of the generic differential equation $dx/dt = ax^2 - bx + c$. Then show the solution using the same notations.

7) Will the numerator in Eq. 15 always be positive? What happens if a species in a given bin has a tendency to evaporate during a time step (e.g., HNO3 displacing HCl from sea salt)?

8) How does the dynamic solver handle simultaneous mass transfer of H2SO4, HNO3, HCl, and NH3 to a size distributed aerosol? It seems mass transfer of H2SO4 is not included in the derivation of the dynamic solver equations. Also, none of the box model test cases include gas phase H2SO4 that condenses along with HNO3, HCl, and NH3.

9) While the present box model tests are useful in showing the benefits of the hybrid solver over the equilibrium approach, they do not demonstrate the accuracy of the new dynamic solver introduced here. To evaluate accuracy, it is necessary to compare the

dynamic solver for monodisperse aerosol (similar to the example shown in Figure 1) and 4 size bins (as already shown in series 1 and 2) against a benchmark dynamic solver (e.g., LSODE) with strict error tolerances for a range of initial gas concentrations (of H2SO4, HNO3, HCl, and NH3) and aerosol sizes, concentrations, and compositions. The predicted equilibrium (after sufficient time) gas and aerosol concentrations (including aerosol pH) for the monodisperse test cases must also be compared to a benchmark thermodynamics model such as AIM (available online) to evaluate the accuracy of the thermodynamics treatment in the present solver. For example, see the dynamic solver evaluation done in Zaveri et al. (2008). Such an evaluation against a benchmark solver is especially warranted in the light of the several discrepancies found in the results of the dynamic solver shown in Figure 1 (see #3).

10) The main goal of the present work is to introduce a new hybrid solver that is computationally efficient. But since any solver can be made efficient by compromising its accuracy, it becomes necessary to evaluate computational efficiency as a function of accuracy. The CPU costs presented for the various dynamic solvers in Table 1 are of little use without stating their accuracies against a benchmark solver (see #9).

11) As the authors have already acknowledged, the comparison of the computational cost of the present solver with that of MOSAIC is fraught with many issues—different model configurations and chemical/physical complexities, computer hardware and computing architectures, etc. Also, the 125 $\mu$s CPU cost for MOSAIC quoted here includes microphysical calculations (in addition to gas-particle mass transfer and thermodynamics) whereas the 20 $\mu$s given for the present hybrid solver appears to be for gas-particle mass transfer and thermodynamics calculations only. Furthermore, the CPU costs given in Table 5 for cases 4-7 (aqueous particles only) range from $\sim$20-40 $\mu$s (depending on the case, hardware, and compiler), which is more directly comparable to the CPU cost presented here for the hybrid solver. Having said that, the accuracy and efficiency of both solvers should be evaluated for the same set of problems on the same machine for the comparison to be meaningful.

12) Table 2: In GLOMAP sea-salt chloride is treated as non-volatile. Then how do H2SO4 and HNO3 condense on sea salt aerosol without evaporating HCl? Non-volatile treatment of HCl becomes problematic especially for condensation of HNO3, because both HNO3 and HCl are semivolatile and the extent of HNO3 partitioning crucially depends on HCl.

13) How is HNO3 uptake on calcite containing dust particles treated in GLOMAP?

Minor comments 1. Page 2, Line 10: Please change "dissolving acid and a dissolving base, such as NH3 and HNO3" to "dissolving acid and a dissolving base, such as HNO3 and NH3".

2. Figure 11. The resolution of this figure needs to be improved.

---

## Author Comment (AC1) · 18 Jun 2016

Response to reviewer #1 comments

We thank the anonymous reviewer #1 for their review and note that they have identified that "the new solver can capture the dynamical, time-dependent features of nitrate partitioning that are not properly by equilibrium assumption".

We take their point about the paper being "somewhat lengthy". However, when writing the paper we considered carefully whether or not we could shorten the paper, and concluded that the level of detail provided is important and will help to ensure clarity and rigour.

In our response here we address each of the points made by the reviewer in the order that is given in his/her review:

Major Comments

1) Throughout the manuscript, the particle phase is referred to as 'liquid phase'. Many recent studies have suggested that there might be multiple liquid phases (aqueous/organic) in an aerosol particle. Therefore, I suggest changing 'liquid' to 'aqueous' in the text.
We agree entirely and have changed the terminology.

2) Eq. 4 and p. 8, l. 15: Usually the effective Henry's law constant H* for acids includes the acid dissociation constant Ka, i.e. H* = H (1+ Ka/[H+]) EqI Does the fact that you assume H2SO4 as being completely dissociated lead from EqI to Eq-4? Would your approach be applicable using EqI for acids that are not completely dissociated?
The total dissociation of acids assumption and the negligence of OH in the ion balance introduce a simplification into the formalism that allows us to focus on other important phenomena, such as the numerical stiffness property of the system and the related computational efficiency. The partial dissociation of acids and the presence of OH each add one additional degree of freedom to the system, that increases the degree/order of any relevant resolving/differential equation accordingly. For instance, within the dynamical solver, neglecting OH reduces Eq. 8 to a second order differential equation rather than third order, and also matches the similar approach taken in the thermodynamic scheme PD-FiTE. Most importantly, within the chemical equilibrium solver, and in relationship with its analytic approach, the simplification allowed us to take the chemical interaction of the dissolving species into account. If the partial dissociation of the dissolving species and H2SO4, and OH were taken into account, the resolving equation for one dissolving base or acid would be of the $5^{th}$ degree. If only the partial dissociation of H2SO4 is taken into account, which is done in the circumstance of one dissolving species, the resolving equation is still of the $3^{rd}$ degree. As one of the underlying reason for the numerical stiffness is chemical interaction, it is necessary to restrict the degree of the resolving equation to a value that is manageable. We have added thefollowing paragraph underneath equation 4 to explain these circumstances more appropriately.

*"In the preceding expression, the partial dissociation property of the dissolving species is neglected, which is an underlying assumption for HyDiS-1.0. It will allow us to reduce the number of degrees of freedom of the considered chemical system by one unit for each dissolving species. In doing so other properties, such as the species' chemcial interaction, may be taken into account more throroughly, as analytical solutions may be derived along with a critical reduction of the degree of the respective resolving equations. We have seen in the preceding section that the chemical interaction between acids and bases plays an essential role to the numerical stiffness property of the system."*

3) Currently, the number of interacting species is limited to three. Will it possible to rigorously extend the solver to more compounds, given the complex composition of ambient atmospheric aerosol?
In line with the preceding point, the number of dissolving species may not be consistently augmented beyond this number, unless the chemical interaction of the dissolving species will necessarily be accounted for less thoroughly within the chemical equilibrium solver. We have added the following sentence for clarity.

*"Similarly, the number of dissolving species whose chemical interaction may be fully taken into account may not exceed three, as no analytical solution is readily available to an equation beyond the fourth degree."*

4) The discussion of the model results (both box and global) in Sections 4.2 and 5.3., respectively, are poorly connected to the figures. All figure panels should be labeled a, b, c, etc and the discussion should be tied more closely to the individual panel so that it is easier to reader to follow.
In accordance with the suggestion made by the reviewer we have numbered all panels of Figures 5-16 and linked them more thoroughly with the text.

5) It is not clear what the 'entire microphysical box model' (e.g., p. 28, l. 32; and Table 1) includes. How far is a comparison of the microphysical model meaningful to the chemical model that includes the new solver? Does using the new solver in the microphysical model lead to redundancy?
We have reshaped the explanation of how the percentage computational expense of HyDiS within the box model runs is obtained. Also, we have mentioned the processes that are comprised within the Glomap microphysical scheme and given more details on the implementation of HyDiS within this scheme.

6) The comparison to the efficiency of previous models is only touched on briefly (end of Section 5.4). Given the extensive length of the current manuscript already, I am not asking for a detailed discussion/ However, a brief statement of the ability of the model by Zaveri et al. (2008) to reproduce the dynamical features as opposed to equilibrium assumptions should be added. In general, how well did the model by Zaveri et al., to observations?
The reviewer asks for an extension of the comparison of HyDiS to other gas/particle exchange schemes with respect to the global aerosol composition. In principle, we agree with the reviewer about the relevance of such a comparison. However, a comparison to other schemes is not trivial and will require a lengthy analysis. At this point a comparison would be premature, as certain important processes, such

as particle cristallization and the dissolution of HCl, are not currently included in TOMCAT-GLOMAP, and certain other processes, such as heterogeneous chemistry and aerosol organic species, are not taken into account for the present preliminary simulations. For these reasons, we limited the analysis herein to solver computational efficiency and numerical reliability considerations, along with a preliminary assessment of the relevance of nonequilibrium dynamics as the one intended main plus of the solver relative to equilibrium approaches.

Minor Comments

p. 2 l. 11: At most RH values, the amount of water exceeds the amount of solute mass, e.g. (Liao and Seinfeld, 2005). Do you mean here the increase of particle size due to solute mass and its associated water? We mean both the increase of particle mass via efficient interaction of nitrate+ammonium and the amount of water that goes with it. The interaction of ammonium and nitrate may be efficient to the point that the loss of the particle water mass via the reduction of hygroscopicity along with the acidity of the aqueous phase does not exceed the gain of particle mass via the chemical interaction of ammonium, nitrate and water. Clarified.

p. 6 l. 3: Do you mean ' the concentration of the dissolving species in the aerosol aqueous phase'? Expression corrected for improved clarity.

p. 11 l. 17: This is unclear, in particular for the conditions described here. 1) Make clear that the aqueous s phase 'is predicted to loose protons'. 2) One would assume that under the acidic conditions described here, there is a huge excess of protons. Sentence is clarified.

p. 14 l. 26: Are these parameters (activity coefficients, liquid water content, and dissociation) held constant throughout the simulation or during one time step? The mentioned variables are not considered as such by the analytical scheme. They are updated at the outer iteration level of the chemical equilibrium solver, this is clarified further down in the text, however cannot be mentioned here as the reader would not be able to follow at this point.

p. 16 l. 21: What is the extent to which the pH varies within the iteration? Are these extreme values or are they realistic over the course of a time step, when aerosol might be exposed to ambient conditions? Extreme variations of the particle pH occur frequently in the course of the execution of the equilibrium solver, and they are not necessarily related to exceptionally rapid modifications of ambient conditions. The extreme variability of particle pH is the most characteristic feature of the numerical stiffness property of dissolution, and occurs under all polluted conditions for c> 1ppb approximately, with the diurnal cycle of temperature serving as one of the potential sources of sufficient perturbation (used in the box model experiments) in the context of an overall time step of 15 or 30 minutes. In this context, the convergence of the pH variability to less than 0.1 is an appropriate prerequisite for one outer iteration that updates the parameters, and a subsequent round of internal iterations with a renewed massive variability of the pH. We refer to Section 2 for further detail on the numerical stiffness properties of chemically interacting species via pH.

p. 17 l. 9: The wording is ambiguous, I think. The convergence criterion is stricter; however, the value is DEcreased from 0.1 to 0.01. If I misunderstood this, clarify. The convergence criterion is increased from 0.001 to 0.01. The criterion for this choice is set to 0.1. Clarified.

p. 18 Eq. 29: Does 'a' have a physical meaning? What is the range of meaningful values for this parameter? The proportionality constant has a qualitative physical meaning, as it stands for the amount of chemical interaction that one is willing to take into account in terms of the determination of the length of the internal time step of the dynamic solver. Although it has this qualitative physical meaning, it is hard to figure out how this would physically translate into a mathematical expression. Notwithstanding, this latter issue is of limited relevance, as the proportionality constant has another more relevant meaning, as it conditions the number of internal time steps of the dynamic solver. If a size class would require more internal time steps due to numerical constraints, it would be assumed to be in equilibrium whether this is accurate or not. It is clear that the computational expense will be roughly proportional to this number. We have chosen 'a' for the number of internal time steps to be limited to roughly 3. This number was shown by the box model experiments to produce reasonably accurate results, even under numerically stiff conditions of strong chemical interaction. It should be noticed at this point that a larger overall time step might require an accordingly smaller proportionality constant, unless the accuracy of the results might be critically diminished. The text was modified for more clarity.

p. 19 l. 18: What are the physical/chemical parameters that determine equilibration time of a dissolving species? Is it e.g. solubility (Henry's law constant) or something else? Could threshold values be given above/below which equilibration is achieved in short time, relatively to the model time steps? By virtue of Eq. 15 the equilibration time is given by the ratio of the difference between the equilibrium and the momentary atmospheric concentration of the dissolving species in the aqueous phase, and the potential of the dissolved species to evaporate from particle surface or the potential of the dissolving species to condense onto the particle surface, whichever potential is the largest. This quantity may be seen as a normalized degree of saturation of the aqueous phase, which correlates with the equilibration time. By virtue of the proportionality factor 'a' and the definition of the distinction criterion between equilibrium and dynamic simulation, the critical equilibration time is defined relative to the overall time step of the model: a particle is considered to be in equilibrium if its equilibration takes less than half the overall time step of the model. Due to its multivariable dependence no individual threshold value may be given that would determine a short equilibration time. We refer to the explanations around Eq. 15 for explanations within the manuscript on this question.

p. 20 l. 13: Does 'a' here have the same meaning as in Eq. 29? If not, choose a different symbol to avoid confusion. The notation of the second coefficient is changed to 'b'.

p. 23 l. 23: Not clear what 'its' refers to here. Corrected.

p. 23 l. 22 and 24: Spell our ADDEM, MANIC. Now spelled.

p. 24 l. 1: What are the particle sizes for the various modes? Mode sizes now indicated.

p. 24 l. 31: Is the temperature dependence of HNO3 dissolution very different than that of NH3? Can you support this trend by numbers (T-dependence of H?)? The temperature dependence of the solubility of HNO3 and NH3 is explained more thoroughly and references are added.

p. 26 l. 3ff.: Are there any simultaneous measurements of NH3 partitioning in gas and particle phase that show similar trends of subsaturation? The observed undersaturation concerns a two-fold phenomenon, each related to a particular kind of numerical stiffness. On the one hand, the aqueous phase remains undersaturated as it tends to equilibrate very slowly in the context of the slow transit of the semi-volatiles through the gas phase. This dynamical phenomenon is now mentioned in the Introduction (Zaveri et al., 2008). On the other hand, the surface pressure of one dissolving species remains underneath the ambient pressure, depending on the choice that is made by the solver as a function of its configuration. This artificial phenomenon is related to the extreme numerical stiffness via particle pH and chemical interaction that is encountered at 100 ppb, which appears to cause a relative imprecision of the solver. This artifact has very little influence on the predicted concentrations of the dissolved species, because these are several orders of magnitude larger than the proton concentration. We point to the explanations that are given within Section 4.2 around Figure 10.

p. 27 l. 14/15: What is different in terms of (physico)chemical properties of chloride and nitrate vs ammonium that could explain their different behavior? The reason for the similarity of ammonium under dynamic and equilibrium conditions may be illustrated with the coarse mode. The particle pH is too low for the solubility of ammonia to be high, such that the equilibration time of ammonium remains relatively low. Furthermore, nitric acid and hydrochloric acid are competitors, as nitric acid drives out the latter and replaces it, and particle pH remains unaffected, which in turn tends to keep the solubility of ammonia low but constant. The only factor that affects the solubility of ammonia is its temperature dependance, which also results in a quick almost instantaneous adaptation, to the overall effect that the equilibrium and dynamic concentrations of ammonium are almost equal. We have added 2 sentences for explanation.

p. 27 l. 31: 'Particle concentrations' usually refers to mass or number concentration of particles. Do you mean 'the concentration of ammonia and chloride in the aqueous phase'? Corrected.

p. 29 l. 20: What is included as sulphur chemistry? Both gas and aqueous phase processes? When sulphate is formed in the aqueous phase, does it contribute to the species that are equilibrated between the phases? An explanation is added to the text.

p. 32 l. (1)5: Reword: ": : : the ambient concentration of ammonia over the Southern Ocean is predicted to be lower by 10-25% than predicted by the equilibrium approach" or similar in order to clarify that you compare two model results. Inserted.

p. 32 l. 28: Is this very low nitric acid concentration in the Arctic in agreement with observations? The question of the validation of the model predictions against observations (and other models) certainly is an important one. Adams et al. (1999) give a few data points for their simulated mixing ratio of ammonia close to the western coast of Alaska during the Arctic summer. Their values of 2-5 ppt are similar to our estimations under the equilibrium assumption. With the hybrid configuration we obtain approx. 10 ppt. In their measurements off the east coast of Baffin Island during Summer 2014, Wentworth et al. (2016) find ammonia mixing ratios of around 30 ppt. With the hybrid configuration we obtain a simulated mixing ratio of approx. 10 ppt, with the equilbrium configuration we obtain approx. 1/3 of this value. It thus seems that non-equilibrium dynamics play an important role and that our simulations get the order of magnitude right in the Arctic. Having said this, the observed and the simulated variability of ammonia is large both in time and space, such that these matches are inconclusive. As detailed within our answer to the 6$^{th}$ major comment, the validation of the model results will be performed in much greater detail in follow-on publications, as the present largely focuses on solver presentation and numerical validation. We have modified the text to clarify the circumstance that we compare two model results.

Fig. 1, caption: 1) Is the particle size the wet radius? At what RH?; 2) Could you mark the steps 1-3 in the figure (e.g. by shading or vertical lines)? 1) the dry radius is given, this circumstance is now indicated along with the relative humidity of 80%, 2) the regimes are now marked as requested.

Fig. 6, 7, 9 and 10 should have a somewhat more detailed caption. What does the 10 ppb etc refer to? The figure caption is reformulated and more details are added to Figs. 5 and 8, respectively.

Fig. 11: Some of the text in the figure is very blurry, e.g., 'HCl', 'soluble nucl.' In the top panel and 'OH, NO2' (?) in the bottom panel. The resolution of the figure is improved.

Fig. 12: In the caption only two panel (top and bottom) are described, but there are three panels (top, middle, bottom) shown. Clarify and improve caption. The figure caption has been partially reworded.

Technical comments: thank you for indicating these, all worked in.

Additional Reference:

Wentworth, G. R., Murphy, J. G., and Croft, B., et al.: Ammonia in the summertime Arctic marine boundary layer: sources, sinks, and implications, Atmos. Chem. Phys., 16, 1937–1953, 2016.

---

## Author Comment (AC2) · 18 Jun 2016

Response to reviewer #2 comments

We thank the anonymous reviewer #2 for their review.

We feel strongly the reviewer overstates the severity of the issues he identifies. It seems to us that many if not all of the reviewer's major comments are related to weaknesses in the clarity of the manuscript. We have revised the manuscript to address this, giving additional explanation where appropriate.

There are 3 elements the reviewer identifies as together meaning major revisions would be required before publication in GMD:

(1) "(…) (T)he box model tests and global modeling results presented do not clearly demonstrate the accuracy of the solver against a benchmark method and fail to properly evaluate its efficiency (for a chosen level of accuracy) or directly compare it with previously published solvers."

We assert strongly that our use of the fully dynamic implementation of the Jacobson (1997) scheme as a benchmark to test the hybrid dissolution solver is entirely justified. The potential of the embedded fully dynamic scheme to serve as a benchmark is documented in the literature (Zaveri et al., 2008). Similarly, this manuscript builds on the previous testing of the thermodynamic scheme against AIM (Topping et al., 2009), and there should be no need to redo this here.

(2) "The new solver is also incomplete because it presently appears to ignore gas-particle mass transfer to solid particles, which may be present under low relative humidity conditions."

Our paper describes a hybrid solver to calculate the dissolution of inorganic gases into the aerosol aqueous phase. We agree that we should perhaps have stated more clearly that aerosol solid phase processes are not included. However, the purpose of the paper is to describe and evaluate the new solver in the box model, and assess its numerical reliability and computational efficiency within the framework of a global model. We content strongly that the paper is consistent and adequate as being directed towards these aspects.

(3) "The manuscript is lengthy and a bit confusing in various places due to imprecise notations and terminologies."

We acknowledge that the paper is long, but when writing it, we considered carefully to what extent it should be shortened. We concluded that the level of detail provided is important and will help to ensure reproducibility of results and to rigorously describe the solver. We note that reviewer 1 states that *"all parts seem essential and show the logical steps from development, over sensitivity studies in a box model to application in a global model."*

We also are puzzled that the reviewer regards the use of 'size bin' and 'size increment', 'dissolving' and 'non-dissolving', the way Eq. 9 is introduced and an apparent typing error in Eq. 15 as major issues,

while the large amount of effort that was put into the explanation of the solver appears not to find any credit.

Finally, some of the reviewer's concerns, like the confusion around Figure 1, could have been easily addressed via a question to the authors.

Within the following we will address the points of the reviewer in the order given within his/her comment.

Major comments:

1) "The new hybrid solver described here treat dynamic (and equilibrium) gas-particle partitioning of HNO3, HCl, and NH3 for aqueous particles only. The implicit oversimplifying assumption is that all aerosol particles are always fully deliquesced at all relative humidities. This assumption may not hold at low relative humidities (below about 35-40%) where dissolved salts can effloresce to form a solid phase. The solver also ignores heterogeneous uptake of HNO3 on dust particles containing calcite, which is an important sink for nitrate. The proposed solver is therefore incomplete and premature for implementation in a global atmospheric transport model. These are major weaknesses that must be rectified before the present work can be considered for publication."

The reviewer observes that the heterogeneous and solid phase processes are not treated by the solver, and maintains that its implementation within a global model is premature. We agree that solid phase processes are not included and that these play an important role for the aerosol inorganc composition. The revised manuscript now makes it much clearer that these processes are not included and gives reference for papers which include them and assess their effects.

Also, we would like to stress that the paper is clearly set out to describe and evaluate a new solver for the reversible dissolution of inorganic species into the aerosol liquid phase. As stated within Section 5.1, the solver is implemented into a global model to verify that it is able to give realistic results for the full parameter space encountered in a global 3D chemistry-aerosol transport model, to demonstrate the importance of the non-equilibrium property of dissolution, and to evaluate its computational efficiency.

All of these do not require a complete mechanism of aerosol/gas exchange. Given the explanations of Section 5.1, we had assumed that the discussion of the global modelling results within Section 5.3 would be understood from this perspective.

We have made additions to the conclusion and Section 5 that emphasize the limitations of the global simulations in the context of their scope.

2) "Page 2, Lines 24-25: The literature review of dynamic partitioning solvers is inadequate as there is lot more work done than what's discussed here. In addition to Capaldo et al. (2000) and Zaveri et al. (2008), the following papers describe different approaches to dynamically solving partitioning of

HNO3, HCl, and NH3, which need to be properly discussed. Also, MOSAIC is incorrectly classified under the hybrid approach as it always performs fully dynamic mass transfer for all size bins."

The reviewer has drawn our attention to the fact that the discussion of the literature requires improvement. We have expanded the discussion on the basis of the additional references that were provided, as Jacobson (1997), (2002) and (2005) are discussed within Section 4.1 in the context of the motivation for the choice of the benchmark scheme, and Zhang and Wexler (2006) and Wexler and Seinfeld (1992) in the Introduction. We preferred to discuss Wexler and Seinfeld (1992) rather than (1991), because the earlier appears to be the more relevant one.

3) "Figure 1. There are quite a few things that I don't understand in this example/figure:
a) This example is shown before the different solvers are introduced in section 3, so it is not clear which solver was used to illustrate this example. Please clarify. Also what relative humidity was used?
b) According to the text on page 3 and the figure caption, the initial particles are pure H2SO4 with radius 50 μm and the number concentration 100 cm-3? If this is correct, then nearly all of the NH3 will be absorbed into the particles in less than 10 s. But since the gas-phase NH3 does not change appreciably with time, I am assuming the initial radius is 0.05 μm (not 50 μm).
c) Assuming the radius is 0.05 μm, why does it take NH3 about 200 s before it begins to dissolve appreciably into H2SO4 particles? My calculation shows that NH4/SO4 molar ratio reaches approx. 0.8 at 200 s.
d) Why does substantial amount of HNO3 gas dissolve in H2SO4 particles during the first 200 s when NH3 hasn't yet neutralized the acidity to some extent?
e) Why are there no oscillations for the time step of 10 s?
f) Please clearly define in the caption the short forms for terms used in the legend: "part" and "atm".
g) Are the units on the Y-axis [molecule m-3(air)] for both gas- and particle-phase species concentrations?
h) Why do the gas- and particle-phase concentrations of become equal (blue line and green line) after 400 s for both NH3 and HNO3?"

The reviewer points to Figure 1 and asks for clarification. First, the reviewer diagnosed correctly that the initial size of the particles is not 50 micrometres but rather 50 nanometres. Second, a great deal of the confusion might have arisen due to what seems to be a lack of sufficient clarity on what is exactly shown within Figure 1. The graph does not compare the gas phase molecule number concentration against the gas phase equivalent concentration of molecules in the liqud phase but rather against the partial pressure at particle surface, which is expressed as a molecule number concentration equivalent via the perfect gas law (as given by the terms 'particle surface concentration' and 'pressure' in the figure caption). For this reason, the accuracy of the dynamic formalism is actually demonstrated by the figure, as the particle surface pressures tend to the gas phase partial pressures in the process of system equilibration.
These explanations should clarify the reviewer's points b), c), d), g) and h).

Concerning question a), the graphs were produced with the same fully dynamic approach that is used later on as a benchmark to demonstrate the accuracy of the hybrid solver, that is the Jacobson 1997 scheme used at fixed time step (see below).

This relates then to the reviewer's point e): The 10 second run does not oscillate because close to equilibrium the Jacobson scheme proves to be sufficiently stable when the time step is in accordance with the dynamical properties of the simulated system and its characteristic equlibration time interval. The oscillation is the result of an initial overshoot close to equilibrium, followed by consecutive of under- and overshoots. Both equilibrium regimes, the steady state and the artificial oscillation regime, are stable, meaning that if the system was sufficiently perturbed at equilibrium, it could quite well be that the system would oscillate at a time step of 10 seconds, just as it could remain at steady state at a time step of 30 seconds if brought sufficiently close to it.

As to point f), we have added an explantion of 'part' and 'atm' to the figure caption. Short additions were made to the figure caption (and within Section 3 for point e) that should correct and clarify the issues raised by the reviewer.

4) "What is the difference between "size bin" and "size increment"? If they mean the same thing, then please stick with "size bin." Otherwise, please clearly define a "size increment." "

The difference between size bin and increment is close to none, we have used them as synonyms. We scrapped both 'increment' and 'bin', and replaced them with the term 'class', for the reason that 'bin' is frequently used in the literature context in reference to a 'bin' in opposition to a 'modal' model, although this distinction might not be totally clear and appropriate either.

5) "Page 5, Line 16: What are the "dissolving" and "non-dissolving (=passive)" species considered in this work? It's not clear what a "non-dissolving" species even means. This terminology becomes especially confusing on page 8, lines 10-15, where a monoacid is treated as "dissolving" but the anions and cations are treated as "nondissolving" in Eq. 11. This equation is supposed to give H+ ion concentration from the difference between anions and cations, so I don't understand why the additional term "c_i" is even included here."

We changed the terminology, as 'non-dissolving' is scrapped for 'passive' to remain, which is now introduced by formal definition along with 'dissolving'. The monoacid within Eq. 11 is the only species whose semi-volatile and time-depending nature is considered at this point: it is the dissolving species while all others are passive. We feel this change in terminology and the adaptations to the text clarify the issue.

6) "The derivation of Eq. 9 is also very confusing. First, please clearly define the terms a, b, and c of the generic differential equation $dx/dt = ax^2 - bx + c$. Then show the solution using the same notations."

We partially followed the reviewer's request, and reformulated the expression of Equation 9, as we now state parameters a, b and c. However, it is not reasonably possible to formulate the solution as an explicit function of these parameters. This is the reason why parameters $\lambda$ and B had been introduced in the first place. Note that there was a typing error within Eq. 9, which is corrected as the parameter b is squared now inside the square root function.

7) "Will the numerator in Eq. 15 always be positive? What happens if a species in a given bin has a tendency to evaporate during a time step (e.g., HNO3 displacing HCl from sea salt)?"
It has to be ensured that the numerator of Equation 15 is always positive. Within the solver this is ensured via the use of the absolute value function. The fact that it was missing here is due to a typing error, which we corrected.

8) "How does the dynamic solver handle simultaneous mass transfer of H2SO4, HNO3, HCl, and NH3 to a size distributed aerosol? It seems mass transfer of H2SO4 is not included in the derivation of the dynamic solver equations. Also, none of the box model test cases include gas phase H2SO4 that condenses along with HNO3, HCl, and NH3."

H2SO4 is not currently treated by HyDiS as ( 1) it is assumed to be non-volatile and (2) the condensation of non-volatiles is formally considered to be a separate process. In principle it could be treated as a semi-volatile species by the solver if found appropriate and termodynamic data is made available, and provided that some slight adaptations to the chemical equilibrium solver are made, considering that the maximum number of chemically fully interacting species that may be accounted for by the solver may not exceed 3. Non-volatile species are treated within the combined condensation and nucleation routine of the microphysics scheme. HyDiS is the last routine to be invoked by the aerosol model. Concurrent condensation of H2SO4 and dissolution of HNO3 and NH3 is not shown within the box modelling experiments as we put priority on showing the transition of the system towards forced dynamic equilibrium. We have made changes within Section 4 and Section 5.2 for improved clarity with respect to these circumstances.

Question 9 is split in two:

9a) "While the present box model tests are useful in showing the benefits of the hybrid solver over the equilibrium approach, they do not demonstrate the accuracy of the new dynamic solver introduced here. To evaluate accuracy, it is necessary to compare the dynamic solver for monodisperse aerosol (similar to the example shown in Figure 1) and 4 size bins (as already shown in series 1 and 2) against a benchmark dynamic solver (e.g., LSODE) with strict error tolerances for a range of initial gas concentrations (of H2SO4, HNO3, HCl, and NH3) and aerosol sizes, concentrations, and compositions."

The reviewer assumes that the credibility of the benchmark solver that we use is not given and that an established stiff solver of ordinary differential equations, like lsode, should be used instead. We acknowledge that the reviewer's suggestion is justified in the context of insufficient explanation of our

methodology and the lack of clarity that surrounded Figure 1. However, although Zaveri et al 2008 used lsode as a benchmark, this is no absolute proof that it is the better choice in any case. We actually believe that the Jacobson 1997 Analytical Predictor for Dissolution (APD) we use as a benchmark is more appropriate, for the reasons that we will now expand upon. A solver may serve as a benchmark, if it is (1) mathematically precise, (2) formally different from the one it is compared against, and (3) compatible with the numerical formalism it is embedded in. First, although it has a propension for oscillations at larger timesteps, the Jacobson 1997 APD is mathematically sound, as by inspection for an internal timestep $\delta t \rightarrow 0$ its numerical integration method tends to the mathematically correct solution. Its capacity to deliver precise results is illustrated by Zaveri et al 2008 (though for the Jacobson 2005 semi-implicit version of the APD, which is formally similar to a large degree), as lsode and the APD tend to give similar results. Second, for Zaveri et al the use of the fully dynamic 1997 exponential APD as a benchmark was not appropriate as formally too close to its 2005 semi-implicit cousin. With HyDiS and its mixed dynamic and equilbrium approach, its selective setting to equilibrium of species within non-equilibrium bins and its use of the pseudo-transition approach, the fully dynamic APD is formally dissimilar. Third, the APD is the better match to HyDiS and its numerical formalism: (1) it is already in use in combination with an as large as possible timestep for species in bins that are simulated dynamically, a comparison between the hybrid and the fully dynamical runs may thus serve to demonstrate that HyDiS chooses the internal timestep without loss of *relative* accuracy, and (2) the APD actually tends to be more precise than lsode provided that the internal timestep is set to an appropriate value. lsode solves for all species within all size bins concurrently, while the APD solves for all size bins however separately for each species, which might appear to be a clear advantage for lsode. However, lsode keeps the parameters constant during the entire overall timestep $\Delta t$, while the APD updates them after each internal timestep $\delta t$. For a chemical process that involves relatively constant parameters lsode definitely is the better stiff solver, but for dissolution and its stiffness causing fast changing parameters, as given by the water content, the activity coefficients and surface pressure, it might not be. Quite on the contrary, it will introduce an amount of imprecision that will not be readily quantifiable. It is thus hard to tell whether the use of lsode is more accurate via the acccurate simultaneous solution of a system of stiff ODE's, or actually less accurate via the variability of the parameters. The only ways we can think of to overcome this uncertainty would be to (1) compare runs with activity coefficients and water content held constant, which might not be the best choice for the purpose of solver verification, or (2) attempt to formally resolve the higly non-linear properties of the parameters via their explicit inclusion as variables, which would be using a sledgehammer to crack a nut at best or might turn out to be not feasible at worst, or (3) drastically reduce the overall timestep at which lsode is used, to an order of magnitude that is more or less similar to the one that is used internally by the APD, which would be largely tantamount to the use of the APD as a benchmark solver, as demonstrated by Zaveri et al 2008. For all of these reasons we chose to use the Jacobson 1997 scheme, which is mathematically precise if the internal timestep is chosen appropriately. This we did by simple inspection, as sketched within Figure 1 for a monodisperse aerosol and total species amounts of 1 ppt: 30 seconds induce oscillation, and 10 seconds are precise to the extent that the simulation is graphically indistinguishable from a run at a much shorter timestep (therefore not shown). We have added a paragraph to the text in order to describe and motivate our choice of the APD as the benchmark solver.

9b) "The predicted equilibrium (after sufficient time) gas and aerosol concentrations (including aerosol pH) for the monodisperse test cases must also be compared to a benchmark thermodynamics model such as AIM (available online) to evaluate the accuracy of the thermodynamics treatment in the present solver. For example, see the dynamic solver evaluation done in Zaveri et al. (2008). Such an evaluation against a benchmark solver is especially warranted in the light of the several discrepancies found in the results of the dynamic solver shown in Figure 1 (see #3)."

The reviewer suggests that we use AIM as a benchmark for the equilibrium solver. Interestingly this is a possibility that we had considered before submission but ultimately had decided against it. From our point of view, the scope of this publication is not so much whether the equilibrium that is determined is accurate thermodynamically (this was done by Topping et al 2009), but rather whether the solver is numerically correct, which is demonstrated as fully dynamic and hybrid runs converge towards full equilibrium, thus validating each other mutually. Also, as pointed out by the reviewer, the online version of AIM does not allow for a size discretized aerosol. We would thus have to add another experiment with a monodisperse aerosol to an already lengthy paper. We have added one sentence detailing that the thermodynamic scheme was tested against AIM by others.

10) "The main goal of the present work is to introduce a new hybrid solver that is computationally efficient. But since any solver can be made efficient by compromising its accuracy, it becomes necessary to evaluate computational efficiency as a function of accuracy. The CPU costs presented for the various dynamic solvers in Table 1 are of little use without stating their accuracies against a benchmark solver (see #9)."

The reviewer asserts that the CPU times are of little use in the absence of a reliable reference as given by a benchmark solver. For the reasons given within 9a) we believe that the Jacobson 1997 APD, if used appropriately, actually is a valuable benchmark solver. No changes to the text apart those made in the context of point 9a).

11) "As the authors have already acknowledged, the comparison of the computational cost of the present solver with that of MOSAIC is fraught with many issues: different model configurations and chemical/physical complexities, computer hardware and computing architectures, etc. Also, the 125 μs CPU cost for MOSAIC quoted here includes microphysical calculations (in addition to gas-particle mass transfer and thermodynamics) whereas the 20 μs given for the present hybrid solver appears to be for gas-particle mass transfer and thermodynamics calculations only. Furthermore, the CPU costs given in Table 5 for cases 4-7 (aqueous particles only) range from approx. 20-40 μs (depending on the case, hardware, and compiler), which is more directly comparable to the CPU cost presented here for the hybrid solver. Having said that, the accuracy and efficiency of both solvers should be evaluated for the same set of problems on the same machine for the comparison to be meaningful."

The reviewer states that a comparison with MOSAIC in terms of CPU time as done within the manuscript is of questionable relevance due to dissimilarity of both the computing facilities that are

used and the schemes themselves. We agree that the relevance of the comparison is limited. However, in this respect we do not believe that the comparison would not carry meaningful information. The fact that we overlooked that for MOSAIC the microphysics are included in their estimation of CPUtime does not affect our conclusion that the expense should be of the same order of magnitude, thus demonstrating our due caution. We believe that our conclusion carries a useful piece of information to the reader, considering that a more detailed assessment is reserved to a global modelling framework that is more comparable, which is not currently available to us. The reviewer might have suggested to carry out the comparison with box model simulations instead, as done by Zaveri et al 2008. The relevance of box model comparisons of computation time is also limited, as these may not be representative to the parameter space that is encountered within a global modelling environment. Likewise, cases 4-7 within Zaveri 2008 might not be representative to the computational expense that is encountered in relationship with fully liquid particles, as the small particle sizes, which should require a smaller timestep, are not assessed with the given monodisperse setups. At this point, we have to mention that reviewer #1 seems to appreciate and does not criticize the way we established the comparison of CPUtime, on the contrary he would favour a similar comparison between the simulated and the observed aerosol composition. Within this partially conflicting situation we would like to leave it to the editor as to whether more CPUtime comparisons are essential, and as to how these should be done. In the meantime, we have further relativised our assessment within Sections 5.4 and also 6, in accordance with the reviewer's observations.

The following questions will be answered together, as related:

12) "Table 2: In GLOMAP sea-salt chloride is treated as non-volatile. Then how do H2SO4 and HNO3 condense on sea salt aerosol without evaporating HCl? Non-volatile treatment of HCl becomes problematic especially for condensation of HNO3, because both HNO3 and HCl are semivolatile and the extent of HNO3 partitioning crucially depends on HCl."
and 13) "How is HNO3 uptake on calcite containing dust particles treated in GLOMAP?"

The reviewer asks a question with respect to the representation of the volatility of HCl in the context of sea salt within Glomap, and the representation within Glomap of gas/solid phase exchange with respect to calcite. We are aware that these are important questions for the simulation of the atmospheric aerosol, although of limited relevance in the context of the present publication (see (1)). HyDiS may account for the chemical interaction of a maximum of 3 dissolving species. As such it may account for the interaction of HNO3, NH3 and HCl, provided that they are comprised within the atmospheric chemistry scheme of the hosting aerosol model. As mentioned within Section 5.2, we cannot treat HCl as a semi-volatile in the present version of Tomcat-Glomap. Gas/solid phase exchange is not currently part of Glomap and it is a process that is not treated by HyDiS, this also applies to calcite. We have made additions to Sections 5.2 and 6 that detail these limitations.

Minor Comments

p. 2 l. 10: order of acid and base is changed.

Fig. 11: resolution of figure is improved.

---

## Author Response (AR2)

Berlin, September 9 2016

Authors' Response to the Anonymous Reviewer 1

The authors would like to thank the Anonymous Reviewer 1 for their commitment to improve the quality of the manuscript.

All comments have been implemented.

Berlin, September 9 2016

Authors' Response to the Anonymous Reviewer 2

We thank the reviewer for their challenging comments.

We cannot follow the reviewer's assessment that the manuscript is difficult to understand. A huge amount of effort is committed to explain the solver in all of its detail. The degree of accuracy of the solver is explored. Its numerical reliability in a global modelling environment is demonstrated. Shortcomings are being acknowledged.

The alleged inaccuracies/inconsistencies of the solver are in part due to insufficient information provided by us. There are no inconsistencies. The degree of accuracy of the solver is adequately investigated through our method. In consequence, the reviewer's deduction of unverified efficiency does not hold.

However, we recognise that including an additional benchmark test for the equilibrium scheme contained within the new solver will enable the reader to make a more complete assessment of its relative precision

In order to accommodate the concerns from the referee's perspective, we have therefore added a new element to compare the equilibrium subsolver in the new solver to the benchmark equilibrium solver AIM III (Clegg et al., 1998). The additional experiments are described within a reorganised section 4.1 which now has a subsection 4.1.2 and 4.2.1 explaining these new benchmark tests which are presented in the new Figure 5.

We have also added additional benchmarking test runs against the fully dynamic (TRANS) simulation for both of the 100 ppb cases (sections 4.2.2 and 4.2.3) and provided additional data on particle composition, pH and size -- both to further demonstrate the accurate functioning of the solver.

1) *Figure 1. Thanks for clarifying the confusion surrounding Figure 1, but I am still not sure I fully understand it. In the caption it says that the initial aerosol is monodisperse with a dry radius of 50 nm, the RH = 80% and the aqueous phase contains no dissolved species. So is this a pure water droplet at 80% RH? How is this even possible, and what is the dry particle made up of?*
According to the definition on page 6 line 11, the term 'dissolved' is used here for convenience in the simplified way that it applies to semi-volatile species only. In a binary solution of $H_2SO_4/H_2O$, $H_2SO_4$ and $H_2O$ are both solvents and solutes. We took the perspective that they are solvents, because this is what distinguishes them from other species. The formulation is now modified to 'binary solution'.

2) *There appears to be something wrong in the predicted NH4 concentrations in Figure 5. According to the initial conditions given in the manuscript, I estimated about 9.2e-4 ppb of SO4 in the Aitken bin and about 0.023 ppb of SO4 in both the accumulation and coarse bins. Assuming the NO3 predictions are correct, then NH4 in the Aitken bin should not exceed 0.00184 ppb and about 0.046 ppb in the*

*accumulation and coarse bins, which correspond to full neutralization of SO4. Instead, the predicted values (by TRANS, HYBR, and PSEUDO) in the manuscript are 10 times higher.*

We forgot to mention that the particle sizes given in the text are particle number average sizes. Within each mode, particle size is assumed to be lognormally distributed. For this reason the sulpur content within the modes is not equal to the values assumed by the reviewer, but: Aitken - 0.019 ppb, accumulation - 0.48 ppb, coarse - 1.6 ppb. This is approximately 1 order of magnitude larger.

3) *The predictions by EQUIL model also do not make sense in Fig 5-7. All the bins should have the same NH4/SO4 and NO3/SO4 ratios at equilibrium (ignoring the Kelvin effect for the nucleation bin). In other words, all the bins should have the same fractional composition at equilibrium. Since accumulation and coarse bins have the same amount of H2SO4 initially, they should have identical NH4 and NO3 concentrations at equilibrium. However, the EQUIL predictions in the manuscript are not at all consistent with this expectation.*

In a similar fashion to the preceding, the NH4 and NO3 in the coarse and accumulation modes are not equal because their SO4 contents are not. The NH4/SO4 and NO3/SO4 ratios are strictly equal as they should be, as demonstrated by the following figures for SO4, NO3 and NH4, respectively, at 1 ppb without sea salt (Figure 5), in molecules per particle:

Aitken - 1832462.2021408435 / 284.53503104 / 854190.872703735
accumulation - 1.1727758093701398E+8 / 18209.7950076 / 5.466710685846889E+7
coarse - 3.87168734326824E+11 / 6.007979059412E+7 / 1.803829726551237E+11

We would also like to point to the NO3/NH4 ratio, which is in contradiction with the reviewer's deduction, as given by his assumptions.

For the sulphate series Figs. 5-7 we have added additional graphs showing the particle pH, and confirming that the particles' composition is equal and thus consistent.

4) *Quite large discrepancies are seen between TRANS (used as benchmark) and HYBR or PSEUDO are seen in Fig 9 for all species, but especially for Cl in the coarse bin. Moreover, no proper evaluation of the HYBR/PSEUDO models against a benchmark was shown for the high concentration cases (Fig. 7 and 10).*

We point to the explanation of the discrepancies in the manuscript. The graphs show the limitations of the solver, which is what they are supposed to do.

Comparison against a benchmark fully dynamic run is now provided in Figs. 7 and 10.

5) *Independent confirmation of both time evolution and equilibrium results are needed for all figures (1-10) to have any confidence in the dynamic model used by authors as benchmark. I also strongly suggest including additional monodisperse cases (similar to Fig 1) to clearly show that the dynamic models match the equilibrium predictions by an independent benchmark such as AIM.*

In our view the reviewer is mislead not to have any confidence in the benchmark model. It seems obvious to us that comprehensive mathematical proof for the reliability of the benchmark model is given through the following:

 i)     PSEUDO and TRANS are entirely independent of each other. HYBRID is apparented to both of them, with particularities of its own. Consequently, the dynamic runs, HYBR, PSEUDO, and TRANS **mutually** verify each other as they give similar results.

ii)     the dynamic runs converge to the equilibrium solution that is provided by the totally independent equilibrium solver.

Finally, the reviewer may also consider that the dynamic solver was shown to be reliable by Zaveri et al. for a sufficiently small time step, as is confirmed by this manuscript.

For additional confidence, we have added a comparison to AIM. As predicted by us, the limited discrepancy that we find may be entirely related to their dissimilar thermodynamic schemes. Mathematically speaking, AIM thus is an inappropriate benchmark for HyDiS, although a comparison is interesting as the overall degree of precision of the new equilibrium solver is shown.

6) *With the above discrepancies and errors in the dynamic and equilibrium predictions, it is difficult to make claims about computational efficiency. As I said in my previous review, any model can be made fast by sacrificing its accuracy. The authors have argued against doing a proper evaluation of speed vs. accuracy and comparison with existing models in the literature. As a result, the revised manuscript continues to be very weak in this area and their claim that their proposed solver is efficient has not been properly demonstrated.*

The alleged errors are shown to be inexistent. The degree of reliability of the solver is amply investigated. The relevance of these investigations is not being argued for, it is mathematically sound. In consequence the reviewer's deduction regarding the impossibility to make claims on computational efficiency is not valid.

7) *Finally, for the new solver to be worthy of publication it must have reasonable applicability and longevity. As it currently stands, it does not handle mass transfer coupled with phase transition and heterogeneous reactions. So the question is will this solver still work when these missing processes are included?*

We cannot see any reason why the solver would not work if these processes are included. The mentioned processes may be formally separated from the ones treated by HyDiS. The new solver was shown to be numerically robust to the perturbations these processes just like any other process might cause.

8) *It's still not clear what the "passive" anions and cations mean? Please, just give an actual list of all the ions (and their designations – active, passive, etc.) in a Table.*

According to our definition on page 6 line 12, the term 'passive' does not designate any ions per se, but rather is used conveniently for species whose semi-volatile nature or chemical interaction is chosen to be ignored at one particular point.

As it seems to stir controversy, the term is now dropped.

[revised manuscript text omitted]

NH₃

H₂SO₄   HNO₃

insoluble Aitken N₅
BC OC

insoluble accum N₆
DU

insoluble coarse N₇
DU

↑ OH
SO₂
↑ OH, NO₃
DMS

soluble nucln N₁
SO4 OC
NH4 NO3

soluble Aitken N₂
SO4 OC
BC
NH4 NO3

soluble accum N₃
SO4 OC
BC Na DU
NH4 NO3 Cl

soluble coarse N₄
SO4 OC
BC Na DU
NH4 NO3 Cl

29 mass
7 number

transported
tracers=36

**(b) GLOMAP-mode standard configuration**

MONOTER

Species in internally mixed modes:

**sulphate**, **sea salt**, **black carbon**, **organic carbon**, **dust**,
**secondary organic**

↓ OH, NO₃, O₃
SEC_ORG

H₂SO₄

insoluble Aitken N₅
BC OC

insoluble accum N₆
DU

insoluble coarse N₇
DU

↑ OH
SO₂
↑ OH, NO₃
DMS

soluble nucln N₁
SO4 OC
SO

soluble Aitken N₂
SO4 OC
BC SO

soluble accum N₃
SO4 OC BC
SS DU SO

soluble coarse N₄
SO4 OC BC
SS DU SO

[revised manuscript text omitted]

---

## Author Response (AR3)

Size-resolved simulations of the aerosol inorganic composition with the new hybrid dissolution solver HyDiS-1.0 – description, evaluation and first global modelling results

Benduhn et al.

Response to the Anonymous Reviewer

3$^{rd}$ Revision

We thank the anonymous reviewer for their continued support in improving the quality of the manuscript.

The reviewer has pointed to the need for a better distinguishability of the cyan line in Figs. 9-11 and suggested the use of symbols for this line. We did not follow the reviewer's suggestion for several reasons. First, we believe that the use of symbols for hybrid only is problematic, because it may suggest that the time discretion of this run would be different from others. Symbols are often associated with discreet measurment data vs. model results. In order to avoid this ambiguity, symbols would have to be used throughout. The overall use of symbols influences the readability of the graphs negatively due to symbol superposition. Second, not only is hybrid difficult to distinguish from pseudo, also trans is difficult to distinguish from hybrid, especially in Fig. 6. Sometimes, trans, hybr and pseudo are all three barely distinguishable. The use of symbols would thus also be required for trans, thus again leading to symbol superposition and poor readability. For these reasons, we now specify line similarity within the figure captions, if necessary, thus reflecting the reviewer's request.

We now point to model bias within the abstract, as requested by the reviewer.

[revised manuscript text omitted]